# Asymmetrical estimator for training encapsulated deep photonic neural networks

Yizhi Wang[1], Minjia Chen [1], Chunhui Yao [1,2], Jie Ma[2], Ting Yan[2], Richard Penty[1] & Qixiang Cheng [1,2] ✉

Photonic neural networks (PNNs) are fast in-propagation and high bandwidth paradigms that aim to popularize reproducible NN acceleration with higher efficiency and lower cost. However, the training of PNN is known to be challenging, where the device-to-device and system-to-system variations create imperfect knowledge of the PNN. Despite backpropagation (BP)-based training algorithms being the industry standard for their robustness, generality, and fast gradient convergence for digital training, existing PNN-BP methods rely heavily on accurate intermediate state extraction or extensive computational resources for deep PNNs (DPNNs). The truncated photonic signal propagation and the computation overhead bottleneck DPNN's operation efficiency and increase system construction cost. Here, we introduce the asymmetrical training (AsyT) method, tailored for encapsulated DPNNs, where the signal is preserved in the analogue photonic domain for the entire structure. AsyT offers a lightweight solution for DPNNs with minimum readouts, fast and energy-efficient operation, and minimum system footprint. AsyT's ease of operation, error tolerance, and generality aim to promote PNN acceleration in a widened operational scenario despite the fabrication variations and imperfect controls. We demonstrated AsyT for encapsulated DPNN with integrated photonic chips, repeatably enhancing the performance from in-silico BP for different network structures and datasets.

Neural network (NN)-based machine learning algorithms are prevalently used in the state-of-the-art (SOTA) industry and academic research[1–3]. The surging need for further acceleration of NN-based applications calls for emerging paradigms such as physical neural networks (PhyNNs), offering high-bandwidth in-propagation analogue computation[4,5] on top of the increased parallelism of dedicated digital hardware such as GPU[6] or TPU[7].

Photonic-based analogue neural networks (PNNs) belong to the subclass of scalable isomorphic PhyNNs[5,8], which enjoy the complexity of PhyNN while maintaining the repeatability and reproducibility to provide acceleration for a wider range of users. Photonics is widely known for its high bandwidth and multiplexing ability in multiple domains[9–12]. The matrix-vector multiplication accelerator with both on-chip[13,14] and free-space[15,16] optics already reveals great potential for photonic processing[17,18] in general. Demonstrations of integrated PNNs have shown superior performance in both processing speed[19] and energy efficiency[8]. These miniaturized integrated PNN systems hold great potential due to their fabrication reproducibility and compactness. Nonetheless, unlike pure digital deep neural networks (DNNs), where the network construction is a general mathematical description, PNN implementations are codesigning processes between the task, the PNN device platform, the application environment, and the training

[1]Centre for Photonic Systems, Electrical Engineering Division, Department of Engineering, University of Cambridge, Cambridge, UK. [2]GlitterinTech Limited, Xuzhou, China. ✉e-mail: qc223@cam.ac.uk

algorithm. The difficulty of training PNN due to the device-to-device and system-to-system variations often means a complex and demanding codesign process.

Backpropagation[20] (BP) finds the gradient update to the loss function for the network parameters, offering fast training gradient convergence[21]. BP's advantage for fast training with DNNs comes from two key aspects: BP's root in gradient-based update logic and BP's layer-by-layer update dynamics. BP's robustness, generality and fast convergence make it a well-balanced industry standard approach for most digital DNN-based tasks. However, standard BP imposes requirements of highly accurate model control and knowledge. The inevitable fabrication variation inherently creates imperfect knowledge for the users. In addition, the lack of perfect control-transformation mapping on the typically noisy and lossy analogue computation platforms[22,23] can't satiate the high requirements of standard BP. The mismatch between the user's interpretation and the PNN's actual state can lead to failed training or dramatic performance degradation[24,25]. Therefore, some non-gradient-based[26–31] or model-free/stochastic methods[32–35] are proposed to bypass the requirement for formulating a model description. Nonetheless, the natural trade-off with non-gradient methods is the higher difficulty in training complex tasks and slower convergence compared to the BP-based methods. When defining the convergence of standard BP $O(T_0)$, the convergences of other non-gradient PNN training methods are typically $>O(T_0)$[36]. PNNs, being mostly operation-centred platforms, are highly compatible with BP-based gradient methods for the typically exhibited isomorphism (see Supplementary Note 12).

Many efforts have been made to search for PNN-tailored BP-based methods. The general attempt of the BP-PNN methods is to find an estimator of the device's physical parameter gradient. The approaches which utilize a separate digital model or propagation pass for estimating the gradient update include physics-aware training[37] (PAT), hybrid training[38] (HT), and dual adaptive training[39] (DAT). There are also approaches which utilize the physically reversed signal input through the structure to obtain in-situ computation of the gradient[40–42]. However, the common limitation of these existing BP-PNN methods is either the heavy reliance on accurate intermediate layer state extraction in a deep PNN (DPNN) (Fig. 1 a) or the extensive computational resources to simulate the training model. We call methods with the need to access internal information intermediate-access physics-aware backpropagation (IP-BP) methods. Due to the intermediate access need, IP-BP codesigned PNNs are truncated, in which the computation acceleration is only available within a hidden layer's structure. For fast propagation platforms like photonics, the overall operation is bottlenecked by the analogue-digital (AD) conversion interfaces[43,44] and data shuttling[45], creating a delay growing with the network complexity (Fig. 1 b). This potentially compromises the incentive of employing photonics for fast and low-delay processing. Instead, it is more desirable to construct encapsulated DPNNs (encapsulation refers to systems whose input signal is maintained within the optical analogue domain without intermediate extraction), allowing the advantage of fast processing to be enjoyed for the entire deep network structure. The number of AD access in a truncated DPNN grows as $O(2M - P)$ (M is the total number of neurons excluding the input layer, and P is the number of output layer neurons, M>P for any DNN), whereas it scales as $O(P)$ for encapsulated DPNN. The access timestep is reduced from $N+1$ (N is the number of hidden layers) in a truncated network to 1 for an encapsulated network (see "Discussion" section.) Equally, it is undesirable for the training algorithm's complexity to be high: the additional computations increase training time, energy consumption, and cost. The higher complexity would also imply the difficulty for application on reproducible platforms, as the complementary control system's overhead is too high.

Consequently, it is desirable to find methods that bypass the IP-BP methods' access limitations, train with reduced computation complexity, and still enjoy BP's advantages. Here, we present the asymmetrical training (AsyT) method for well-balanced training of DPNN systems with single-structured design (doesn't require special structure for training, needs the minimum photonic component requirement same as the inference for reduced cost), encapsulated computation (the signal is maintained within the optical domain for the entire DPNN structure), error-tolerance (tolerant to PNN's device-to-device and system-to-system variations), and low computation resource requirement (comparable to standard BP). AsyT utilizes an additional forward pass in the digital parallel model compared to the existing estimator approaches[37–39] to eliminate the requirement for accessing intermediate DPNN state information (for a total access point of P). AsyT's gradient-like layer-specific update dynamic is compatible with standard update optimizers such as gradient descent[46,47] or Adam[48]. AsyT's goal to increase training efficiency and reduce cost is in unison with PNN's general purpose for faster and cheaper computing[9].

We demonstrate the AsyT method with encapsulated DPNN utilizing photonic integrated circuits (PICs) as the physical processing module (PPM). In this paper, we start by explaining the concept and working principle of the AsyT algorithm. We then experimentally demonstrated AsyT for a fully encapsulated DPNN for classification. We further experimentally validated AsyT's ability to achieve training with only output neuron information for different tasks (Iris-flower classification and modified hand-written digit classification) and scaled-up structures through repetitive use of PPMs free of local characterization. The scalability and repeatability of AsyT are analysed with more complex datasets such as digit-MNIST[49] (95.8%), fashion-MNIST (FMNIST[50], 87.5%) and Kuzushiji-MNIST[51] (KMNIST, 85.6%) through simulation under experimental level error, showcasing the repeating enhanced performance from in-silico BP. The results achieved are close to the benchmarks of ideal error-free BP training. We systematically investigated AsyT's performance for varying network structures and showed its high error tolerance. We finish by discussing the advantages of constructing encapsulated DPNN with AsyT, alongside a discussion of the applicational scenarios of AsyT and how most of the computational overheads of AsyT can be bypassed for overall efficient PNN training and operation.

## Results
### Asymmetrical training method
The general mathematical description of a time-independent multi-layer perceptron (MLP) system is expressed in Eq. (1). The net output $\mathbf{z}$ for a given layer $[l]$ is determined by the connection strength of $\mathbf{W}$ and bias $\mathbf{b}$ acting on the activation level $\mathbf{a}$ from the previous layer. The neuron's activation level is modulated by some nonlinearity $g(\cdot)$. The overall mathematical expression is summarized in Eq. (1).

$$\mathbf{z}^{[l]} = \mathbf{W}^{[l]}\mathbf{a}^{[l-1]} + \mathbf{b}^{[l]}; \mathbf{a}^{[l]} = g^{[l]}\left(\mathbf{z}^{[l]}\right) \tag{1}$$

For $l = 1, 2, \ldots, N+2$, where N is the total number of hidden layers, the forward propagation of information leads to a prediction at the output layer. Under perfect information, we summarize the total mathematical transformation of a hidden layer as $f_m(\mathbf{a}; \mathbf{W}_m, \mathbf{b}_m, g_m)$ for incoming activation $\mathbf{a}$, subject to the specific parameters of $\mathbf{W}_m, \mathbf{b}_m$ and $g_m(\cdot)$. For an isomorphic PNN, where the hardware topology and perturbation-able parameters resemble the neuron connectivity, the physical transformation for a physical module of a single hidden layer can be expressed as $f_p(\mathbf{a}; \mathbf{W}_p, \mathbf{b}_p, g_p)$. The two transformations of the digital (mathematical) and analogue (physical) domains are not equivalent even when set with the same parameters as they are described by different functional spaces ($f_m(\mathbf{a}; \mathbf{W}, \mathbf{b}, g) \neq f_p(\mathbf{a}; \mathbf{W}, \mathbf{b}, g)$). The overall transformation of a deep

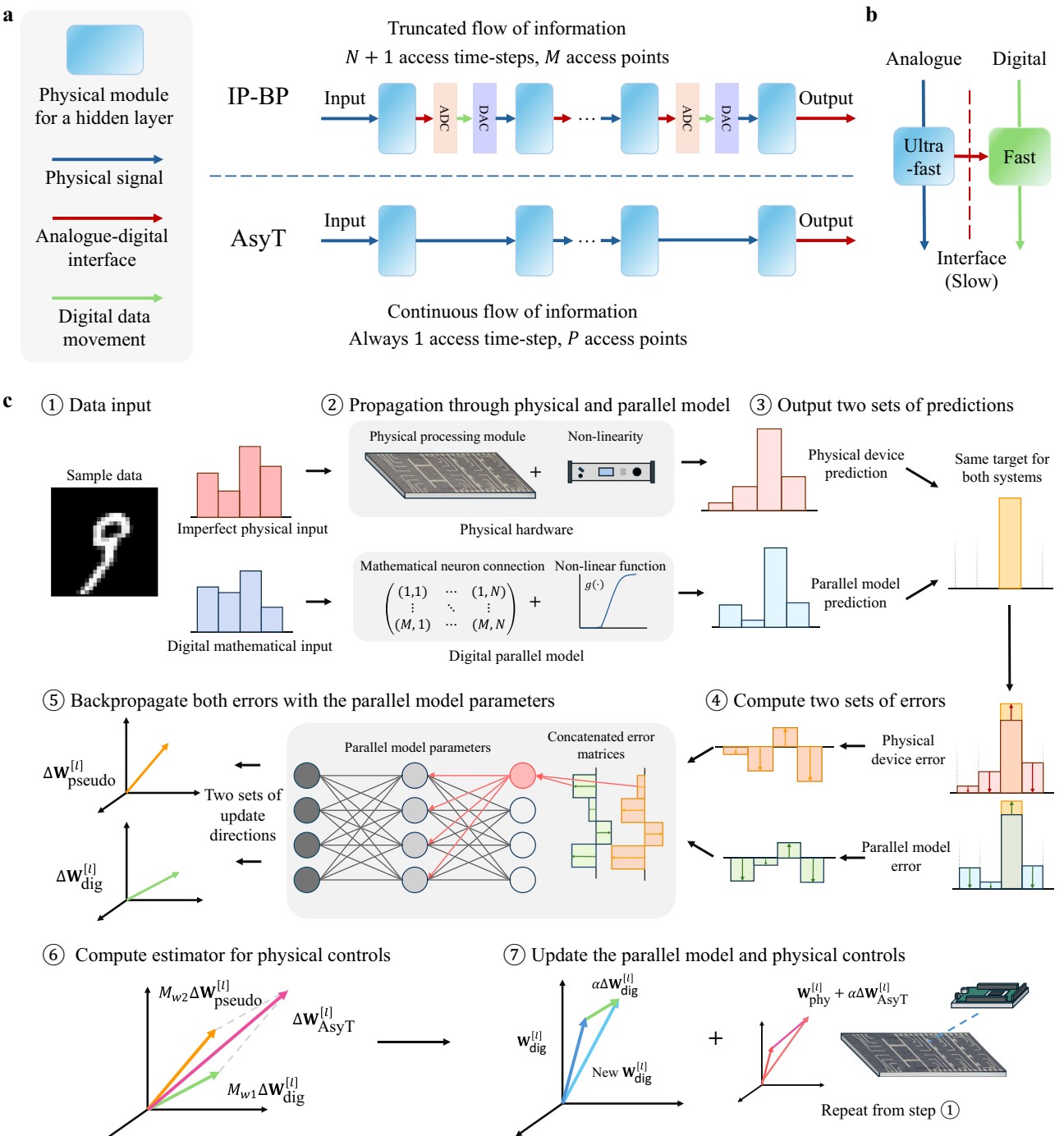

**Fig. 1 | Concept of the AsyT method. a** AsyT only requires the neuron information at the output layer, allowing for the continuous propagation of physical information in the entire DPNN structure. For IP-BP methods, the analogue information propagation is truncated at each hidden layer, not fully utilizing the advantage of fast PNN processing. Moreover, the introduced AD interface bottleneck time scales with $O(2M - P)$ for IP-BP methods, but as $O(P)$ for AsyT. **b** Computation solely within the analogue or digital domain is optimized compared to the interface. AsyT uses the additional digital parallel pass to avoid information traffic at the interface while compensating for the fewer physical information accesses. **c** Workflow for using the AsyT method. AsyT utilizes both the forward and backward passes of the digital parallel model. The digital model's operation is general to a set of statistically varied PNNs, requiring only one computation for a task to allow lowered distributed overhead (see "Discussion" section).

network structure is $f_{m,1}(f_{m,2}(f_{m,3}(\ldots)))$ for the mathematical NN, and $f_{p,1}(f_{p,2}(f_{p,3}(\ldots)))$ for the DPNN structure. For perfect information digital DNN, we know the full description of $f_m(\cdot)$, that is the exact response relation of the resulting transformation for perturbating each parameter. This means that we can easily perform auto-differentiation of the loss function $L$ with respect to the parameter in each hidden layer and update parameters with a gradient-based strategy such as Eq. (2).

$$\mathbf{W}^{[l]} = \mathbf{W}^{[l]} - \alpha\Delta\mathbf{W}^{[l]} \qquad (2)$$

However, in a DPNN with only access to output neuron information, the encapsulated total transformation $f_{p,1}(f_{p,2}(f_{p,3}(\ldots)))$ can't

provide separation of transformation $f_{p,i}(\cdot)$ for the individual physical hidden layer $i$ to interpret the layer-wise effect. This is the reason why IP-BP-based methods[37–40] rely on knowing the intermediate state of the deep network structure. Consequently, only utilizing the backward pass in the digital domain is not sufficient for well-behaved training to co-construct an encapsulated DPNN.

Instead, AsyT utilizes an additional forward pass in a parallel digital model to avoid this problem. The complete set of the forward and backward pass in the digital domain provides a regulated reference point for projecting the encapsulated physical transformation towards the mathematical (digital) transformation $f_{m,1}(f_{m,2}(f_{m,3}(\dots)))$, which we know has a loss-reducing behaviour as the update is purely formulated by BP and not affected by the local PNN copy's variation. As we have high integrity knowledge of $f_{m,i}(\cdot)$, AsyT's update at each instance is modulated by the digital model's BP process and protected from sudden shock to the system.

To understand the AsyT method, we first define two separate sets of parameters for the parallel model $\mathbf{W}_{(t,\,\text{dig})}$ and physical control $\mathbf{W}_{(t,\,\text{phy})}$ (we refer to all the controllable parameters in the following discussion for simplicity). At the initialization $t = 0$, we set the digital and physical control with the same randomized values, $\mathbf{W}_{(0,\,\text{dig})} = \mathbf{W}_{(0,\,\text{phy})}$ (the two resulting transformations are different: $f_m(\mathbf{W}_{(0,\,\text{dig})}) \neq f_p(\mathbf{W}_{(0,\,\text{phy})})$). The digital and physical systems simultaneously forward-propagate the same sample information, resulting in a pair of predictions $\mathbf{a}_{(0,\,\text{dig})}^{[N+2]}$ and $\mathbf{a}_{(0,\,\text{phy})}^{[N+2]}$. The pair of predictions leads to a pair of errors ($\mathbf{E}_{(0,\,\text{dig})}$ and $\mathbf{E}_{(0,\,\text{phy})}$) at the output layer for the same targets. Despite there being no access to intermediate neuron information of the physical system, we have the complete information of the parallel model ($\mathbf{a}_{\text{dig}}, \mathbf{W}_{\text{dig}}, \mathbf{b}_{\text{dig}}, g_{\text{dig}}$).

The collection of parallel model parameters is used for backpropagating the two sets of errors. We define two update terms based on backpropagating the two errors $\mathbf{E}_{(\text{dig})}$ and $\mathbf{E}_{(\text{phy})}$. When backpropagating $\mathbf{E}_{(\text{dig})}$, we get the gradient update to the digital parallel model $\Delta\mathbf{W}_{\text{dig}} = \partial L / \partial\mathbf{W}_{\text{dig}} \equiv \delta\mathbf{W}_{\text{dig}}$. On the other hand, results from backpropagation of $\mathbf{E}_{(\text{phy})}$ have no direct physical significance on its own; we call it the pseudo update $\Delta\mathbf{W}_{\text{pseudo}}^{[l]}$. The AsyT estimator is defined as the mixture of these two updates under the regulation of the mixing ratios $M_{w1}$ and $M_{w2}$, which normally takes equal values $M_{w1} = M_{w2} = 0.5$ (see "Methods" and Supplementary Note 8). The overall expression is shown in Eq. (3), where the digital parallel and the pseudo updates are mixed at each layer for the AsyT update. (See Methods and Supplementary Note 7).

$$\Delta\mathbf{W}_{\text{AsyT}}^{[l]} = M_{w1} \cdot \Delta\mathbf{W}_{\text{dig}}^{[l]} + M_{w2} \cdot \Delta\mathbf{W}_{\text{pseudo}}^{[l]} \tag{3}$$

At every later instance ($t > 0$), the digital parallel model is trained purely by BP's gradient update $\Delta\mathbf{W}_{\text{dig}}^{[l]}$. PNN's physical control parameters are updated with $\Delta\mathbf{W}_{\text{AsyT}}^{[l]}$. We see that the pseudo update is always non-identical to the digital update for non-ideal PPMs. Consequently, the control settings to the digital and physical system become asymmetrical ($\mathbf{W}_{(t,\,\text{dig})} \neq \mathbf{W}_{(t,\,\text{phy})}$, for $t > 0$) as the training progresses, giving the name of this method. The key idea behind AsyT is that the initially identical control parameters lead to different resulting transformations. The later training updates attempt to use asymmetrical settings to correct the transformation of the photonic system towards the mathematical description in the parallel model. (See more discussion in Methods.) As long as the learning rates are the same for both systems at a given time instance, they can vary as time progresses, making the AsyT estimator compatible with conventional optimizers[46–48]. The overall workflow of the AsyT method is illustrated in Fig. 1c.

In AsyT's workflow, acquiring the digital update is independent of the local training of the PNN device (and the AsyT update is simply an additional operation between the pseudo and digital update). The digital update can be computed separately prior to PNN's local

training. Thus, the local PNN training has the same time-step as a standard BP to avoid any sequential delay. The same set of digital updates can be repeatedly used to train multiple local PNN copies. Consequently, the computational overhead of the parallel model spreads across PNN copies, reducing the local computational overhead associated with each device.

AsyT's training workflow allows for the construction of an encapsulated DPNN that requires a reduced neuron state access ($P < M$) and information access time-step ($1 < N + 1$). AsyT distributes the workload across the digital and photonic domains, bypassing most AD interfaces, to avoid bottlenecks. The encapsulation of PNN allows the fast computation of photonic platforms to be fully exploited, a highly desirable feature to promote PNNs (see "Discussion" section).

## AsyT method for training fully encapsulated DPNNs

We demonstrate AsyT's ability to train fully encapsulated DPNNs experimentally through implementation with PIC devices. A 4 × 4 SiN PIC with the broadcast-and-select topology (see Supplementary Note 4 for topology) is used as a PPM resembling a fully connected topology between nodes. We demonstrate the training of a classifier for differentiating the two Iris flower[52] species of Setosa and Versicolor based on the four input features. With the broadcast-select topology of the PIC, the neuron information is split into equal copies by the $1 \times 2$ multimode interferometers (MMIs). The accumulation stage is similarly based on the $2 \times 1$ MMIs. The strength of neuron connection to the next layer is defined by a Mach–Zehnder interferometer (MZI) cell with thermos-optic phase shifters through power transmission.

For constructing the encapsulated DPNN, the propagation of the analogue signal needs to be continuous from the input to the output layer without any intermediate readouts (Fig. 2a). We utilize two copies of the PIC devices to represent the two connections for a deep network structure. The packaging of the PIC and an enlarged picture can be found in Fig. 2c, d. A non-linear response is required as part of the network structure to generalize towards the separation of non-linear data. We use erbium-doped fibre amplifiers (EDFAs) with high current near saturation to generate a non-linear response for the input signal (see Supplementary Note 1 and 2 for setup details). The overall schematic for implementing the experimental setup is shown in Fig. 2b. The input information is modulated and passed onto the first chip for the first set of neuron connections. The output signal out of the first chip is directly passed to the EDFAs before going through tunable filters centred roughly at 1550 nm. The signal after the photonic nonlinearity is passed to the second PIC directly to implement a second neuron connection. This setup allows the signal to be maintained within the analogue domain until the readout at output neurons for classification.

Conventional PNN training requires precise and detailed system-level characterization to control the physical parameters. However, when the entire deep structure is encapsulated, the input-to-output response is unreliable for separating the effects from different layers. Consequently, the characterization of an encapsulated DPNN requires intermediate neuron access information. To rigorously construct full-encapsulation, where the local (device-specific) hidden neuron information is not accessed during any stage of the training, we employ an estimation profile for applying the physical control parameters. The estimation profile is a sensible statistical deduction of the PPM's control transformation without considering the local variation of individual devices (characterization-free). In addition to the estimation profile's ability to avoid intermediate readouts, it also brings the advantage of reduced memory requirement compared to standard fully characterized training. The total number of control parameters scales roughly as $\propto M^2$. For a finite resolution of $R$ of the analogue system control and neuron size of $M$, the required memory scales as $O(RM^2)$ for lookup tables in fully characterized training. In

**a** Fully-encapsulated PNN for continuous in-propagation processing

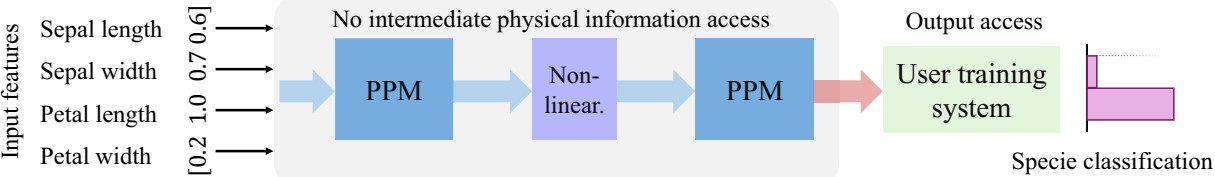

**b** Experimental setup for implementing fully encapsulated deep PNN

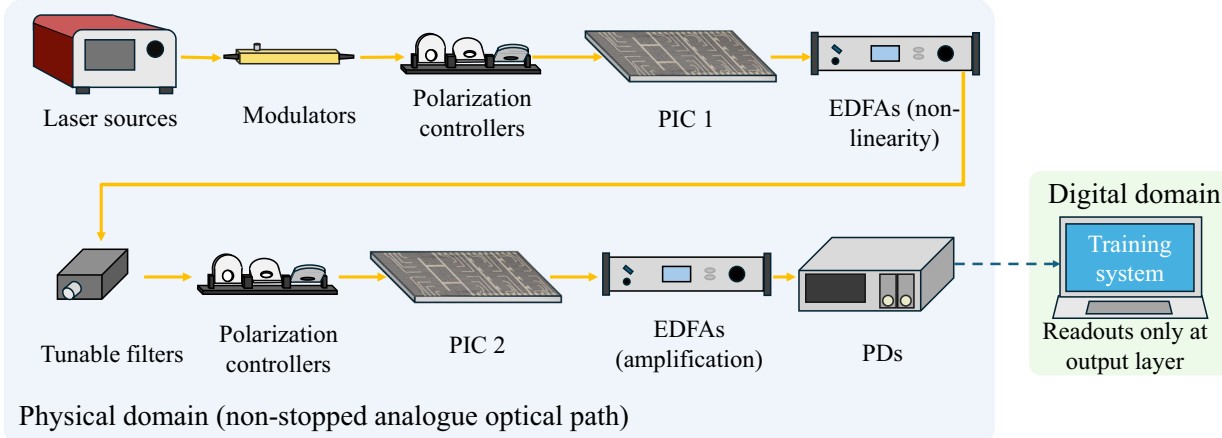

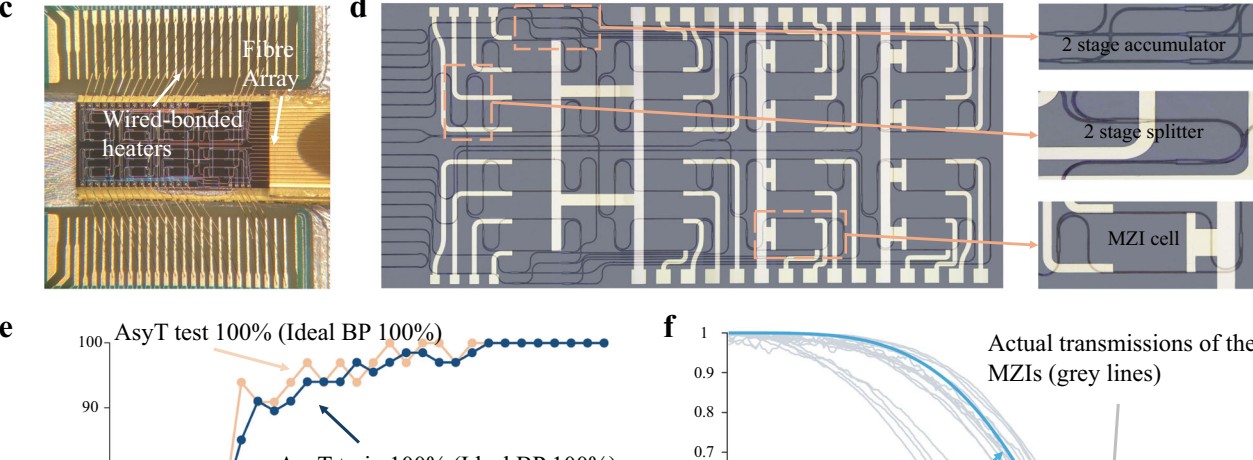

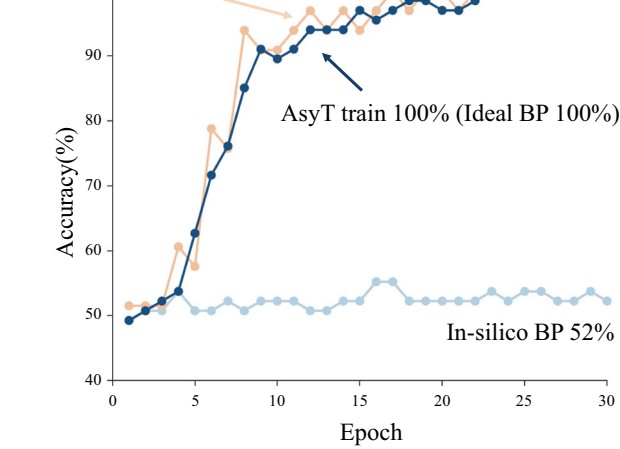

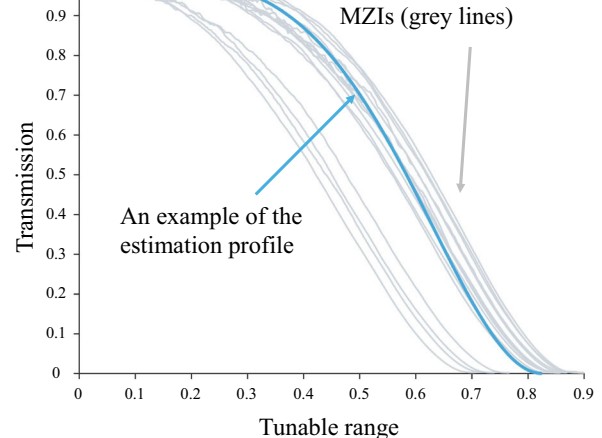

comparison, the estimation profile is applied to all controllable physical parameters, offering a reduced memory requirement $O(R)$.

Taking the construction of PIC-PNN here as an example, the neuron connection strength is determined by the transmission of the MZI unit, which is directly controlled by the power added to the thermos-optic phase shifter via voltage $V$. We can thus formulate an estimated guideline of how to apply the physical parameter. The transmission of the MZI cell can be described theoretically with a simple cosine-like shape as Eq. (4), where $\gamma$ is some constant (see Supplementary Note 5). This profile can be obtained without the need to perform any local characterization.

$$T = \frac{1}{2}\left(1 + \cos\left(2\gamma V^2\right)\right) \qquad (4)$$

**Fig. 2 | Results of AsyT for training fully encapsulated DPNN. a** The schematic of a fully encapsulated DPNN, where the analogue signal propagation is preserved for the entire deep network structure and only readout at the output layer. **b** Experiment setup for demonstrating the fully encapsulated DPNN. Two separate PIC devices are used to represent the deep neuron connectivity. The yellow arrows indicate the propagation of the optical signal. **c** Packaging of the PIC, showing the wire-bonded heaters and edge coupling to the fibre array. **d** A photo of the PIC device, with the enlarged components of the accumulator, splitter, and MZI cell.

The schematic can be found in Supplementary Note 4. **e** The training result of the fully encapsulated DPNN. AsyT (dark blue line: AsyT train; orange line: AsyT test) shows significant improvement in performance compared to in-silico BP (faded blue line). **f** An estimation profile based on the theoretical estimation of the MZI unit response, showing the large mismatch between the estimation profile (light blue line) and the actual transmission behaviours of the MZI cells (faded grey lines in the background). PPM physical processing module.

Figure 2f shows the estimation profile's shape. The grey lines in the background indicate the MZIs' actual control-transmission behaviours, showing significant differences from the estimation profile employed. Most PNNs are operation-based systems, where sufficient information for constructing a theoretical estimation profile is inherently available from the fundamental fabrication information, preserving the completeness of the optical encapsulation.

To benchmark AsyT's performance improvement, we first analyse the performance of in-silico BP training (referring to the process of purely training the network in a digital model and applying the parameters to the PPMs with the estimation profile). Foreseeably, due to the significant mismatch between the estimation profile and the actual transmission controls (equivalently, the mismatch between the expected transformation $f_m(\mathbf{W})$ and the actual physical transformation $f_p(\mathbf{W})$), in-silico BP training fails to classify the species. The resulting accuracy is 52%, close to the random guessing level for the binary classification task.

On the other hand, when the encapsulated DPNN is trained with AsyT, performance is significantly enhanced to 100% for training and 100% for testing after 30 epochs. The ideal BP performance (the maximum achievable accuracy when the same network structure is trained with BP solely in the digital domain without error) is also 100% for training and 100% for testing. The results are shown in Fig. 2e. AsyT showcases its ability to train fully encapsulated DPNNs, significantly improving the training performance from in-silico BP while not breaking the continuity of the analogue signal propagation.

### AsyT for training DPNN with only output neuron information
The premise for constructing encapsulated DPNN with AsyT is its ability to train solely with the output layer neuron information. Consequently, we further experimentally demonstrate AsyT for different tasks when the training system only has access to output layer neuron information. For the following demonstrations, we add the photonically plausible non-linearity digitally to the DPNN. However, intermediate neuron information is not provided to the training algorithm at any stage; further errors are deliberately added where appropriate for replicating the scenario of signal degradation in a fully encapsulated DPNN (see "Methods" and Supplementary Note 3).

We first showcase AsyT for the complete three-class classification of the Iris dataset for Setosa, Versicolor, and Virginica. To showcase the insufficiency of the sole backward pass in the parallel model of IP-BP methods for encapsulated systems, we further define a benchmark called the pseudo-IP-BP methods, referring to the scenario where IP-BP methods only have access to the output neuron information.

In-silico BP training is insufficient to train the DPNN, resulting in a significantly degraded performance of only 38%, a value near the random guessing level of the three-class task. Pseudo IP-BP has access to some level of physical response at the output layer. Consequently, the performance is slightly improved compared to pure in-silico BP, achieving 64% during training (Fig. 3a). However, the absence of the intermediate hidden neuron information implies that the differentiable model in the backward pass is no longer a good description of the actual physical transformation, showcasing IP-BP methods' inability to train encapsulated PNNs.

In comparison, when AsyT is applied, the training accuracy is improved to 96% and testing accuracy to 94%. For reference, an ideal BP achieves the maximum training accuracy of 97% and testing accuracy of 94% for the same network structure (Fig. 3a). In both cases, AsyT can retrieve near-ideal BP performances. The improvement of AsyT accuracy (96%) from pseudo-IP-BP methods (64%) showcases the necessity to employ the additional forward pass in the digital domain.

We further study the mechanism for training with AsyT by investigating the alignment angle and magnitude difference between the digital and physical transformations (Fig. 3c). AsyT quickly reduces the alignment angle between the two models and then holds it at a low level that allows the training of the dataset. We repeat the training experiment with additional estimation profiles to validate the generality of the estimation profile used. (Details of the additional estimation profiles can be found in Supplementary Note 6.) As shown in Fig. 3d, all repeats reach 96% training accuracy, showcasing the generality of different estimation profiles to retrieve near the ideal BP performance. We can equally interpret the generality of estimation profiles as equivalent to applying the same digital model for different copies of the PNN device. This allows the generalized computation of the digital update to be mixed with the locally computed pseudo update for local training with distributed and reduced computational overheads (see "Discussion" section).

### AsyT's compatibility with scale-up techniques
Due to the typically fixed dimensionality of the PPM, the construction of an isomorphic layer of PNN for complex tasks often relies on some scaling techniques in either spatial[8] or temporal[44,53] domains (Fig. 4a). With sufficient copies of PPM, the encapsulated DPNN can theoretically be constructed to the desired network topology. Nonetheless, using multiple PPMs to represent a hidden layer of the DPNN also implies further error accumulation within each layer. We implement a more complex hand-written digit classification task, where two representations of the PIC PPM are required to embody the neuron connection 8 × 4 in one of the layers, a dimensionality higher than the individual PPM. The four-class classification task is constructed for classifying the hand-written digits of 0, 1, 2, and 3. A similar experimental setup and the same estimation profile are used for training (see "Methods" details on the dataset and experiment setup.)

With in-silico BP, the accumulation of errors within each layer exacerbates the performance degradation, leading to a training accuracy of 29%. This degraded performance is close to the task's random guessing level (25%). In comparison, AsyT achieves a training accuracy of 84% and a testing accuracy of 82%. The ideal BP performance is 85% for training and 83% for testing (Fig. 4b, c). AsyT again retrieves performance near the ideal BP level. AsyT treats the collection of PPMs as one large structure and is not misdirected by the error accumulation from PPMs. Consequently, AsyT showcases its compatibility with scaling-up techniques for application to complex tasks and larger hardware dimensions.

### Generalization and repeatability of the AsyT method
We further investigate AsyT's generalization and repeatability across different datasets and larger network structures through simulation.

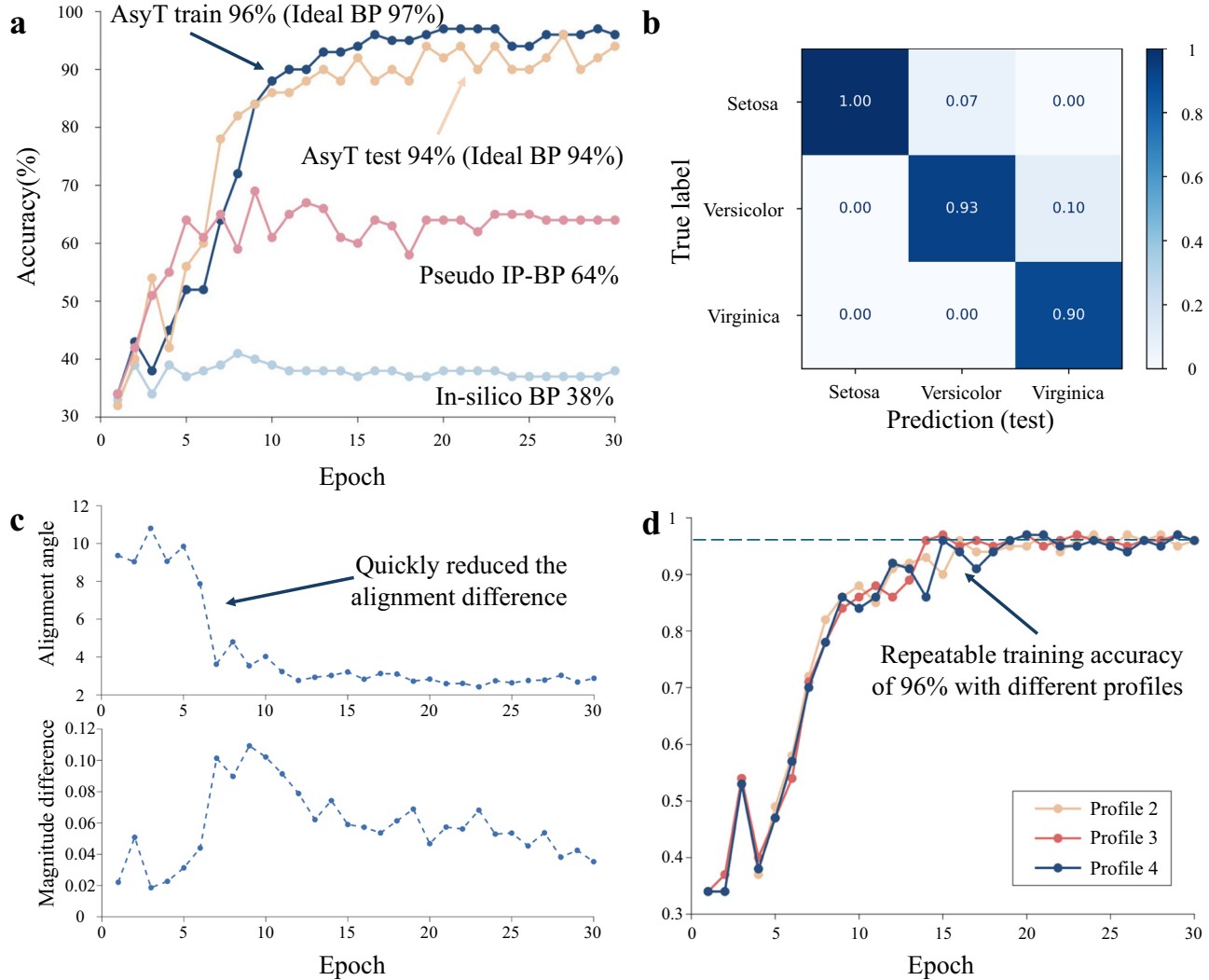

**Fig. 3 | Performance and analysis of AsyT with Iris dataset. a** The training results for AsyT (dark blue line: train; orange line: test), pseudo-IP-BP (red line) and in-silico BP (faded blue line). AsyT retrieves performance near the maximum ideal BP for both training and testing. On the other hand, the performance of in-silico BP and pseudo-IP-BP shows degradation. **b** The confusion matrix for the testing data of the AsyT method. **c** AsyT quickly reduces the alignment angle difference at the early stage of training and maintains the value for subsequent epochs. **d** Repeated experiments with three additional estimation profiles, showcasing that the AsyT method can generalized to different estimation profiles. All three estimation profiles achieve a training accuracy of 96%.

The level of information mismatch is replicated based on the experiments for reliable simulation results (see "Methods" section). The experimental level mismatch is defined as one standardized unit of distortion $\sigma_{phy}$. For an overestimation of the error, we choose a noise floor of 10 dB signal-to-noise ratio where applicable, a higher noise than the typical operation of analogue systems.

We use three datasets of MNIST, FMNIST, and KMNIST for the simulation analysis. We implement an MLP model with the network structure of [784, 256, 256, 10] for all three cases. With the AsyT method, the test accuracies of the encapsulated deep networks are 95.8% (97.0%), 87.5% (88.6%), and 85.6% (87.6%) for MNIST, FMNIST, and KMNIST, respectively (the value in the bracket indicates the maximum ideal BP performance); the results are shown in Fig. 5a–c. AsyT repeatably achieves performance comparable to the ideal error-free BP model. For comparison, the in-silico BP performances are significantly degraded to 25.7%, 16.6%, and 12.8%, respectively. The performance improvement by employing AsyT is significant.

For pressure testing the AsyT method, the standardized distortion varies between 0 and $2\sigma_{phy}$ to investigate the behaviour under different error levels. AsyT shows a high error tolerance, maintaining the test accuracy for the MNIST dataset over 90% for up to twice the error seen at the experimental level (Fig. 5d). Due to the physical boundaries (representable range of control) of the PNN systems, we can define a physically relevant region for the system (see "Methods"). For all the physically relevant regions, AsyT shows consistent performance with over 90% test accuracy while in-silico BP is heavily degraded. The network structure is varied by changing the number of neurons in each hidden layer for a 2-hidden-layer structure and changing the number of hidden layers, each containing 128 neurons. For all neuron sizes, the performance difference between AsyT with 1 $\sigma_{phy}$ is maintained within a 2% difference from the ideal error-free BP maximum (Fig. 5e). Similarly, the test accuracy difference is maintained within 2.5% for all layer depths (Fig. 5f). For the various datasets and network structures simulated, we see repeatable performance of AsyT to achieve accuracy near the ideal BP level.

Existing IP-BP PNN methods, such as PAT, rely on the accuracy of neuron information acquired at the hidden layers. This means that incorrect information accumulation from non-ideal signal readouts (due to quantization, unexpected system perturbation, and low signal

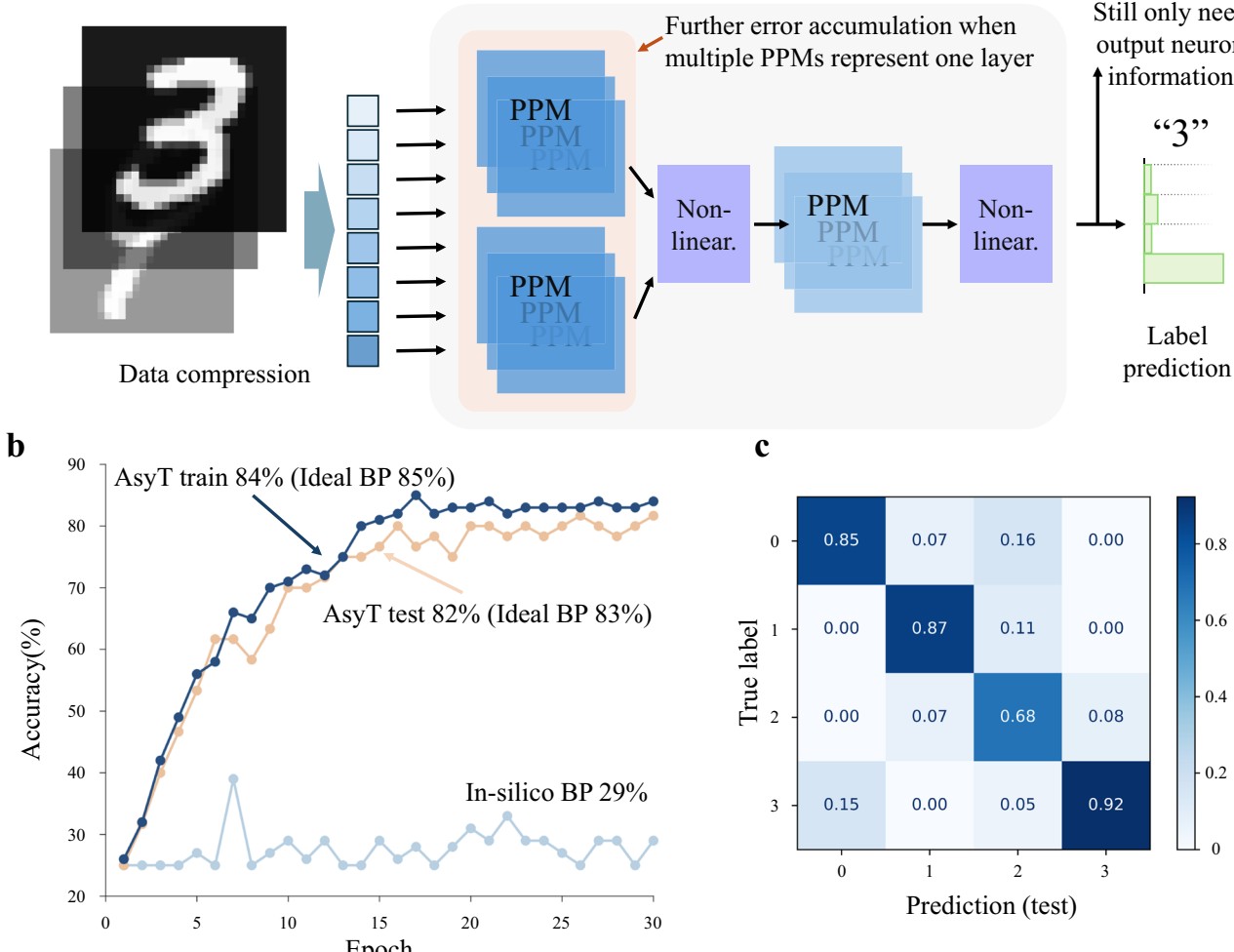

**Fig. 4 | AsyT for scaled-up DPNNs. a** The schematic of scaled-up DPNN systems, where either spatial or temporal copies of the PPM can be used to construct a hidden layer with dimensionality higher than the hardware dimensionality. There is further error accumulation from each representation of the PPM. For training, AsyT only requires the output neuron information and treats the multiple PPMs collectively as if there were one larger-sized PPM. **b** AsyT is not affected by the error accumulation and achieves near-ideal BP performance, while in-silico BP is degraded to random guessing. **c** The confusion matrix for testing data of AsyT for the hand-written digits.

fidelity) can lead to failure to train. In comparison, AsyT utilizes readout at fewer places, suffering less from the noisy system. Furthermore, the digital update is regulated by the parallel model's ideal BP update and is independent of the local PNN (Eq. (3)). The update at each individual timestep is thus protected from a sudden change in the system, offering high error tolerance for non-ideal systems. We investigated two extreme cases of a sudden unexpected system-level hard perturbation (Fig. 5g) and constantly fluctuating signal in a low fidelity system (Fig. 5h). When a sudden hard perturbation occurs during the training process, the prior knowledge of the differentiable model from PAT no longer applies to the new system state, and the mismatch can lead to failure of training. On the other hand, the digital parallel update of AsyT is unaffected by the perturbation, reducing the impact of this perturbation on system training. For the noisy system, the intermediate layer readouts don't represent the actual transformation well. The accumulation of errors at each epoch can lead to a diverging loss, resulting in failure of training. In comparison, the update at each time step of AsyT is protected by the overall estimator (where half of the estimator is unaffected by noise), thus suffering less from the noisy system.

## Discussion

### Efficiency of AsyT-compatible encapsulated structures

While PNNs can offer unique NN acceleration strategies that are not possible on traditional digital NN devices, they also face realistic challenges since the codesign (construction) process can be limited by demanding training operations. Here, we discuss the efficiency that can be enabled by constructing an encapsulated DPNN with AsyT in comparison to the truncated PNN of existing IP-BP methods.

The encapsulation of DPNN bypasses most of the AD interfaces, significantly reducing the signal conversion and data shuttling time for the sample propagation (during both training and inference). For each propagation of information through the PNN structure, we can define the extraction time, $T_{\text{extract}}$, as the minimum time required to obtain an output for further operation (obtaining a prediction or computing update). A simple formulation of the extraction time is the combination of the propagation time $T_{\text{prop}}$ with the readout/conversion time $T_{\text{interface}}$ at the AD interface. The propagation time for PNN is much shorter than the interface and readout time $T_{\text{prop}} \ll T_{\text{interface}}$ (see Supplementary Note 10). Consequently, we could approximate the extraction time scaling $O(P)$ for encapsulated systems and $O(2M - P)$

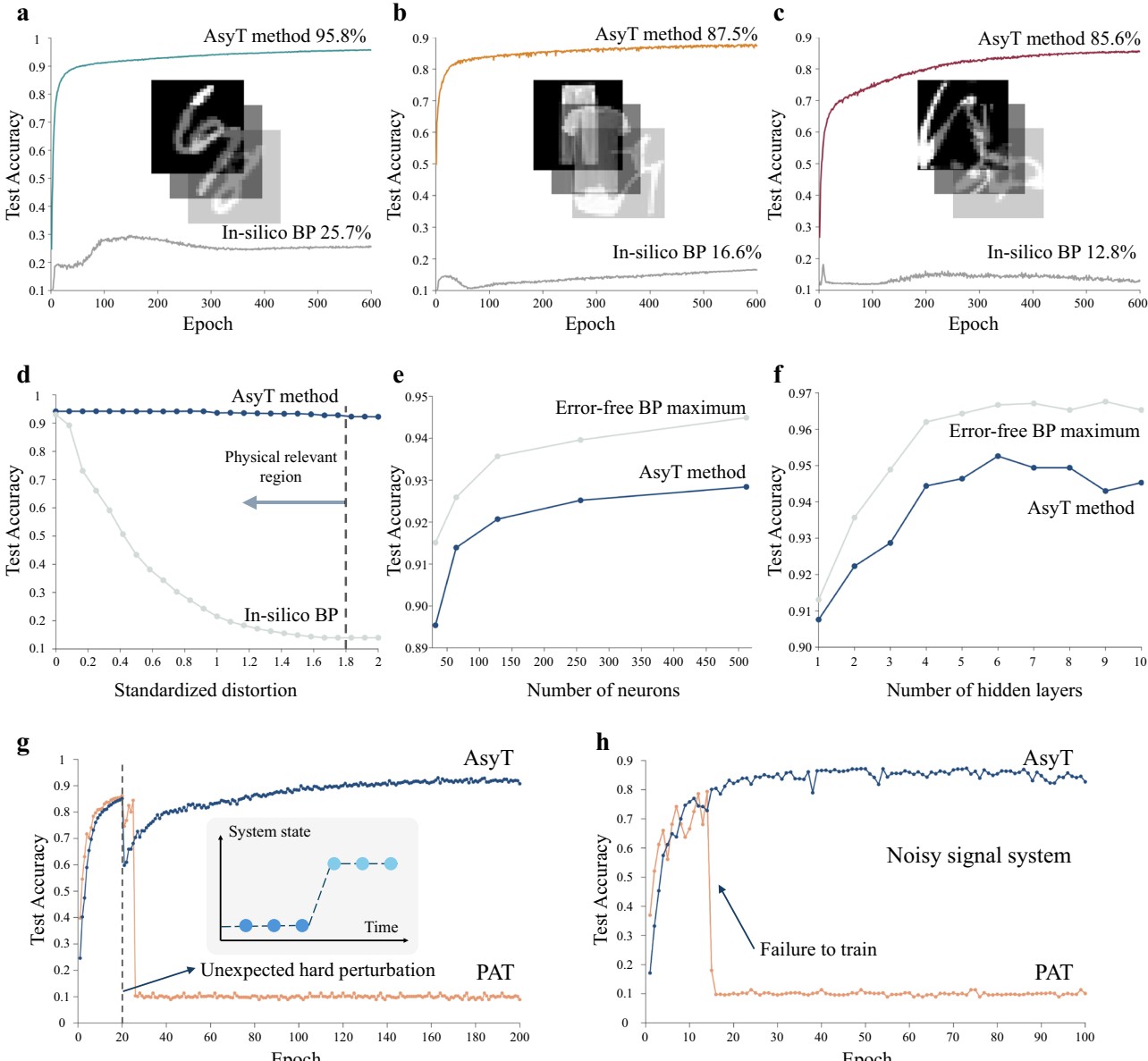

**Fig. 5 | The repeatability of AsyT's performance. a–c** Training three datasets of MNIST, FMNIST, and KMNIST with the AsyT method for PNNs. The test accuracy is significantly improved from in-silico BP (grey line) in all three cases. **d** AsyT (blue line) is pressure tested with varying levels of information mismatch, where the standardized distortion is defined based on the experimental level error. **e** AsyT (blue line) shows good performance for different neuron numbers in a hidden layer, maintaining the difference to the ideal BP maximum (grey line) below 2%.

**f** AsyT's performance (blue line) to maintain the test accuracy difference compared to error-free BP maximum (grey line) below 2.5% for different network sizes. **g** AsyT's (blue line) forward pass in the digital parallel model protects the training from unexpected strong perturbation. PAT (orange line) experiences degradation due to the mismatch in the description of the differentiable model. **h** For very noisy physical systems, AsyT's (blue line) parallel model's digital update is unaffected (orange line: PAT). Thus, granting resistance to noise in the system.

for truncated systems (each AD interface corresponds to a pair of readout-and-write operations, thus the factor of 2). For DPNNs, $P < (2M - P)$ this always holds true. The extraction time is directly related to how fast the prediction is made, and thus, the operation speed of the PNN. $T_{extract}$ for encapsulated PNN is dependent on the specific task instead of the model complexity. Taking the MNIST 10-class task as an example, $P$ stays ten regardless of the model's complexity (Fig. 6a). Consequently, the encapsulated PNN offers better compatibility overall with increasing model complexity.

Furthermore, the sequential layer-by-layer operation of a truncated DPNN imposes an increased bottleneck timestep for computation that is not addressable through higher parallelism. For the $N$ hidden layers, a total of $N + 1$ timestep is required for each single

forward propagation, while there is always only 1 timestep for encapsulated DPNN (Fig. 6b). The constant timestep offers good compatibility with schemes of higher parallelism, such as wavelength division multiplexing.

The extraction time can also be related to the energy consumption of the photonic device during the inference. For a photonic system with power consumption $P_{photonic}$, the minimum operation time is the extraction time for obtaining a prediction. Thus, the minimum energy required to operate the photonic system would be $E_{photonic} = P_{photonic} \cdot T_{extract}$. For systems where the limiting power consumption is photonic hardware, the longer the $T_{extract}$, the larger the energy inefficiencies are created due to the prolonged operation.

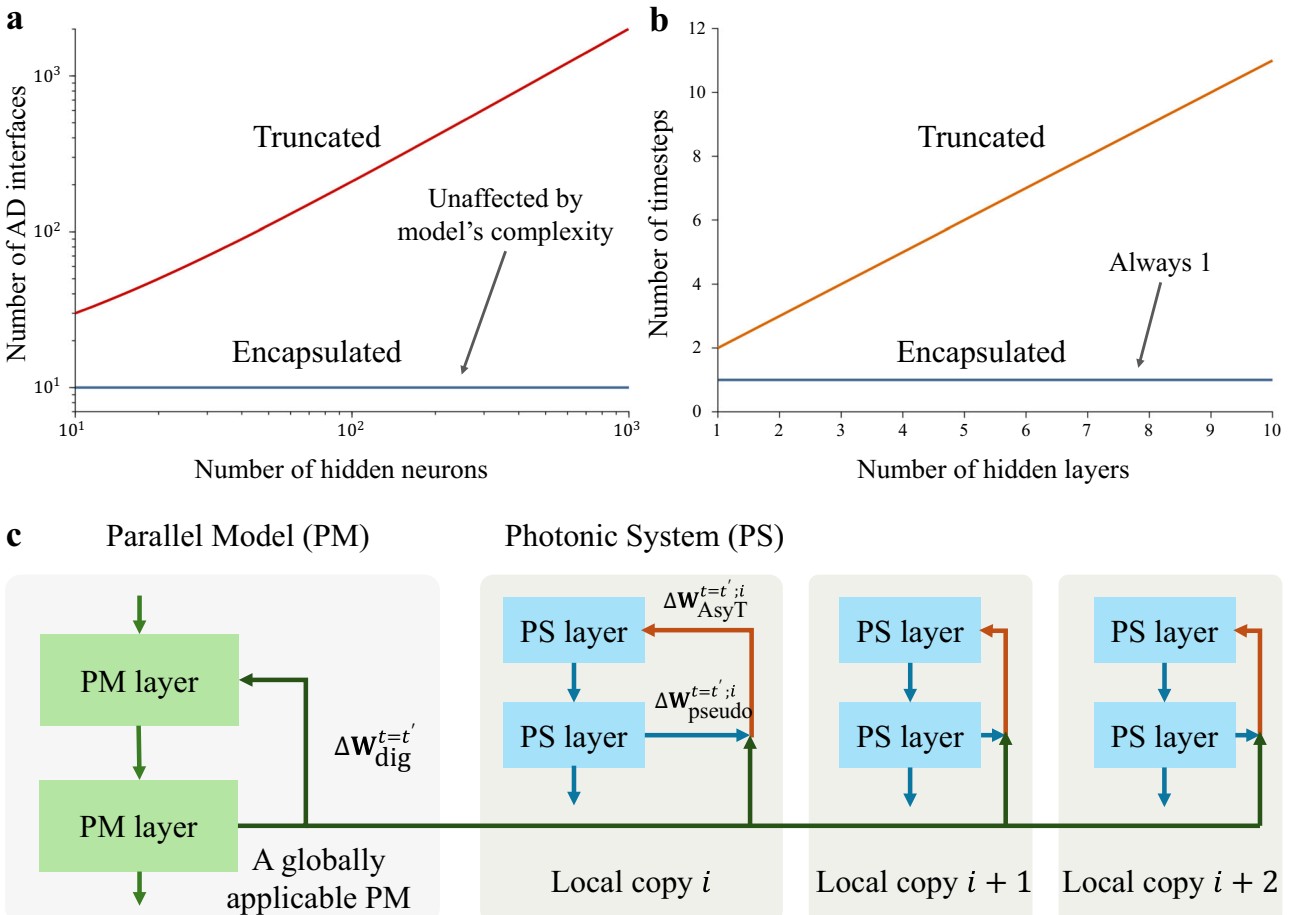

**Fig. 6 | AsyT's efficient training for reproducible DPNNs. a** A truncated PNN's access points scale as $O(2M − P)$ (red line), with $T_{extract}$ determined by the specific model's complexity. In comparison, AsyT's encapsulated PNN structure's access point is determined by the task and not the model's complexity ($P < (2M − P)$) (blue line). **b** Due to the sequential nature of propagation through layers, a truncated PNN's access timestep (orange line) is associated with network depth, while encapsulated PNN (blue line) is not affected (this is under the assumption that $T_{prop} \ll T_{interface}$, see Supplementary Note 10). **c** Through re-applying a general parallel model to different copies of the PNN device, the computational overhead is distributed and reduced for each device copy. This allows the computational overhead to be comparable to standard BP without sacrificing the ability to construct the encapsulation.

## Distributed overhead of AsyT with repeatable application to trainable PNN structures

While PNN training methods utilizing a parallel/twin digital model can provide a more accurate estimate of the physical gradients, the typical concern is the computational overhead that might come with the application of the parallel digital model. Here we discuss how computational overhead (and the associated additional energy consumption) in AsyT can be avoided to achieve a computational overhead similar to standard BP.

In AsyT, the digital update of the parallel model can be computed independent of the local PNN's error, allowing for the same computational timestep as BP without sequential delay. The key motivation for reproducible PNN platforms such as PICs is often associated with reproducible acceleration despite the device-to-device variations. The same PNN structure might be fabricated for many copies to achieve local acceleration of a specific task. For a total of $C$ copies of PNN devices, the overall description of each PNN system can be formulated as $f_{p,1;j}(f_{p,2;j}(\dots))$ $j = 1, 2, \dots, C$, denoting the specific system under discussion. In the case of chip fabrication variation, the overall descriptions of these systems form a statistically relevant set. Thus, we note the equivalence of different estimation profiles and physical device copies. The generality of successful training with different profiles (Fig. 3d) can also be interpreted as different copies of the PNN systems trained equally with a single digital model.

From Eq. 3, the overall AsyT estimator is obtained through two parts: the digital parallel update ($\Delta \mathbf{W}_{dig}$) and the pseudo update ($\Delta \mathbf{W}_{pseudo}$). For a given task and initialization condition, the update of the digital model is independent of the local PNN device and can be described as $\Delta \mathbf{W}_{dig}^{t=t'}(\mathbf{W}_o^{t=t_0}; \mathbf{X}, \mathbf{Y}, \alpha^{t=t'})$ a time instance $t = t'$. On the other hand, $\Delta \mathbf{W}_{pseudo}$ is local to the specific PNN device copy (subject to variation) $\Delta \mathbf{W}_{pseudo}^{t=t'}(f_{p;i}, \mathbf{W}_0^{t=t_0}; \mathbf{X}, \mathbf{Y}, \alpha^{t=t'})$. The generality of a single digital model to different variations of the PNN systems means that the collection of the digital updates can be stored and reapplied to different PNN devices without the need to repeat the process of computing the digital updates (thus no need to repeat the parallel model's backpropagation). For multiple PNN systems, the parallel model only needs to be computed once. Effectively, the computational overhead required to utilize the digital parallel model is distributed across the PNN copies (Fig. 6c). In comparison, the existing method requires specific digital model construction for each PNN copy, meaning a single digital model can't be applied to multiple local PNN copies (see Supplementary Note 16). The overhead of operation in AsyT is

significantly reduced for each individual device. When $C \gg 1$, the overall computational overhead associated with each individual PNN device is reduced towards the overhead of a standard BP process. This allows the benefits of encapsulation to be enjoyed while little overhead is added on top of the standard BP process for the training of many devices.

In summary, this paper presents AsyT as a training method for reproducible DPNNs. Unlike the current IP-BP methods, which construct a truncated system due to the need for intermediate neuron information, AsyT has a reduced requirement of only the output layer information. AsyT's encapsulation allows fast in-propagation transformation for the entire deep network structure instead of within a single layer to fully exploit the potential of photonic platforms. Bypassing the AD interfaces in encapsulated DPNN, sample information propagation can be faster and more efficient. The simple construction of the parallel model also grants AsyT lower computation complexity in comparison to existing methods. We experimentally demonstrated the viability of AsyT for PIC-based encapsulated PNNs. The generality of the AsyT for different network structures and control-transformation deviation is demonstrated with various datasets, including Iris, MNIST, FMNIST, and KMNIST, through experiments and simulation. The error tolerance of the AsyT method is showcased by changing the error level and applying the method to extreme cases. We have discussed how AsyT's computational overhead can be distributed across physical copies of the PNN device for efficient operation. AsyT's generality and efficiency for training and codesigning aim to promote reproducible PNN acceleration to a wider range of users.

## Methods

### The training mechanism of the asymmetrical training estimator

AsyT is designed to be a BP-based method which utilizes both the information in the analogue and digital domain to achieve training. The expression for performing a standard backpropagation is given by the three equations in Eqs. (5–7).

$$\frac{\partial L}{\partial \mathbf{z}^{[l-1]}} = \left[\mathbf{W}^{[l]}\right]^T \cdot \frac{\partial L}{\partial \mathbf{z}^{[l]}} \otimes g^{[l-1]'}\left(\mathbf{z}^{[l-1]}\right) \quad (5)$$

$$\frac{\partial L}{\partial \mathbf{W}^{[l]}} = \frac{\partial L}{\partial \mathbf{z}^{[l]}} \cdot \left[\mathbf{a}^{[l-1]}\right]^T \quad (6)$$

$$\frac{\partial L}{\partial \mathbf{b}^{[l]}} = \frac{\partial L}{\partial \mathbf{z}^{[l]}} \quad (7)$$

The three equations utilize chain rules to relate the error at the output layer to the parameters in the earlier layers. The transformations for the digital model and the physical model can be defined as: $f_m(\mathbf{a}; \mathbf{W}, \mathbf{b}, g) \neq f_p(\mathbf{a}; \mathbf{W}, \mathbf{b}, g)$, where the insertion of the same control parameters will result in different transformations due to distortion of the transformation behaviour understanding for the physical module. The collection of parameters and task input ($\mathbf{W}$, $\mathbf{b}$, $\mathbf{a}$, $g$) define the overall transformation of the model. For each time instance, defined by the specific set of parameters, the deviation between the physical and digital transformations can be described as a projection term $\mathbf{D}$ $f_p^t(\cdot) = \mathbf{D}^t f_m^t(\cdot)$. We can invoke the general expression for the alignment angle between two transformations to be Eq. (8).

$$\cos(\beta) = \frac{\mathbf{B}^T \mathbf{A}}{||\mathbf{B}^T|| \cdot ||\mathbf{A}||} \quad (8)$$

For a total control parameter of $X$ an arbitrary system, we can view the loss-function space of the system as the parameter pair ($\mathbf{W}_{\text{set}}$, $f_{\text{set}}(\cdot)$) for a total $2X$ dimension space. The mismatch between the parameters that we set, and the resulting transformation determines the positions on different loss-function surfaces for different mismatch levels. The ideal mathematical system of ($\mathbf{W}_m, f_m(\cdot)$) is a special case where our knowledge of the expected transformation matches perfectly with the actual resulting transformation. Consequently, the physical domain of the PNN ($\mathbf{W}_p, f_p(\cdot)$) and the digital domain ($\mathbf{W}_m, f_m(\cdot)$) are interchangeably describable. This means that by setting the control parameters asymmetrically to the two systems, the two systems can reach a similar representation (might not be identical) in the loss-function space even though they are on different surfaces.

Two aspects must be satisfied for the training behaviour of AsyT to converge towards an ideal BP as if there were no errors. The first condition is that the AsyT estimator update must conform to the alignment criterion within $90°$ the ideal BP update $\delta\mathbf{W}_{\text{dig}}$[26]. The second is that the AsyT estimator must contain sufficient physical information for the update to be relevant to the physical gradient update of the system. Consequently, we logically see the necessity of utilizing two terms, described by the expression of the AsyT estimator in Eq. (3), with the overall update as a mixture of the two models. The parallel term makes a connection to the digital model, and the pseudo term makes a connection to the physical model as a proxy for the physical gradient.

### The robustness of AsyT and how the physical boundaries of PNNs protect AsyT

To ensure the robustness of the AsyT method for successful training. The first condition can be satisfied through using a mixing ratio $M_{w1}$ of 0.5, such that $\Delta\mathbf{W}_{\text{AsyT}}$ always aligns with $\delta\mathbf{W}_{\text{dig}}$ (for the two updates with similar magnitudes). The second criterion is essentially a question of how each individual digital transformation $f_{m,i}(\cdot)$ aligns with each individual physical transformation $f_{p,i}(\cdot)$. This thus traces back to the initial relation of the transformations at $t = 0$ ($f_m^{t=0}(\cdot)$ and $f_p^{t=0}(\cdot)$). We can describe this term for each individual timestep as the projection of $\mathbf{D}^t$. For $\mathbf{D}^{t=0}$, the fulfilment of the alignment condition becomes a statistical problem of how much deviation can naturally exist between the two transformations when inserted with the same set of controls $\mathbf{W}^{t=0}$ (we are initializing the control for the two models with the same set of parameters).

For two unbounded systems (where the transformations can take arbitrary values), we potentially need to be careful about the alignment. However, physical systems are always bounded systems with their relevant physical boundaries. For example, the signal power encoding for a neuron in the PNN system will not have an activation that is higher than the input signal plus the amplification. Consequently, we see that the physical boundaries of the PNNs enforce a limit for possible mismatch between the mathematical and physical transformations. For periodically bounded systems such as MZIs, we return to a state with effectively lower mismatch when going beyond the turning point of π difference. On the other hand, for range-bounded systems such as MRR or phase change materials, the action of further increasing/decreasing the tuning at the boundaries always returns the system to the boundary state.

Consequently, the robustness of AsyT is statistically protected by the physical boundaries of the systems. For example, for a high level of distortion that accounts for 10% of the entire tunable range, the probability of breaking the alignment is $1.5 \times 10^{-21}$%. For an extreme level of distortion which accounts for a quarter of the entire tunable range, the probability that alignment is broken is $6.3 \times 10^{-3}$%. For realistic PPM platforms with modern fabrication technologies, the deviation we described here exceeds the standard three-deviation variation for fabrication. Consequently, AsyT offers robust training for any statistical relevance in the estimation profile used, a condition that is achievable for fabrication on existing scalable platforms.

## Construction of the digital parallel model

As the robustness of the AsyT method is enforced by the physical boundaries of the PNN systems, it is important that the parallel model is constructed with the physical system in mind. The digital parallel model should have roughly the same encodable physical boundaries as the physical device. When dealing with the analogue encoding for the isomorphic PNN devices, the value encoded will always be in some normalized form. Therefore, the same logic applies to the physical components responsible for the tuning of individual neuron connection strength. The tuning of those connections must also be bound by the physical boundary values. We note that the complexity of the DPNN is not affected by the exact value encoding of the system: for a given finite resolution of the analogue system control, any scaling of the value encoded is equivalent when the activation function is described by the physical tunable range instead of the value itself (as is the case for physical non-linearities).

The first thing to consider is the topological connectivity of the PNN system. Take a fully connected layer $A \times A$ as an example, the representation of the physical neuron can only contribute to a maximum net output $A$. Therefore, we limit the maximum neuron output representation of each layer in the parallel digital model based on the connectivity. Rather than considering the exact type of physical boundary in the system, we treat all physical systems as range-bounded ones. This allows us to perform a simple and effective implementation of the digital parallel model by clipping the physical system to the maximum allowed net output $A$. The added physical boundary to the system helps preserve the statistical relevance between the physical and digital models.

For the case of utilizing multiple PPMs to represent the connection for a single layer, we consider the overall representation of the combined PPMs. By considering the overall topological connectivity, the physical boundaries constructed remain relevant. On the other hand, if we think about the encoding for each individual PPM, the existing systematic deviation can easily lead to skewed boundaries for each channel/neuron output. However, this systematic deviation might not be the same for the subsequent layers. As a result, it is better to consider that the physical model channel/neurons have the same boundaries and use the algorithm to correct the mismatch as a whole.

## Experimental demonstration of AsyT with PNNs

For the implementation of the fully encapsulated training of PNNs with AsyT, we utilize two copies of the $4 \times 4$ PIC PPMs. The task demonstrated is the Iris dataset for classifying the two species of Setosa and Versicolor with a network structure of [2, 2, 2], where the output classes are one-hot encoded with the optical signal intensity. The input compressed from the four features to two principle components with principal component analysis[54]. To allow the full encapsulation of the deep network structure, the two chips are directly connected through a photonic non-linearity added by a high-current EDFA. Due to PIC's sensitivity to the polarization of the signal, we use a polarization controller for each instance that the signal is passed onto the chip. We use two sets of EDFAs for the experiment: The first set acts as the photonic non-linearity to provide a tanh-like shape of the transmission. Due to the insertion losses of the two serially connected PICs, the second set of EDFAs is set with a lower current density for amplification behaviour in the linear region without adding further non-linear transformations to the signal. The wavelength used for the experiment is near 1550 nm for the SiN PIC. The control of the physical parameters is added to the PIC via DAC with a driver board. The control is tuned with 16 bits in a range of 0 V and 32 V. We limit the input and transmission modulation to 100 quantized levels. A detailed list of the equipment used for the demonstration can be found in Supplementary Note 1.

For the demonstration of AsyT on a DPNN with only the output layer neuron information, we utilize a similar setup of the PIC chip for the input and control modulation. We digitally add a photonically compatible non-linearity with a sigmoid-like shape, which is often seen from the response of structures such as MRRs[19,35,55–57]. While the activation is added digitally, the training algorithm only gets access to the output layer information which helps validate AsyT's ability to train with only the output neurons. For each time the signal is put back to the analogue domain, we deliberately add further error (in terms of erroneous control) to represent the potential signal fidelity degradation in the scenario of a fully encapsulated network. The Iris data utilizes an overall deep network structure of [4, 4, 3]. The hand-written digit classification uses the network structure of [8, 4, 4], where the connection $8 \times 4$ has a dimensionality higher than the hardware dimensionality. The $8 \times 4$ connection is decomposed into two copies of $4 \times 4$ connectivity. Similar to the other experimental demonstrations, the training system only gets the output layer information. The dataset used for the Iris flower classification is based on the original dataset[52], where samples of digits 0, 1, 2, and 3 are selected through a random number generator in Matlab. The MNIST dataset is modified to be suitable for demonstration on the PIC PNN: the original $28 \times 28$ pixel image which flattens to 784 features contains largely zeros and is compressed to 8 features through principal component analysis. In all the experiments, the classes of samples are one-hot encoded by the output channel. More details on the setup in the experiments can be found in Supplementary Note 1–3.

## Simulation of PNN training with AsyT method

For determining the experimental level error to be applied to the simulations, we characterize the actual transmission behaviours of the MZI cells and compare how much they deviate from the estimation profile used. The grey lines in the background of Fig. 2f show the actual transmission behaviours of the MZI cells. We define the worst-case deviation between the leftmost and rightmost transmissions of the MZIs as one standardized distortion of 1 $\sigma_{photon}$. This is intended to serve as an overestimation of the realistic deviation that we see in experiments.

For adding the distortion to the simulation, apart from the systematic distortion in our knowledge of the transformation, we also add other physical noise and fluctuations to the model. The overall expression for the transformation of the physical module is given as Eq. (9).

$$f_p^t(\mathbf{W}^t) = \left(\mathbf{I} + \mathbf{P}_{sys}^t\right) \otimes \left(f_m^t(\mathbf{W}^t) + \mathbf{N}_{init}\right) + \mathbf{N}_{rand}^t \tag{9}$$

The systematic distortion in the transformation at each time step is defined by $\mathbf{P}_{sys}^t$, where the $\otimes$ denotes the element-wise Hadamard product for linear overestimation of the deviation across the tunable range (through element-wise multiplication with the addition of a unity matrix $\mathbf{I}$). The transformation is subject to some fixed initial distortion $\mathbf{N}_{init}$. At each instance, the information is passed through the physical structure, and a randomized error $\mathbf{N}_{rand}$ is added to represent the signal degradation.

For the three datasets of MNIST, FMNIST, and KMNIST, we choose a one-hot encoding for the labels with the network structure of [784, 256, 256, 10]. Each sample image has an original $28 \times 28$ pixel resolution, which is flattened to 784 input features. The subsequent layers are fully connected layers. The hidden layers are connected by ReLU activation functions emulating on-chip MRR nonlinearities[19], while the output activation is a SoftMax layer. The learning rate is varied depending on the task, but within a general range of $3 \times 10^{-5}$ to $1 \times 10^{-4}$. We use a simple gradient descent optimizer for our simulations, alternative optimizer can also be used in conjunction with the AsyT estimator. We use a batch size of 600 for training. At each epoch, the data is shuffled to avoid memorization of the dataset. All the simulations are written with Python predominantly using Numpy[58], Tensorflow[59], Sklearn[60], and Pytorch[61] packages.

**Reporting summary**

Further information on research design is available in the Nature Portfolio Reporting Summary linked to this article.

## Data availability

The data supporting the figures in this study have been deposited in the University of Cambridge Repository at: https://doi.org/10.17863/CAM.115822.

## Code availability

The demonstration code for AsyT in this study has been deposited in the Zenodo database at: https://doi.org/10.5281/zenodo.14860376.

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

## Acknowledgements

This work was supported by the European Union's Horizon 2020 research and innovation programme, project INSPIRE (Q.C.), and UK EPSRC, project QUDOS (EP/T028475/1) (Q.C.).

## Author contributions

Y.W. conceptualized the idea, demonstrated the experiments, and performed the simulation. C.Y. and M.C. helped with the experiments and simulations. C.Y. designed the photonic chip layout. M.C., J.M., and T.Y. helped with the packaging of the photonic chip. Q.C. and R.P. supervised the project. Y.W. and Q.C. prepared the manuscript. C.Y. and M.C. assisted with the preparation of the manuscript.

## Competing interests

The authors declare no competing interest.
