## [Transparent Peer Review file · Nature Communications]

Asymmetrical estimator for training encapsulated deep photonic neural networks

Corresponding Author: Dr Qixiang Cheng

Version 0:

Reviewer comments:

Reviewer #1

(Remarks to the Author)

The paper discusses a training method for in-hardware training of physical neural networks. The proposed approach uses a combination of two gradient estimators: one minimizing with respect to the task loss and another minimizing with respect to the loss on the output of the digital twin, which should match the output of the physical system. The proposed method is then used for experimental training of small-scale photonic hardware. The method is clearly explained. I appreciate that the authors experimentally tested their method and have also discussed the scalability and robustness of the method.

I have a number of questions/comments:

1. I would like to understand the difference between the proposed method and the physics-aware training of Wright et al. <https://www.nature.com/articles/s41586-021-04223-6>

In their work, the output of the physical hardware is used for the forward pass, and the digital twin is used for the gradient estimate, which to me looks similar to the method described in this manuscript. In particular, you have one loss which is the task loss, added to the physics-aware method, which merges these two objectives together. I suggest the authors comment on this point and, if different, in what aspects. Also, I suggest benchmarking against that method as well.

2. In Figure 5b, the results for Fashion MNIST are around 88%, while the results for MNIST are 92% (Figure 5a). An 88% accuracy using MLPs is a state-of-the-art result:

<https://github.com/csbannon/mnist-classifiers/blob/main/mnist-fashion-items/mnist-fashion-item-classification-with-a-fully-connected-neural-network.ipynb>

Can the authors explain how they achieved state-of-the-art results for Fashion MNIST but not for MNIST (92%), especially considering that Fashion MNIST is more challenging?

3. One important requirement of the asymmetry method proposed here is that an estimate of the function of some variables has to be available. The authors also study, I presume in simulation, the scaling of the hardware. Can they comment on how sensitive the method is to these estimates for a scaled system? For example, in a scaled version, these estimates might be less accurate, and other non-idealities introduced. What constitutes a good estimate?

4. There are some grammar issues and typos, for example, at line 86.

Reviewer #2

(Remarks to the Author)

Wang et al. report a new method for training physical neural networks (PNNs) which they call "asymmetrical training" (AT). The manuscript is well-written and incorporates theory, simulation, and experiments on a proof-of-principle photonic integrated circuit. While the idea is interesting, I found some points unclear that need to be addressed before I could make a recommendation on publication.

I have a few major concerns regarding the central claim of the paper:

1. I am skeptical of the claim in the abstract and repeated multiple times in the paper that AT can achieve calibration-free training of physical neural networks. The authors claim this is the advantage over prior work, but I am not convinced this claim holds up.

The authors claim they only need information on the input, the output, and the neuron connectivity. However, it is clear there needs to be some calibration, at the very least to understand the response function of the physical hardware. In the experiments presented in the paper, the authors know a priori the transfer function of the MZIs used on the chip. It makes sense AT will work for matrix-vector multiplication accelerators like the one used in this work, as we know how MZIs behave and how they can implement computation.

However, the term “physical neural network” is more broadly encompassing to describe systems where the response function is not known a priori—for example, the SHG system or mechanical systems reported in Wright & Onodera, *Nature* 601 (2022). In that paper, a digital twin of the hardware is trained in silico to obtain gradients. For these systems, I think AT would similarly require some digital twin of the hardware, thereby casting doubt on the claim that only input/output and neuron connectivity information is needed.

In general, the manuscript is confusing on this point. Sometimes the authors claim no information is needed on the physical transformation (for instance, see line 80, but the claim is repeated multiple times) and call the method “calibration-free” (line 60), while other times it is indicated some information is needed (for example, line 151: “As part of the procedure for conducting the AT method, we need to acquire an estimation profile mapping for the physical control parameters”). Please clarify this in the manuscript and revise the terminology.

2. A related point is how similar the initialization needs to be between the digital model and the physical hardware when starting off training. Can the two be truly randomly initialized? Or does the digital model need to be initialized in a setting that is “close” to the current hardware settings? I imagine this could be challenging without some calibration. More concretely: for a given hardware initialization, if the digital model is initialized randomly many times to different settings, will AT always successfully train?

3. There seems to be some similarities to physics-aware training (PAT) introduced in Wright & Onodera et al., *Nature* 2022. Hardware updates are a mix of two update terms: one is inputting the physical error vector, measured on the hardware, into the backwards pass of the digital model; and the second is auto-differentiating the digital model itself. It seems to me that the first term is similar to the update procedure in physics-aware training (PAT) (for instance, see Fig. 3a in the PAT paper). The novelty would then appear to be introducing the second term, which is auto-differentiating the digital model, but it’s not clear to me how this provides an advantage. It would be helpful to clarify in the manuscript the difference between these two approaches and the relative advantages of AT.

4. How does the efficiency of AT scale with the number of model parameters? Does it scale as well as backpropagation on digital hardware? The tradeoff with many of these algorithms is the reduced efficiency, and so it would be useful to understand the scaling for AT.

5. I would highly recommend making the code for the algorithm public so that it can be implemented on other platforms by the community.

Below are a list of more minor technical points that should be addressed:

6. Figure 2c: please include a schematic of the chip and not just chip images, as it will be difficult for many readers to understand the hardware architecture.

7. Figure 2e/f: what do the faded lines represent? Are they from other heaters? Please indicate this in the figure legend.

8. Figure 2: the caption refers to the PIC as a deep PNN. It looks like it implements a single layer. Please clarify.

9. Figure 3c: the confusion matrices do not add up to 100%. Please double check the data.

10. Figure 3/4: are the accuracies being reported for the training or test set? For training experiments, it is also important to know the performance on unseen data.

11. Figure 3d: why is there a training curve shown for calibrate and transfer? Isn’t the model trained offline and then ported to the hardware?

12. Figure 3a implies the activation function is done in the photonics, but the PIC in Figure 2 implies it is performed digitally. Please clarify.

13. In both Fig. 3 and Fig. 4 the authors compare to the “theoretical maximum accuracy” of the model. How is this obtained?

Reviewer #3

(Remarks to the Author)

This work proposes an asymmetrical training method for training photonic neural networks to address inaccurate network information caused by physical errors. The proposed method utilizes a digital parallel model to assist in gradient calculation during parameter updates, allowing photonic integrated circuits to be treated as a grey-box during these updates. That is, only prior knowledge of topological neuron connectivity, input, and output is required. Experimental results based on a photonic chip with 4x4 matrix-vector multiplication demonstrate improved classification accuracy compared to direct backpropagation and calibrate & transfer methods.

However, after carefully evaluating the manuscript, I found that this work is insufficient for publication in Nature Communications due to the lack of comparisons with state-of-the-art training methods, insufficient motivations, and a small network scale.

The research doesn't provide insightful experimental comparisons with state-of-the-art training methods. With some comparisons, we could identify the major drawbacks of the proposed method. For example, compared to the physical-aware training method [45] and hybrid training [c1], besides the fact that the digital parallel model might be inaccurate for characterizing the physical system, the inaccurate output of the digital parallel model is further used in gradient calculation for parameter updating. This makes the gradient even less accurate than with physical-aware training, which results in lower performance compared to the physical-aware training method. Additionally, comparisons with other training methods, including feedback alignment training [c2], dual adaptive training [47], forward-forward (backpropagation-free) training [49, c3], etc., have not been conducted. The experimental results presented in the manuscript also did not provide side-by-side comparisons with state-of-the-art methods.

The motivation for designing the training method as a grey-box is unclear, as the fundamental building blocks for chip design are typically known. As stated in the manuscript, models of MMI and MZI-based photonic integrated circuits can be explicitly described. Therefore, the scenarios for the proposed grey-box photonic neural network training method should be clearly provided.

The experimental results only cover a very small network scale, with a theoretical maximum MNIST classification accuracy of only 85%. Evaluation on a larger network scale should be performed.

[c1] Spall, J., et al. Hybrid training of optical neural networks. *Optica*. 2022 Jul 20;9(7):803-11.

[c2] Filipovich, Matthew J., et al. "Silicon photonic architecture for training deep neural networks with direct feedback alignment." *Optica* 9.12 (2022): 1323-1332.

[c3] Momeni, Ali, et al. "Backpropagation-free training of deep physical neural networks." *Science* 382.6676 (2023): 1297-1303.

Reviewer #4

(Remarks to the Author)

This paper proposes asymmetric training of photonic neural networks as a way of training neural networks by aligning gradient directions with a parallel model of the device. Overall, the concept of averaging errors from the output of the actual and parallel models for training is an interesting idea for the practical training of PNNs without calibrating each of the layers.

As far as the results are concerned, I think the results for AT are solid and most graphs / eqs are fairly clear but I am still unclear on many things. What do "direct BP" results mean. "We see a significant performance degradation with BP when the physical information is not available." What physical information? What is an "error-free digital reference"? Unclear on what "calibrate and transfer" entails; why are the results so poor for the simulated results? I'd like more details on how the calibration is done so others can reproduce; I don't understand why the calibration resulted in such poor performance.

I have a few higher level issues with some of the attempts to compare backpropagation based approaches in the paper. Mainly, this paper invokes a lot of the same motivations as "Deep physical neural networks trained with backpropagation" (Wright et al, Nature, 2022) but appears to address these problems in a different manner through alignment. I think a closer comparison of these methods is warranted both in simulation and in experiment because of the similarities, i.e. backpropagating through a "simulated model" of the system. What are the advantages / disadvantages and what's different here (the results in the Wright paper seemed quite good...)? Is there a relationship between Wright et al's method to any of the "BP" approaches discussed in this paper?

Additionally, what makes calibration inferior (this is made fairly clear in the intro)? I would like some very specific calculations of the memory and time cost of calibration. Calibration is very useful for debugging and probing physical systems (the paper dismisses this using a somewhat vague analogy to Ising machines / reservoir computing, when MZI networks are really a fairly specific system). Most PNNs are made of MZI meshes which are readily calibratable, and such procedures have been successfully demonstrated at fairly large scales in the commercial sector (e.g. Lightmatter in the past). So why revert to a procedure like this for such systems? What benefit would it provide (and to what extent?) compared to the costs in convenience of directly flashing NN weights and especially less effective debugging of the overall system?

Surely there are other systems to experimentally demonstrate AT than a PNN; and if that was the intent of the paper, there is

currently no compelling argument for the superiority of this method over the accelerated training method proposed by Bandyopadhyay et al. which is actually model free and doesn't require calibration and is much more energy efficient (since you don't have to backprop on a digital machine). Is it convergence time?

Although most of the points above area addressable, acceptance to this journal in my view relies on better addressing existing literature (comparison to Wright et al's approach especially); if the authors can show AT has sufficient advantage over calibration-based approaches and make sufficient novelty claim over what Wright et al did, that would really help their acceptance case in my opinion. Happy to give more feedback after the authors address the above major points.

Version 1:

Reviewer comments:

Reviewer #1

(Remarks to the Author)

Thank you for your detailed explanations. I believe the manuscript has improved significantly. My concerns are now resolved.

(Remarks on code availability)

Reviewer #2

(Remarks to the Author)

Thanks to the authors for their response to my comments. The main issue, as raised by all reviewers in the previous round, was the comparison to prior work. While I find AT to be interesting, I have multiple concerns about the author's new comparisons to prior training experiments.

1. I find the new claims in energy and time advantage for training are not supported. This is included in the abstract: "For any deep network structure, AT offers significantly improved time and energy efficiency compared to existing BP-PNN methods, and scales well for large network sizes." Unfortunately, their calculations are based on flawed assumptions and the algorithmic scaling claim, i.e. does it scale as well as BP as network size increases, is also not proven in the paper. The authors argue that near the end of the training, AT's update will converge to an ideal BP update, but this is only true at the end and not for most of the training runtime.

Moreover, the authors do not accurately consider the power consumption and speed of the digital system in their calculations. In the supplementary, they compute the energy and time of the digital system for training, arguing that an A/D conversion interface is much slower than running a model on the digital system. This is based on two assumptions:

- They estimate an ADC throughput of 60 Mbps. I'm not sure where this comes from, but it is clearly incorrect – commercial ADCs are available at 10s of Gbps and are used in the optical communications industry. For instance, see Murmann's survey at <https://github.com/bmurmann/ADC-survey>. It looks like they assume the bottleneck is SPI, but faster communication protocols such as JESD204B exist for precisely this application. Thus the estimate here is orders of magnitude off.

- They claim that a state of the art GPU (Nvidia A100) has a computation speed of 624 teraFLOPs, which is why the speed of the digital system used for executing the model in their algorithm is negligible. The authors are confusing throughput and latency here – a digital system can have high throughput, which is achieved through pipelining, amortizing memory access, and reducing bit precision, but that does not equate to the latency. For example, Nvidia's own numbers on the MLPerf benchmarks indicate about ms latencies for inference (probably larger for training): <https://developer.nvidia.com/deep-learning-performance-training-inference/ai-inference>.

In Figure 6 of the main text, the authors now introduce a new plot showing significant speed up over other methods that rely on digital systems for the backward pass. It is hard to evaluate this plot, as in addition to the flawed assumptions I mention, there is not enough information on the exact parameter values used or for what model this is being calculated. The GPU is estimated in their calculation to be 500 fJ/FLOP. This is not correct (624 TOPS number quoted by Nvidia is for INT8). The total energy per inference will depend on the model size (i.e. number of FLOPs and bit resolution) but it is not clear in the calculation what is assumed here. It would help if the calculation were explained, step-by-step, in the supplementary.

In summary: 1) it is not likely that A/D is the bottleneck compared to having to run the model on a digital system; 2) the digital system energy and latency is likely greatly underestimated, and I think it is likely the GPU will dominate these quantities, not any A/D interface; 3) the algorithmic scaling is not studied, and the authors claim without any simulation or experiment that it is equivalent to backprop; and 4) not enough information is provided on these calculations to evaluate the claimed advantage over other methods.

2. The authors argue that AT is an improvement on physics aware training (PAT) in both training time and energy

consumption, demonstrated by Wright & Onodera et al., because it skips the A/D conversion between layers and relies only on output information. While this is certainly better, it is not clear if it is a significant advantage because, as I mentioned in point 1, ADCs can convert at tens of Gbps. So it is not clear if the energy consumption/latency due to the A/D interface is significant compared to the energy of the digital processor. If the A/D cost is incremental relative to the total energy consumption/speed of the GPU, then PAT may be better as it uses true BP, and therefore should train faster, while AT may not be as efficient.

3. I still do not agree that AT is “calibration-free.” My concern earlier was that it is demonstrated on a programmable PIC, and of course we know how MZIs work and that they have a cosine transfer function. The authors note that no readout is performed from the PIC – yes, I agree, but some a priori knowledge about the system is leveraged here.

What would we do for a physical system where we have no information about the transfer function at all? How would AT be used for any of the systems in the original PAT paper? Clearly some calibration would be needed.

In any case, the authors have adequately narrowed this claim by indicating AT only applies to “scalable isomorphic physical neural networks.” Instead of inventing new terminology, I think it would be preferable to use the terminology already standard in the community – either photonic neural network or optical neural network, both of which are usually understood to utilize optical matrix-vector multipliers.

4. I think there is a miscommunication regarding my question about model initialization. I understood that when training the parameters should be randomly initialized. My question was whether the digital twin needs to start with parameter settings close to the physical hardware, or can the digital twin be completely “off” from the physical setup and AT will align the two over time. For instance, in an MZI mesh, suppose all of the hardware is set to the “cross” state. Does the digital model also need to be initialized such that all the software MZIs are in the “cross” state, or can they be random?

The authors’ response suggests they should be close to the physical hardware. This supports my point that AT is not calibration-free—if it were, it should not matter and we could start training with zero information about the system. The authors provide some experiments showing the use of different estimation profiles, but they all are qualitatively cosines with some shift.

To be clear, this is all fine and I do not consider this a significant limitation of the method. But the description of the method should be accurate to avoid confusing readers.

5. My question on algorithmic scaling was not adequately addressed. The authors provide a new table in the main text comparing the scaling, but they focus primarily on number of readout time steps and intermediate readout. First, some of the entries for other algorithms are incorrect and thus the comparison is inaccurate. Finite stochastic (FS), as demoed in Bandyopadhyay et al., did not require intermediate readout. The demonstration of in-situ BP in Pai et al., 2023 did require multiple readout time steps, but it is not strictly necessary as the backpropagation can be performed all-optically – see Spall et al, arXiv:2308.05226. DFA should not require calibration, as the backwards pass can use a fixed, random weight matrix for feedback (for instance, Nakajima et al. demonstrated DFA on a reservoir computer).

The authors also claim that AT has identical scaling to backpropagation, but no simulation or experimental evidence is provided to support this. I'm not sure this is true, since alignment methods usually are not as efficient as true BP—the authors acknowledge this in the table, as for direct feedback alignment (DFA) they claim the scaling is worse than BP. They categorize the scaling of prior algorithms as “ $O(T)$ ”, “ $>O(T)$ ”, and “ $\gg O(T)$ ”, where $O(T)$ is equivalent to backprop, but it is not clear how they determine these scalings for each given algorithm without some rigorous comparison. The exact scaling is important, as some of these algorithms, such as PAT and AT, require digital systems and others, such as FS and in-situ BP, do not, and thus the question is whether the added power consumption and slow processing of a digital system is worth the improved training efficiency, as what we care about is the total energy and time spent to fully train a model.

6. In their response, the authors also compare to the finite stochastic (FS) method. There are two issues with the comparison: first, they claim a faster training time because they only require S physical propagations of each training sample, while FS requires $2S$. In principle, FS could do a single-sided estimate (just measuring $f(W+\delta)$), for instance, so this is not a fundamental distinction, and in any case a factor of two is not a big advantage anyway. Second, and more importantly, the authors do not consider that AT requires backprop on a digital system, while FS can be run entirely on the optical system. Thus, even if AT is algorithmically as fast as backprop (which the authors claim, but is not proven), it is slowed down by the need to operate at the digital system’s clock speed.

In fact, the use of a digital system essentially eliminates the advantage of in-domain processing for training, as the bottleneck will always be the digital system for BP. If the authors want to claim a wall-clock training time speedup, a far more rigorous analysis, including the latency of the digital system, is required to compare the two.

They also claim AT is more energy efficient, but again the comparison is not valid because they do not consider the energy of the digital system. This impacts the comparison not only to FS, but all the algorithms in Table 1 that train only on the photonics. A more detailed comparison, which accounts for the tradeoff between improved algorithmic efficiency using a digital system and the energy advantages of training in-situ on the photonics, would be needed to support this claim.

Ultimately, I find the proposed method interesting and it is important to develop new alternatives for backpropagation, so the topic of the paper is quite timely. Unfortunately, there are enough issues with the comparisons to other work and the claims

of advantage in training time/energy that I feel the current version of the manuscript is not suitable for publication in Nature Communications without significant revision.

(Remarks on code availability)

I was able to successfully run the code and reproduce the plot provided in the repository. The code has minimal documentation, however, so I was not able to check the internals of the code to understand how the error of the physical system is simulated.

Reviewer #3

(Remarks to the Author)

After reviewing the author's response letter and revised manuscript, I still feel that I cannot support the publication of this work in Nature Communications.

Although the author has added a comparison with the PAT method, I believe it is not entirely fair, and the provided arguments are not very solid because the PAT method can also function without measuring intermediate states.

In each training session of AT, the same sample is input into both the simulation model and the physical system, and measurements are obtained at the final output to calculate the Loss. Due to errors in the simulation model and the physical system, the Loss values differ. Different Loss values are backpropagated through the simulation model to calculate gradients for the parameters, referred to in the paper as ΔW_{dig} and ΔW_{pseudo} , as seen in Equation (3) of the main text. The gradients used to update the parameters are a weighted sum of the first two gradients, with coefficients of 0.5 used in the paper.

The AT proposed in this paper is actually closer to the work of PAT. The AT in this paper does not involve modeling errors, and the biggest difference from PAT is that PAT updates gradients using pure ΔW_{pseudo} , while AT uses a weighted sum. I believe this method involves relatively minor changes.

Regarding the multiple mentions in the text and rebuttal that "For existing BP-based methods such as PAT, the need to readout the intermediate hidden neuron scales... AT only needs to access neuron information at the output layer...", this statement is incorrect. The reasons are: (1) The frequent conversion between AD and DA is closely related to the structure of photonic neural networks, not network training. (2) PAT can also train the network by measuring only the final output state. The full text of PAT does not state that it needs to measure intermediate states.

Therefore, I have concerns about the methodological innovation and motivation of this paper. Regarding the method itself, my doubts mainly lie in the gradient weighting. The paper mentions that AT training requires two conditions, one of which is: "The first condition is that the AT estimator update must conform to the alignment criterion of within 90° from the ideal BP update." The intuitive explanation is that the directional angle between ΔW_{dig} and ΔW_{pseudo} must be less than 90 degrees, meaning their dot product is positive, and the two gradients must be positively correlated. However, how can this be ensured in environments with large errors? For example, when environmental errors are large, the update directions of the gradient ΔW_{dig} calculated by the simulation model and the gradient ΔW_{pseudo} calculated by the physical model may be completely opposite, resulting in parameters that cannot be updated at all. How can the convergence of the algorithm be guaranteed? Therefore, I also have concerns about the method's ability to correct errors.

(Remarks on code availability)

A simple demo code has been released.

Reviewer #4

(Remarks to the Author)

I believe the authors have addressed my main concerns showing PAT performing below AT with new experiments. I would recommend for acceptance.

(Remarks on code availability)

Version 2:

Reviewer comments:

Reviewer #2

(Remarks to the Author)

Thanks to the authors for considering my comments. The authors' revisions have addressed my remaining concerns, and I'm happy to recommend publication in Nature Communications.

(Remarks on code availability)

Reviewer #3

(Remarks to the Author)

I would like to thank the authors for addressing the questions raised earlier and for clarifying my misunderstanding of PAT regarding the measuring of intermediate state readouts. I also have gained a deeper understanding of this work during the revision. However, I still have a concern about the comparison between the proposed method and other methods as the comparison with the relevant PNN training methods has been removed during the revision. Besides, I am also curious about the effectiveness of the proposed method on spatial DPNNs.

1. The manuscript categorizes DAT [39] as an IB-BP method, stating that it heavily relies on intermediate layer state extraction. However, the original DAT paper explicitly notes that the use of intermediate layer states is optional during training. While providing intermediate states can enhance performance, DAT can still function effectively without them (referred to as DAT w/o IS). Moreover, DAT w/o IS has been demonstrated on both diffractive PNN and MZI-based PNN architectures. I suggest that the authors clarify this point and discuss the differences in methodology and application scope between AT and DAT w/o IS. Besides, I would also suggest using other abbreviations, instead of "AT", for the proposed asymmetrical estimator training method, as "AT" could also stand for adaptive training method proposed in [16].

2. The authors only compared AT with PAT (I assume the 'pseudo IP-BP' in Fig. 3 represents PAT without intermediate states) and in-silico BP. How does the performance of AT compare to DAT w/o IS? DAT models errors using an additional error prediction network. Although this increases training complexity, it also enhances training performance. I guess that the performance of AT may represent a trade-off between PAT and DAT w/o IS.

3. The evaluation of AT for large-scale DPNNs is based solely on simulations of MZI-based DPNNs with a noise level of 10 dB signal-to-noise ratio. While this is valuable, the applicability of AT to other types of DPNNs, such as spatial architectures (e.g., diffractive DPNNs), which are prone to geometric and fabrication errors, remains unaddressed. How effective is AT in handling geometric and fabrication errors present in spatial DPNNs like diffractive DPNNs? Would the authors consider conducting simulation or physical experiments with discussions on a type of spatial DPNN to evaluate the effectiveness of AT?

(Remarks on code availability)

More results could be demonstrated if possible.

Version 3:

Reviewer comments:

Reviewer #3

(Remarks to the Author)

The authors have addressed my concerns during the revision. Therefore, I can now recommend the publication of this work in Nature Communications.

(Remarks on code availability)

General message to editor and all reviewers:

Many thanks to the editor and all reviewers for the valuable comments. Your precious time to provide the feedback is very appreciated. We strive to present our work in a manner that is most beneficial for the community, and we wouldn't be able to do this without you. We have now fully revised the manuscript based on your comments, where changes in the main article are tracked as: **revised content**. For quoting text from the original manuscript to the response letter here, we use red text: **quoted content**. The quoted section which has been revised according to the comment is shown as: **quoted revised**.

The reviewers can find the specific response to their comments on the following pages:

Reviewer #1: Page 1-8

Reviewer #2: Page 9-21

Reviewer #3: Page 22-29

Reviewer #4: Page 30-38

The text color above is for specifying the comments from each reviewer. Places where we have quoted the review's original comments for a specific response, we use the highlighter of **quoted comments**.

Response to Reviewer #1:

Original comment:

Reviewer #1 (Remarks to the Author):

The paper discusses a training method for in-hardware training of physical neural networks. The proposed approach uses a combination of two gradient estimators: one minimizing with respect to the task loss and another minimizing with respect to the loss on the output of the digital twin, which should match the output of the physical system. The proposed method is then used for experimental training of small-scale photonic hardware. The method is clearly explained. I appreciate that the authors experimentally tested their method and have also discussed the scalability and robustness of the method.

I have a number of questions/comments:

1. I would like to understand the difference between the proposed method and the physics-aware training of Wright et al. <https://www.nature.com/articles/s41586-021-04223-6>

In their work, the output of the physical hardware is used for the forward pass, and the digital twin is used for the gradient estimate, which to me looks similar to the method described in this manuscript. In particular, you have one loss which is the task loss, added to the physics-aware method, which merges these two objectives together. I suggest the authors comment on this point and, if different, in what aspects. Also, I suggest benchmarking against that method as well.

2. In Figure 5b, the results for Fashion MNIST are around 88%, while the results for MNIST are 92% (Figure 5a). An 88% accuracy using MLPs is a state-of-the-art result:

<https://github.com/csbanon/mnist-classifiers/blob/main/mnist-fashion-items/mnist-fashion-item-classification-with-a-fully-connected-neural-network.ipynb>

Can the authors explain how they achieved state-of-the-art results for Fashion MNIST but not for MNIST (92%), especially considering that Fashion MNIST is more challenging?

3. One important requirement of the asymmetry method proposed here is that an estimate of the function of some variables has to be available. The authors also study, I presume in simulation, the scaling of the hardware. Can they comment on how sensitive the method is to these estimates for a scaled system? For example, in a scaled version, these estimates might be less accurate, and other non-idealities introduced. What constitutes a good estimate?

4. There are some grammar issues and typos, for example, at line 86.

Response:

Dear Reviewer #1,

Thank you for your valuable comments, and we really appreciate your interest in our AT method. Hopefully, we can clarify some of the points and address your comments with the following response and the revised manuscript.

“1. I would like to understand the difference between the proposed method and the physics-aware training of Wright et al. <https://www.nature.com/articles/s41586-021-04223-6>

In their work, the output of the physical hardware is used for the forward pass, and the digital twin is used for the gradient estimate, which to me looks similar to the method described in this manuscript. In particular, you have one loss which is the task loss, added to the physics-aware method, which merges these two objectives together. I suggest the authors comment on this point and, if different, in what aspects. Also, I suggest benchmarking against that method as well.”

The key difference between asymmetrical training (AT) and physics-aware training (PAT) originates from the motivation for creating AT. Despite some BP-based PNN training methods such as PAT exist, the key limitation of these methods is the requirement to know the physical neuron information at each hidden layer to construct a differentiable digital twin model for performing backpropagation (BP). However, this necessarily means two things: 1. The propagation of the analogue signal must be truncated at each layer. 2. The need to access the neuron information creates excessive readouts and conversions at the analogue-digital interfaces.

The need to truncate the physical information propagation in methods such as PAT inevitably leads to slower processing, compromising the initiative of PNN implementations for fast in-propagation processing. On the other hand, AT preserves the completeness of the physical information within the analogue domain until the output layer,

allowing for fast in-propagation processing for the entire deep network structure to be encapsulated within the analogue domain.

Furthermore, the speed of methods such as PAT is bottlenecked by the interface speed between the analogue and digital domain. For existing BP-based methods such as PAT, the need to readout the intermediate hidden neuron scales with $O(M)$, where M represents the total number of neurons in the system excluding the output layer. This means that by extracting the physical information, and then convert the information back into the analogue domain requires at least $O(2M - P)$ analogue-digital interfaces. In comparison, AT only needs to access neuron information at the output layer, scaling as $O(P)$ with P output neurons. We know that for any sensible deep network structure, $M \geq 2P$. Therefore, for any deep network structure, we can see a significant improvement in speed for AT-compatible systems compared to PAT-compatible systems.

Also, PAT can't train with solely the output layer neuron information; this can be seen with the newly added experimental demonstration in Fig.3 (a). The pseudo intermediate-access physics-aware backpropagation (IP-BP) performance indicates the training behavior when PAT only has access to the output layer information and not the hidden layer information. The constructed differentiable model is no longer an accurate description of the actual physical transformation, leading to degradation of performance.

There are also practical advantages for using AT in realistic systems compared to PAT. PAT relies on the fidelity of the analogue information acquired at each hidden layer. However, we know that for realistic physical systems, there are external effects which can affect the system state and the signal fidelity. For example, when the physical system state experiences a sudden unexpected strong perturbation, the twin model in PAT is no longer a suitable description for the physical system. This can lead to failure to train with a diverging loss. Similarly, when the system is sensitive to the operational environment (such as temperature fluctuations), the signal fidelity can be affected, and the system state might experience slight fluctuations. This can also lead to failure of training with PAT. On the other hand, the

additional forward digital pass in AT always being trained with an ideal BP. The digital twin is not affected by the variation in the physical domain, protecting the update at each time instance. These scenarios are studied in simulation and shown in Fig.5 (g) and (h) of the revised paper.

As briefly mentioned above, AT allows faster, and more energy efficient training compared to PAT due to the continuous propagation of analogue signal and bypassed analogue-digital interfaces. Even when considering the additional forward pass required in the digital domain, we can see a significant improvement in efficiency for different network sizes. We duplicate the results from Fig.6 here for easier viewing:

We have now fully revised manuscript to better illustrate the difference between AT and PAT, and further highlight the motivation for utilizing AT. Please see the new version of the paper for details. We would like to highlight a few places where we made the modifications:

Introduction section (P1): “Backpropagation¹³ (BP) finds the gradient update to the loss function for the network parameters, offering fast training time step convergence¹⁴. However, existing BP-based PNN training methods such as physics-aware training¹⁵ (PAT), in-situ BP¹⁶, dual adaptive training¹⁷ (DAT), and hybrid training¹⁸ (HT), relies on knowing the net output and activation level information for hidden layer neurons in a deep network structure. This means that the propagation of the analogue signal needs to be truncated and readout at each layer. The truncation of the physical information propagation undermines the advantage of ultra-fast in-propagation processing with PNNs¹⁹ (See Fig.1 (a)). Furthermore, the need to shuttle the information back-and-forth across the analogue-digital (AD) interface creates excessive analogue-to-digital conversions (ADCs) and digital-to-analogue conversions (DACs). ADC and DAC are known to be speed bottlenecks^{20,21}, where the AD interface speed²² is much slower than solely in-analogue¹⁹ or in-digital⁴ computation (see Fig.1 (b)). These factors lead to overheads of increased training time, additional energy consumption, and increased fabrication cost, compromising PNNs’ goals for fast, energy-efficient and cheap processing²³.”

Results section (P3-8): Entire subsection of “Asymmetrical training method”, “AT method for training fully encapsulated PhPNNs”, and “AT for training deep PNN with only output neuron information”

Results section, subsection of “Generalization and repeatability of the AT method” (P10,11): Fig.5 (g) and (h), and last paragraph of the subsection: “Existing IP-BP PNN training methods rely on the accuracy of neuron information

acquired at the hidden layers ... the update at each time step is protected by the overall AT estimator, thus not affected by the noisy system.”

Discussion section(P11-13): Entire section and Fig.6.

“2. In Figure 5b, the results for Fashion MNIST are around 88%, while the results for MNIST are 92% (Figure 5a). An 88% accuracy using MLPs is a state-of-the-art result:

<https://github.com/csbanon/mnist-classifiers/blob/main/mnist-fashion-items/mnist-fashion-item-classification-with-a-fully-connected-neural-network.ipynb>

Can the authors explain how they achieved state-of-the-art results for Fashion MNIST but not for MNIST (92%), especially considering that Fashion MNIST is more challenging?”

There are a few reasons for this. In the first version of the manuscript, we used the MLP structure of [784, 128, 128, 10] for the MNIST dataset, while the FMNIST and KMNIST are trained with a larger structure of [784, 256, 256, 10]. We realized that this can indeed be confusing because Fig.5 (a, b, c) are listed together, so it might confuse readers to think that they are trained with the same network structure. Thank you for pointing this out, we have now updated Fig.5 (a) to the result trained by a network with the same network structure. Please see the revised main article.

The second reason is that for the first version we rounded the accuracy of the task to two significant figures, so to three significant figures, the accuracies are 95.8% (MNIST, updated), 87.5% (FMNIST) and 85.6% (KMNIST). Consequently, the actual test accuracy for FMNIST is slightly lower than the 88% SOTA result you have mentioned. However, it is expected that accuracy of PNN training will be slightly lower than an ideal error-free BP reference. For all the datasets, we see that the performance with experimental level error is very close to the case of an error-free BP reference, 95.8% vs 97.0% (MNIST), 87.5% vs 88.6% (FMNIST), and 85.6% vs 87.6% (KMNIST). We know that the performance for all BP-based PNN methods must be capped by the maximum error-free BP performance, because it is unrealistic that systems with error can achieve better performance than systems without error for an error-sensitive method like BP. The important thing is that these results show AT’s ability to retrieve near maximum performance even with only the output neuron information for PNN systems, acting as the foundation for AT’s improvements in time and energy efficiency for encapsulated PNN training.

We have made the following modifications to address the comment:

Results section, subsection of “Generalization and repeatability of the AT method” (P10): “We use three datasets of MNIST, FMNIST, and KMNIST for the simulation analysis. For all three cases, we implement a MLP model with the network structure of [784, 256, 256, 10]. With the AT method, the achieved accuracies are 95.8% (97.0%), 87.5% (88.6%), and 85.6% (87.6%) for MNIST, FMNIST, and KMNIST, respectively (the value in the bracket indicates the maximum ideal BP performance). For all three datasets, AT achieves comparable performance of the

ideal error-free BP model. For comparison, the in-silico BP performances are significantly degraded to 25.7%, 16.6%, and 12.8%, respectively. The results are shown in Fig.5 (a-c)”

Results section(P10): Fig.5 (a, b, c).

“3. One important requirement of the asymmetry method proposed here is that an estimate of the function of some variables has to be available. The authors also study, I presume in simulation, the scaling of the hardware. Can they comment on how sensitive the method is to these estimates for a scaled system? For example, in a scaled version, these estimates might be less accurate, and other non-idealities introduced. What constitutes a good estimate?”

The main reason for using an estimation profile is to provide some reference to apply the physical parameters to the controls. As you mentioned, there are further non-idealities for larger scaled systems, such as noise and resolution loss due to the attenuation of physical propagation. These non-idealities would exist for all existing physical training methods, which is the reason for requiring amplification and filtering components in large hardware. However, the estimation profile itself isn’t sensitive to the scaling of the system for isomorphic PNNs. As long as the estimation profile has any statistical relevance to the actual tuning of the system, it should be usable for training.

The high tolerance of AT to the estimation profile originates from the physical behavior of the PNN systems. If we are considering two mathematical transformations which can take arbitrary values, then statistically randomizing these two mathematical systems without boundaries can easily result in very different transformations. However, this would not be the case for any PNN systems. Any analogue computation must be defined based on the boundary conditions of the physical platform. Taking integrated photonics as an example, the encoding value from the output can never exceed the input power plus the amplification in the system. This means that the value encoding for any PNN system must be bounded.

We can see that the bounded representation of the physical system actually restricts the deviation of the physical transformation from our estimation based on the tunable range of the physical control. For example, for periodically bounded systems such as MZIs, the further deviation over a π range brings the transformation back towards the boundaries and appears as if it were in a state with “less” deviation. There are also range bounded systems such as micro-ring resonators or phase change materials, where applying increase/decrease operator to the maximum/minimum state will always return to the boundary states.

Consequently, the co-construction of the digital parallel system is important. When we form the digital parallel model, due to the isomorphism of the system, we can define boundaries for the allowed weight matrix and activation level of the digital model, making the system compatible with the physical modules. This is the reason why we can

define the concept of the physical relevant region during the simulation when changing the level of error in the physical system.

Therefore, the scaling itself doesn't break the boundary conditions for each individual control with the estimation profile. But instead, on a system level, we do need to be aware when the physical system size changes and apply the new physical boundaries to the digital parallel model accordingly. When the parallel model's boundary conditions are appropriately defined along the physical module, the training behavior is statistically protected (see Methods section for more details).

We have also simulated the cases with changing network sizes, we see a consistent performance with AT to near the level of ideal error-free BP. (Fig.5 (e) and (f)).

To demonstrate the generality of the estimation profile, we added a new experiment, repeating the training with three additional profiles of varying mismatches to the actual transmissions. The results show consistent performance even with different estimations. (Fig.3 (d)).

To better illustrate the generality of estimation profiles and error-tolerance of AT to the readers, we have made the following modifications:

Results section, “AT for training deep PNN with only output neuron information” subsection (P7, 8): “To validate the generality of the estimation profile used, we repeat the training experiment with multiple varying estimation profiles. Details of the additional estimation profiles can be found in Supplementary. As shown in Fig.3 (d), the training accuracies all reached 96%, showcasing that different estimation profiles can all allow training near the ideal BP performance.”

Results section (P8): Fig.3 (d)

Results section, “Generalization and repeatability of the AT method” subsection (P10, 11): “For pressure testing the AT method, the standardized distortion is varied between 0 and $2\sigma_{\text{phy}}$ to investigate the behavior under different levels of error. AT shows a high error-tolerance, maintaining the test accuracy for the MNIST dataset over 90% for up to twice the error seen at the experimental level. Due to the physical boundaries of the PNN systems, we can define a physically relevant region for the system (see Methods). For all the relevant regions, AT shows consistent performance.”

Results section (P10): Fig.5 (d), (e), (f)

Methods section (P13-15): subsections of “The training mechanism of asymmetrical training estimator”, “The robustness of AT and how the physical boundaries of PNNs protects AT”, and “Construction of the digital parallel model”

4. There are some grammar issues and typos, for example, at line 86.

Thank you for your detailed comment. We have now fully revised the manuscript and read through the manuscript multiple times among the authors to check for grammatical errors.

Response to Reviewer #2:

Original comment:

Reviewer #2 (Remarks to the Author):

Wang et al. report a new method for training physical neural networks (PNNs) which they call “asymmetrical training” (AT). The manuscript is well-written and incorporates theory, simulation, and experiments on a proof-of-principle photonic integrated circuit. While the idea is interesting, I found some points unclear that need to be addressed before I could make a recommendation on publication.

I have a few major concerns regarding the central claim of the paper:

1. I am skeptical of the claim in the abstract and repeated multiple times in the paper that AT can achieve calibration-free training of physical neural networks. The authors claim this is the advantage over prior work, but I am not convinced this claim holds up.

The authors claim they only need information on the input, the output, and the neuron connectivity. However, it is clear there needs to be some calibration, at the very least to understand the response function of the physical hardware. In the experiments presented in the paper, the authors know a priori the transfer function of the MZIs used on the chip. It makes sense AT will work for matrix-vector multiplication accelerators like the one used in this work, as we know how MZIs behave and how they can implement computation.

However, the term “physical neural network” is more broadly encompassing to describe systems where the response function is not known a priori—for example, the SHG system or mechanical systems reported in Wright & Onodera, Nature 601 (2022). In that paper, a digital twin of the hardware is trained in silico to obtain gradients. For these systems, I think AT would similarly require some digital twin of the hardware, thereby casting doubt on the claim that only input/output and neuron connectivity information is needed.

In general, the manuscript is confusing on this point. Sometimes the authors claim no information is needed on the physical transformation (for instance, see line 80, but the claim is repeated multiple times) and call the method “calibration-free” (line 60), while other times it is indicated some information is needed (for example, line 151: “As part of the procedure for conducting the AT method, we need to acquire an estimation profile mapping for the physical control parameters”). Please clarify this in the manuscript and revise the terminology.

2. A related point is how similar the initialization needs to be between the digital model and the physical hardware when starting off training. Can the two be truly randomly initialized? Or does the digital model need to be initialized in a setting that is “close” to the current hardware settings? I imagine this could be challenging without some calibration. More concretely: for a given hardware initialization, if the digital model is initialized randomly many times to different settings, will AT always successfully train?

3. There seems to be some similarities to physics-aware training (PAT) introduced in Wright & Onodera et al., Nature 2022. Hardware updates are a mix of two update terms: one is inputting the physical error vector, measured on the hardware, into the backwards pass of the digital model; and the second is auto-differentiating the digital model itself. It seems to me that the first term is similar to the update procedure in physics-aware training (PAT) (for

instance, see Fig. 3a in the PAT paper). The novelty would then appear to be introducing the second term, which is auto-differentiating the digital model, but it's not clear to me how this provides an advantage. It would be helpful to clarify in the manuscript the difference between these two approaches and the relative advantages of AT.

4. How does the efficiency of AT scale with the number of model parameters? Does it scale as well as backpropagation on digital hardware? The tradeoff with many of these algorithms is the reduced efficiency, and so it would be useful to understand the scaling for AT.

5. I would highly recommend making the code for the algorithm public so that it can be implemented on other platforms by the community.

Below are a list of more minor technical points that should be addressed:

6. Figure 2c: please include a schematic of the chip and not just chip images, as it will be difficult for many readers to understand the hardware architecture.

7. Figure 2e/f: what do the faded lines represent? Are they from other heaters? Please indicate this in the figure legend.

8. Figure 2: the caption refers to the PIC as a deep PNN. It looks like it implements a single layer. Please clarify.

9. Figure 3c: the confusion matrices do not add up to 100%. Please double check the data.

10. Figure 3/4: are the accuracies being reported for the training or test set? For training experiments, it is also important to know the performance on unseen data.

11. Figure 3d: why is there a training curve shown for calibrate and transfer? Isn't the model trained offline and then ported to the hardware?

12. Figure 3a implies the activation function is done in the photonics, but the PIC in Figure 2 implies it is performed digitally. Please clarify.

13. In both Fig. 3 and Fig. 4 the authors compare to the "theoretical maximum accuracy" of the model. How is this obtained?

Response:

Dear Reviewer #2,

Thank you for your comments, we have carefully read your comments and made modifications based on your suggestions. We have fully revised the manuscript and added new experimental and simulation data. The discussion and theories are also re-written to clearly present the motivation and characteristics of the AT method compared to existing approaches. We hope to address your concerns with the revised manuscript and following response.

1. I am skeptical of the claim in the abstract and repeated multiple times in the paper that AT can achieve calibration-free training of physical neural networks. The authors claim this is the advantage over prior work, but I am not convinced this claim holds up.

The authors claim they only need information on the input, the output, and the neuron connectivity. However, it is clear there needs to be some calibration, at the very least to understand the response function of the physical hardware. In the experiments presented in the paper, the authors know a priori the transfer function of the MZIs used on the chip. It makes sense AT will work for matrix-vector multiplication accelerators like the one used in this work, as we know how MZIs behave and how they can implement computation.

We would like to take this opportunity to re-explain the concept and motivation of the asymmetrical training (AT) method compared to existing methods. While some BP-based PNN training methods exists (such as the physics-aware training, PAT), the limitation of these existing methods is the requirement to know the hidden neuron information for a deep PNN structure to construct a reliable differentiable digital model in the backwards pass:

This means that flow of physical information is truncated at every layer represented by the physical processing module (PPM). The truncation of the analogue physical propagation means two things: 1. The processing speed is slowed because the system can't perform in-propagation analogue processing for the entire network structure. 2. Excessive interfaces between the analogue and digital domains are created due to the need to shuttle data across. To solve these problems, the most important thing is to find a method which can train with minimum information access points and allow the information propagation of the PNN to be encapsulated within the analogue domain.

Consequently, the idea is that AT can train even with only the output layer neuron information. Previously, when referring to “knowing only the input, output and connectivity of the network”, what we meant is that we only access the neuron information at the output layer for an isomorphic PNN. In comparison, PAT would require the neuron output information for every layer, which results in undesired delay in processing, leading to higher training time and more energy consumption.

We apologize if the previous terminologies have caused confusions, we have now fully revised the manuscript to better describe the difference between AT and existing methods, and the advantages of utilizing AT.

We would like to highlight that knowing a very limited level of physical information is not sufficient for training. For example, you can see in Fig.2, 3, and 4, where we experimentally demonstrated that when the network is trained in-silico then transferred to the device, the performance is significantly degraded. Also, for the experiments, the same estimation profile is applied to all the physical controls, the significant difference between the estimation profile and the actual transmission behaviors can be found in Fig.2 (f). Consequently, it is necessary to use AT for training when there is limited information regarding the transformation of the PPM. More importantly, construction of the estimation profile with the cosine-like shape doesn't involve any readouts from the PIC device. Consequently, the encapsulation of the deep PNN structure for ultra-fast processing is preserved with the AT method.

However, the term “physical neural network” is more broadly encompassing to describe systems where the response function is not known a priori—for example, the SHG system or mechanical systems reported in Wright & Onodera, Nature 601 (2022). In that paper, a digital twin of the hardware is trained in silico to obtain gradients. For these systems, I think AT would similarly require some digital twin of the hardware, thereby casting doubt on the claim that only input/output and neuron connectivity information is needed.

In general, the manuscript is confusing on this point. Sometimes the authors claim no information is needed on the physical transformation (for instance, see line 80, but the claim is repeated multiple times) and call the method “calibration-free” (line 60), while other times it is indicated some information is needed (for example, line 151: “As part of the procedure for conducting the AT method, we need to acquire an estimation profile mapping for the physical control parameters”). Please clarify this in the manuscript and revise the terminology.

As you have mentioned, methods such as PAT require a digital twin model for the backwards pass during training. In fact, AT’s novelty originates from the fact that AT utilizes the digital model for both the forward and backward pass. We apologize if this was not well-communicated in the first version. We have included a summary in the table below.

Method	Digital forward pass?	Digital backwards pass?	Physical forward pass?	Neuron access for training	Access time steps
AT	Yes	Yes	Yes	P	1
PAT	No	Yes	Yes	M	$N + 1$

(P : number of output layer neuron, M total neuron excluding input layer, $P < M$, N : total number of hidden layers.)

As we have briefly discussed above, AT’s motivation is to help training of fast encapsulated deep PNNs with readouts at only output layer. This means that AT is accessing much less physical neuron information compared to PAT. Consequently, there needs to be some mechanism to compensate for the lack of hidden neuron information. We achieve this by utilizing the additional forward pass in the digital domain. The forward pass in the digital domain is known to be a well-regulated pass trained by ideal BP (see Eq.3 and Results section), which offers a reference point for the physical transformation of the PPM to align towards.

For most of the PNN training methods, such as PAT, the concept is to extract a lot of physical information and rely on the physical domain to solve the problem. However, this necessarily means that the digital domain is idle for the majority of the training duration, creating significant inefficiencies. In contrast, the idea of AT is to leverage the ability of processing in both the analogue and digital domain to accelerate training overall.

We have fully revised the manuscript and changed the terminologies where appropriate to avoid confusion for the readers. We have now removed the previous terminology of “grey-box” networks and replaced it with “encapsulated” networks to describe the system where in-propagation processing is preserved for the entire deep network structure until the output layer. We have also more rigorously used the term “scalable isomorphic physical neural networks” to avoid confusion with the non-isomorphic networks seen in methods such as reservoir computing.

Some of the key modifications include:

Abstract (P1): **Entire abstract**

Introduction section (P1,2): **Second and third paragraphs: “However, existing BP-based PNN training methods... and higher energy consumption (to maintain the operation of the PNN system for longer).”**

Introduction section (P2): **Table 1**

Results section (P3-5): **Entire section, but more specifically in the subsection of “Asymmetrical training method”, where the difference between AT and previous methods are explained in more detail.**

2. A related point is how similar the initialization needs to be between the digital model and the physical hardware when starting off training. Can the two be truly randomly initialized? Or does the digital model need to be initialized in a setting that is "close" to the current hardware settings? I imagine this could be challenging without some calibration. More concretely: for a given hardware initialization, if the digital model is initialized randomly many times to different settings, will AT always successfully train?

We think there might be some miscommunication regarding the procedure of AT; the initialization of the digital model and physical model are always set with the same randomized parameter. We have now fully revised the Results section to describe the procedures for implementing AT. We would like to take this opportunity to re-explain the procedure of AT.

At the initial time instance, the physical and digital model's parameters are always set with the same values. The specific values set to the model is randomized, just like the initialization for any training. However, we must note that setting the two systems with the same parameters will not lead to the same transformation: $f_m(W) \neq f_p(W)$. This is because there is always a mismatch between our knowledge of the physical control-transformation mapping and the actual behavior. More specifically, we can note that the estimation profile we used is significantly different from the actual transmission behavior of the MZI units (Fig.2 (f)).

Consequently, despite the two systems are always set with the same set of values, the difference between the resulting transformation always leads to different updates to the digital and physical model. Therefore, the name "asymmetrical training" isn't referring to the systems being asymmetrically initialized, but instead, describing that the initial symmetrical parameter setting ($t = 0$) gradually becomes asymmetrical ($t > 0$). Hope this makes more sense now.

We have made the following modifications to better explain the procedure of AT:

Results section, "Asymmetrical training method" subsection (P3-5):

"For the AT method, we first define two sets of parameters for the parallel model $\mathbf{W}_{(t,dig)}$ and physical control $\mathbf{W}_{(t,phy)}$. (We use \mathbf{W} to refer to all the controllable parameters in the following for simplicity.) At the first time-instance of $t = 0$, we set the digital and physical control with the same randomized values, $\mathbf{W}_{(0,dig)} = \mathbf{W}_{(0,phy)}$ (note that the two resulting transformations are different: $f_m(\mathbf{W}_{(0,dig)}) \neq f_p(\mathbf{W}_{(0,phy)})$). The digital and physical systems simultaneously forward propagate the same sample information, resulting in a pair of predictions $\mathbf{a}_{(0,dig)}^{[N+2]}$ and $\mathbf{a}_{(0,phy)}^{[N+2]}$. The pair of predictions leads to two different sets of errors ($\mathbf{E}_{(0,dig)}$ and $\mathbf{E}_{(0,phy)}$) at the output layer for the same targets. Despite there is no access to intermediate neuron information of physical system, we have the complete information of the parallel model ($\mathbf{a}_{dig}, \mathbf{W}_{dig}, \mathbf{b}_{dig}, \mathcal{G}_{dig}$).

The collection of parallel model parameters is used for backpropagating the two sets of errors. We define two update terms based on the backpropagating the two errors $\mathbf{E}_{(dig)}$ and $\mathbf{E}_{(phy)}$. When backpropagating $\mathbf{E}_{(dig)}$, we get the gradient update to the digital parallel model as $\Delta \mathbf{W}_{dig} = \partial L / \partial \mathbf{W}_{dig} \equiv \delta \mathbf{W}_{dig}$. On the other hand, results from backpropagating $\mathbf{E}_{(phy)}$ has no direct physical significance on its own; we call it the pseudo update $\Delta \mathbf{W}_{pseudo}^{[l]}$. The AT estimator is defined as the mixture of these two updates under the regulation of the mixing ratios M_{w1} and M_{w2} , which normally takes the equal values of $M_{w1} = M_{w2} = 0.5$. The overall expression is shown in Eq. (3). It is important to note that the AT estimator is obtained through the mixing of the updates at each layer, and not the direct mixture of errors at the output. (See Methods and Supplementary for more theoretical discussions).

$$\Delta \mathbf{W}_{AT}^{[l]} = M_{w1} \cdot \Delta \mathbf{W}_{dig}^{[l]} + M_{w2} \cdot \Delta \mathbf{W}_{pseudo}^{[l]} \quad (3)$$

At every later time instance ($t > 0$), the digital parallel model is updated solely with the gradient update of $\Delta W_{\text{dig}}^{[t]}$, leading to an ideal BP behavior for the parallel model. The PNN’s physical control parameters are updated with $\Delta W_{\text{AT}}^{[t]}$. We see the two updates are always different at every time instance. Consequently, the control settings to the digital and physical system become asymmetrical ($W_{(t,\text{dig})} \neq W_{(t,\text{phy})}$, for $t > 0$) as the training progresses, giving the name of this method.”

More concretely: for a given hardware initialization, if the digital model is initialized randomly many times to different settings, will AT always successfully train?

Consequently, because the initialization is always the same for the digital and physical model, we don’t choose different digital model settings for a single hardware setting. However, to reassure the generality of the AT method, we have now added additional experiments: we repeated the training experiment with three additional varying estimation profiles, the training results show consistent performance (Fig.3 (d)). For all three cases, we are able to achieve consistent performance, showing the generality of the estimation profile.

The consistent result for AT under different levels of mismatch is also studied through simulation in Fig. 5 (d). Also, we have now included further discussion in the Methods section, describing how the physical boundaries of the PNN systems help to regulate the robustness of the AT method.

To illustrate the generality and robustness of the AT method, we have made the following modifications:

Results section, “AT for training dep PNN with only output neuron information” subsection (P7, 8): “To validate the generality of the estimation profile used, we repeat the training experiment with multiple varying estimation profiles. Details of the additional estimation profiles can be found in Supplementary. As shown in Fig.3 (d), the training accuracies all reached 96%, showcasing that different estimation profiles can all allow training near the ideal BP performance.”

Results section (P8): Fig.3 (d)

Results section, “Generalization and repeatability of the AT method” subsection (P9-11): “For pressure testing the AT method, the standardized distortion is varied between 0 and $2\sigma_{\text{phy}}$ to investigate the behavior under different levels of error. AT shows a high error-tolerance, maintaining the test accuracy for the MNIST dataset over 90% for up to twice the error seen at the experimental level. Due to the physical boundaries of the PNN systems, we can define a physically relevant region for the system (see Methods). For all the relevant regions, AT shows consistent performance.”

Results section (P10): Fig.5 (d)

Methods section (P13-14): Entire subsections of “The training mechanism of asymmetrical training estimator”, “The robustness of AT and how the physical boundaries of PNNs protects AT”.

3. There seems to be some similarities to physics-aware training (PAT) introduced in Wright & Onodera et al., Nature 2022. Hardware updates are a mix of two update terms: one is inputting the physical error vector, measured on the hardware, into the backwards pass of the digital model; and the second is auto-differentiating the digital model itself. It seems to me that the first term is similar to the update procedure in physics-aware training (PAT) (for instance, see Fig. 3a in the PAT paper). The novelty would then appear to be introducing the second term, which is auto-differentiating the digital model, but it’s not clear to me how this provides an advantage. It would be helpful to clarify in the manuscript the difference between these two approaches and the relative advantages of AT.

The key difference between asymmetrical training (AT) and physics-aware training (PAT) originates from the motivation for creating AT. Despite some BP-based PNN training methods such as PAT exist, the key limitation of these methods is the requirement to know the physical neuron information at each hidden layer to construct a differentiable digital twin model for performing backpropagation (BP). However, this necessarily means two things: 1. The propagation of the analogue signal must be truncated at each layer. 2. The need to access the neuron information creates excessive readouts and conversions between the analogue-digital interfaces.

The need to truncate the physical information propagation in methods such as PAT inevitably leads to slower processing, compromising the initiative of PNN implementations for fast in-propagation processing. On the other hand, AT preserves the completeness of the physical information within the analogue domain until the output layer, allowing for fast in-propagation processing for the entire deep network structure.

Furthermore, the speed in methods such as PAT is bottlenecked by the interface speed between the analogue and digital domain. The need to readout the intermediate hidden neuron scales with $O(M)$, where M represents the total number of neurons in the system excluding the output layer. This means that by extracting the physical information, and then convert the information back into the analogue domain requires at least $O(2M - P)$ analogue-digital interfaces. In comparison, AT only needs to access neuron information at the output layer, scaling as $O(P)$ with P output neurons. We know that for any sensible deep network structure, $M \geq 2P$. Therefore, for any deep networks, we can see a significant improvement in speed for AT-compatible encapsulated systems compared to PAT-compatible systems.

Also, PAT can't train with solely the output layer neuron information; this can be seen with the newly added experimental demonstration in Fig.3 (a). The pseudo intermediate-access physics-aware backpropagation (IP-BP) performance indicates the training behavior when PAT only has access to the output layer information and not the hidden layer information. The constructed differentiable model is no longer an accurate description of the actual physical transformation, leading to degradation of performance.

There are also practical advantages for using AT in realistic systems compared to PAT. PAT relies on the fidelity of the analogue information acquired at each hidden layer. However, we know that for realistic physical systems, there are external factors which can affect the system state and the signal fidelity. For example, when the physical system state experiences a sudden unexpected strong perturbation, the twin model in PAT is no longer a suitable description for the physical system. This can lead to failure to train with a diverging loss. Similarly, when the system is sensitive to the operational environment (such as temperature fluctuations), the signal fidelity can be affected, and the system state might experience slight fluctuations. This can also lead to failure of training with PAT. On the other hand, the additional forward digital pass in AT is always being trained by an ideal BP. The digital twin is not affected by the variation in the physical domain, protecting the update at each time instance. These scenarios are studied in simulation and shown in Fig.5 (g) and (h) of the revised paper.

As briefly mentioned above, AT allows faster, and more energy efficient training compared to PAT due to the continuous propagation of analogue signal and bypassed analogue-digital interfaces. Even when considering the additional forward pass required in the digital domain, we can see a significant improvement in efficiency for different network sizes. We duplicate the results from Fig.6 here for easier viewing:

We have now fully revised manuscript to better illustrate the difference between AT and PAT, and further highlight the motivation for utilizing AT. Please see the new version of the paper for details. We would like to highlight a few places where we made the modifications:

Introduction section (P1): “Backpropagation¹³ (BP) finds the gradient update to the loss function for the network parameters, offering fast training time step convergence¹⁴. However, existing BP-based PNN training methods such as physics-aware training¹⁵ (PAT), in-situ BP¹⁶, dual adaptive training¹⁷ (DAT), and hybrid training¹⁸ (HT), relies on knowing the net output and activation level information for hidden layer neurons in a deep network structure. This means that the propagation of the analogue signal needs to be truncated and readout at each layer. The truncation of the physical information propagation undermines the advantage of ultra-fast in-propagation processing with PNNs¹⁹ (See Fig.1 (a)). Furthermore, the need to shuttle the information back-and-forth across the analogue-digital (AD) interface creates excessive analogue-to-digital conversions (ADCs) and digital-to-analogue conversions (DACs). ADC and DAC are known to be speed bottlenecks^{20,21}, where the AD interface speed²² is much slower than solely in-analogue¹⁹ or in-digital⁴ computation (see Fig.1 (b)). These factors lead to overheads of increased training time, additional energy consumption, and increased fabrication cost, compromising PNNs’ goals for fast, energy-efficient and cheap processing²³.”

Results section (P3-8): Entire subsection of “Asymmetrical training method”, “AT method for training fully encapsulated PhPNNs”, and “AT for training deep PNN with only output neuron information”

Results section, subsection of “Generalization and repeatability of the AT method” (P10,11): Fig.5 (g) and (h), and last paragraph of the subsection: “Existing IP-BP PNN training methods rely on the accuracy of neuron information acquired at the hidden layers ... the update at each time step is protected by the overall AT estimator, thus not affected by the noisy system.”

Discussion section(P11-13): Entire section and Fig.6.

4. How does the efficiency of AT scale with the number of model parameters? Does it scale as well as backpropagation on digital hardware? The tradeoff with many of these algorithms is the reduced efficiency, and so it would be useful to understand the scaling for AT.

The key motivation of AT is that it can allow for more time and energy efficient training compared to existing methods. We have included the newly added Table 1 below:

Table 1: Comparison of relevant PNN training methods								
Method	No. of physical propagation	Digital model?	Need calibration?	Intermediate hidden layer readout?	Information access points	Readout time steps	Expected training convergence	Ref.
BP-based methods								
AT	1	Yes	No	No	P	1	$\approx O(T_0)$	This work
PAT (HT)	1	Yes	No	Yes	M	$N + 1$	$\approx O(T_0)$	20,23
In-situ BP	3	No	No	Yes	M	$N + 1$	$\approx O(T_0)$	21,31
DAT	1	Yes	Yes	Yes	M	$N + 1$	$\approx O(T_0)$	22
Non-BP-based methods								
DFA	1	No	Yes	No	P	1	$> O(T_0)$	32,33
FF	2	No	No	No	$2P$	2	$> O(T_0)$	16
Finite stochastic	1	No	No	Yes	M	$N + 1$	$\gg O(T_0)$	34,35
Gradient free	1	No	No	No	P	1	$\gg O(T_0)$	17,18

(M represents the total number of neurons excluding the input layer, N is the number of hidden layers, P is the number of neurons in the output layer, and T_0 is a reference convergence of perfect information BP. The two variables M and P always follow the behavior of $P < M$, and $N \geq 1$ for deep PNNs.)

As you mentioned, there are often tradeoffs with PNN training methods. For example, the existing BP-based methods all require access to hidden neuron information, which creates excessive ADCs and DACs as we discussed above. These additional conversions and data movements not only slow the training speed, but also add significantly to the energy consumption. Most importantly, the need to access intermediate neuron information implies that the analogue propagation needs to be truncated or split, which means slowed processing speed and higher input power to maintain the fidelity. All these factors can undermine the incentives of PNNs for faster, more energy efficient and cheaper processing.

On the other hand, methods which require less access points in the PNN typically have longer training convergences or more than one physical propagation for each sample. For most training tasks, the sample size is a considerable factor. The additional iterations required due to the slower convergence of these methods can mean a significant time and energy consumption overhead added.

Consequently, the motivation of AT is to combine the fast convergence of BP-based methods, with the reduced neuron information access of some non-BP-based methods. Most importantly, serving as a compatible method for constructing encapsulated deep PNNs to achieve fast processing for both inference and training.

From the aspect of the physical domain, we see that AT only requires the output neuron information, this means that it scales well for larger size networks (as the output size is typically unchanged for a given task). If we consider a task which requires P output neurons, and M total neurons (excluding the input layer). When we scale up the network parameters for higher complexities, existing BP-based methods would require information access point to scale with the network size. In comparison, AT's access point is always P , meaning that it scales much better than other BP-based method. This can be seen in Fig.6.

From the aspect of the computational overheads, the computation for the two parts of the AT estimator (the digital BP part and the pseudo update part) are both computed with the digital model's parameters. This means that the computation of the update direction is a parallel process for the digital and physical domain by simply concatenating the error matrices during the backpropagation. Consequently, AT has the same computational time step as a conventional digital BP. While there will always be some overheads for training PNN compared to pure digital training without errors, AT's overheads are significantly reduced compared to existing methods.

To better describe the motivation and efficiency of using AT to the readers, we have made the following modifications:

Abstract (P1): Entire abstract

Introduction section (P1,2): Paragraph 2 and 3

Introduction section (P2): Table 1

Discussion section (P11-13): Entire Discussion section

5. I would highly recommend making the code for the algorithm public so that it can be implemented on other platforms by the community

Thank you for mentioning this, we are happy to share the code for demonstration of the AT algorithm.

We have uploaded a demo of the AT method at: <https://github.com/yz-david-wang/AsymmetricalTrainingDemo>

6. Figure 2c: please include a schematic of the chip and not just chip images, as it will be difficult for many readers to understand the hardware architecture.

We have now included a schematic of the chip in the Supplementary Information. Due to the newly added experiment for the fully encapsulated network, we have moved some of the previous information in Fig.2 to the

Supplementary. We have left a note in the description to direct the readers to the Supplementary if they would like to see the schematic.

7. Figure 2e/f: what do the faded lines represent? Are they from other heaters? Please indicate this in the figure legend.

The faded lines in Fig. 2 (e) and (f) are the characterized transmissions of the MZI units. The idea was to show that there is a clear separation between the estimation profile used in the experiment and the actual transmissions of the physical controls. (The same estimation profile is applied to all the physical controls.) We have now clarified this in the description and in Fig.2.

8. Figure 2: the caption refers to the PIC as a deep PNN. It looks like it implements a single layer. Please clarify.

We have now added a new experiment of fully encapsulated deep PNN, where two PICs are simultaneously used for constructing the network. We have now made the modification in the Results section of the schematic to show the implementation of a fully encapsulated deep PNN, where the entire processing is contained within the analogue domain.

We have also further clarification for the experiments information in both Results section and Methods section.

9. Figure 3c: the confusion matrices do not add up to 100%. Please double check the data.

Thank you for mentioning this, this was due to a formatting error in Python plotting. For example, in the “versicolor” category, it had 34 total samples which had 33 correctly classified, so that would be $(33/34) 0.97058824\dots$ for the correct prediction and $(1/34) 0.02941176\dots$ for the incorrect prediction. We previously used the formatting of “.2g”, which leaves only two significant figures, so it ended up as “0.97” and “0.029”, which lead to the confusion. We have now updated the two confusion matrices to the new testing data result and used “.2f” for leaving two figures to the right of zero rather than as significant figures. These issues have now been addressed, please see the new Fig.3 and 4.

10. Figure 3/4: are the accuracies being reported for the training or test set? For training experiments, it is also important to know the performance on unseen data.

It was previously for the training accuracy of those systems; we have now added testing experiments to both tasks. As mentioned above, the confusion matrices have also been updated to represent the testing data.

11. Figure 3d: why is there a training curve shown for calibrate and transfer? Isn't the model trained offline and then ported to the hardware?

Originally, the performance for calibrate-and-transfer (with a very rough calibration) was recorded alongside the training with AT at each iteration. However, we realized that the idea of a “rough” calibration is confusing to the readers and doesn't contribute much as a benchmark. Consequently, after careful thought, we decided to remove those results. Instead, we added a new line of “pseudo IP-BP” which refers to the performance of existing BP-based methods (such as PAT), when only the output neuron information is provided. This should now serve as a much more clarified and useful benchmark for illustrating the advantage of AT for training encapsulated deep systems.

12. Figure 3a implies the activation function is done in the photonics, but the PIC in Figure 2 implies it is performed digitally. Please clarify.

Previously, non-linearity was digitally added. We have now included a new experimental result where the non-linearity is directly added photonically with near-saturation EDFAs. The new experiment results can be found in Fig.2 and the subsection “AT method for training fully encapsulated PhPNNs”.

The results which are trained with the digitally added non-linearity are kept for illustrating other important concepts in the subsections of “AT for training deep PNN with only output neuron information” and “AT’s compatibility with scale-up techniques”. We have made clarification where the activation is added digitally.

13. In both Fig. 3 and Fig. 4 the authors compare to the “theoretical maximum accuracy” of the model. How is this obtained?

This was meant to refer to the benchmark of an ideal BP training of the same network structure. For ideal BP, we train the model solely digitally using BP without any added errors. This was meant to serve as a benchmark for the maximum achievable performance of the hardware network size. We have now replaced the term with “ideal error-free BP” to avoid ambiguity and defined the term when first used in the Results section.

Response to Reviewer #3:

Original comment:

Reviewer #3 (Remarks to the Author):

This work proposes an asymmetrical training method for training photonic neural networks to address inaccurate network information caused by physical errors. The proposed method utilizes a digital parallel model to assist in gradient calculation during parameter updates, allowing photonic integrated circuits to be treated as a grey-box during these updates. That is, only prior knowledge of topological neuron connectivity, input, and output is required. Experimental results based on a photonic chip with 4x4 matrix-vector multiplication demonstrate improved classification accuracy compared to direct backpropagation and calibrate & transfer methods.

However, after carefully evaluating the manuscript, I found that this work is insufficient for publication in Nature Communications due to the lack of comparisons with state-of-the-art training methods, insufficient motivations, and a small network scale.

The research doesn't provide insightful experimental comparisons with state-of-the-art training methods. With some comparisons, we could identify the major drawbacks of the proposed method. For example, compared to the physical-aware training method [45] and hybrid training [c1], besides the fact that the digital parallel model might be inaccurate for characterizing the physical system, the inaccurate output of the digital parallel model is further used in gradient calculation for parameter updating. This makes the gradient even less accurate than with physical-aware training, which results in lower performance compared to the physical-aware training method. Additionally, comparisons with other training methods, including feedback alignment training [c2], dual adaptive training [47], forward-forward (backpropagation-free) training [49, c3], etc., have not been conducted. The experimental results presented in the manuscript also did not provide side-by-side comparisons with state-of-the-art methods.

The motivation for designing the training method as a grey-box is unclear, as the fundamental building blocks for chip design are typically known. As stated in the manuscript, models of MMI and MZI-based photonic integrated circuits can be explicitly described. Therefore, the scenarios for the proposed grey-box photonic neural network training method should be clearly provided.

The experimental results only cover a very small network scale, with a theoretical maximum MNIST classification accuracy of only 85%. Evaluation on a larger network scale should be performed.

[c1] Spall, J., et al. Hybrid training of optical neural networks. *Optica*. 2022 Jul 20;9(7):803-11.

[c2] Filipovich, Matthew J., et al. "Silicon photonic architecture for training deep neural networks with direct feedback alignment." *Optica* 9.12 (2022): 1323-1332.

[c3] Momeni, Ali, et al. "Backpropagation-free training of deep physical neural networks." *Science* 382.6676 (2023): 1297-1303.

Response:

Dear Reviewer #3,

Thank you for your comments. After carefully reading your comments, we believe that there was miscommunication in the first manuscript that overshadowed the benefits and motivation of AT. We have now fully revised the manuscript with newly added experiments, analysis and discussion. We hope the new version and our response here will clarify your concerns.

Response part 1:

“However, after carefully evaluating the manuscript, I found that this work is insufficient for publication in Nature Communications due to the lack of comparisons with state-of-the-art training methods, insufficient motivations, and a small network scale.

The research doesn't provide insightful experimental comparisons with state-of-the-art training methods. With some comparisons, we could identify the major drawbacks of the proposed method. For example, compared to the physical-aware training method [45] and hybrid training [c1], besides the fact that the digital parallel model might be inaccurate for characterizing the physical system, the inaccurate output of the digital parallel model is further used in gradient calculation for parameter updating. This makes the gradient even less accurate than with physical-aware training, which results in lower performance compared to the physical-aware training method. Additionally, comparisons with other training methods, including feedback alignment training [c2], dual adaptive training [47], forward-forward (backpropagation-free) training [49, c3], etc., have not been conducted. The experimental results presented in the manuscript also did not provide side-by-side comparisons with state-of-the-art methods.”

We would like to start by explaining the motivation for creating the AT method. While there exist some PNN training methods such as physics-aware training (PAT), the limitation of these methods is the requirement to access the neuron information at every layer. However, intermediate neuron access necessarily means two things: 1. The propagation of the analogue signal must be truncated at each layer. 2. The need to access the neuron information creates excessive readouts and conversions between the analogue-digital interfaces.

The need to truncate the physical information propagation in methods such as PAT inevitably leads to slower processing, compromising the initiative of PNN implementations for fast analogue in-propagation processing. On the other hand, AT preserves the completeness of the physical information within the analogue domain until the

output layer, allowing for fast in-propagation processing for the entire deep network structure to be encapsulated within the analogue domain.

Furthermore, the speed of methods such as PAT is bottlenecked by the interface speed between the analogue and digital domain. For existing BP-based methods such as PAT, the need to readout the intermediate hidden neuron scales with $O(M)$, where M represents the total number of neurons in the system excluding the output layer. This means that by extracting the physical information, and then convert the information back into the analogue domain requires at least $O(2M - P)$ analogue-digital interfaces. In comparison, AT only needs to access neuron information at the output layer, scaling as $O(P)$ with P output neurons. We know that for any sensible deep network structure, $M \geq 2P$. Therefore, for any deep network structure, we can see a significant improvement in speed for AT-compatible systems compared to PAT-compatible systems.

We have now added new experimental results based on your comment to show that existing methods, such as PAT, can't train with solely the output layer neuron information; this can be seen with the newly added experimental demonstration in Fig.3 (a). The pseudo intermediate-access physics-aware backpropagation (IP-BP) performance indicates the training behavior when PAT only has access to the output layer information and not the hidden layer information. The constructed differentiable model is no longer an accurate description of the actual physical transformation, leading to degradation of performance.

Consequently, it is necessary to have methods such as AT for training an encapsulated deep PNN. We have now also added an experimental validation of AT for fully encapsulated deep networks in the subsection “AT method for training fully encapsulated PhPNNs”.

There are also practical advantages for using AT in realistic systems compared to PAT. PAT relies on the fidelity of the analogue information acquired at each hidden layer. However, we know that for realistic physical systems, there are external effects which can affect the system state and the signal fidelity. For example, when the physical system state experiences a sudden unexpected strong perturbation, the twin model in PAT is no longer a suitable description for the physical system. This can lead to failure to train with a diverging loss. Similarly, when the system is sensitive to the operational environment (such as temperature fluctuations), the signal fidelity can be affected, and the system state might experience slight fluctuations. This can also lead to failure of training with PAT. On the other hand, the additional forward digital pass in AT is always being trained by an ideal BP. The digital twin is thus not affected by

the variation in the physical domain, protecting the update at each time instance. These scenarios are studied in simulation and shown in Fig.5 (g) and (h) of the revised paper.

As briefly mentioned above, AT allows faster, and more energy efficient training compared to IP-BP methods due to the continuous propagation of analogue signal and bypassed analogue-digital interfaces. Even when considering the additional forward pass required in the digital domain, we can see a significant improvement in efficiency for different network sizes. We duplicate the results from Fig.6 here for easier viewing:

AT is dedicated to combining the advantages of the fast convergence of BP-based methods, and the low level of information requirement in other non-BP-based methods to improve the time and energy efficiency of training overall. Some of key metrics for relevant training methods are shown below (and in Table 1 of the main article).

Table 1: Comparison of relevant PNN training methods								
Method	No. of physical propagation	Digital model?	Need calibration?	Intermediate hidden layer readout?	Information access points	Readout time steps	Expected training convergence	Ref.
BP-based methods								
AT	1	Yes	No	No	P	1	$\approx O(T_0)$	This work
PAT (HT)	1	Yes	No	Yes	M	$N + 1$	$\approx O(T_0)$	20,23
In-situ BP	3	No	No	Yes	M	$N + 1$	$\approx O(T_0)$	21,31
DAT	1	Yes	Yes	Yes	M	$N + 1$	$\approx O(T_0)$	22
Non-BP-based methods								

DFA	1	No	Yes	No	P	1	$> O(T_0)$	32,33
FF	2	No	No	No	$2P$	2	$> O(T_0)$	16
Finite stochastic	1	No	No	Yes	M	$N + 1$	$\gg O(T_0)$	34,35
Gradient free	1	No	No	No	P	1	$\gg O(T_0)$	17,18

(M represents the total number of neurons excluding the input layer, N is the number of hidden layers, P is the number of neurons in the output layer, and T_0 is a reference convergence of perfect information BP. The two variables M and P always follow the behavior of $P < M$, and $N \geq 1$ for deep PNNs.)

To illustrate the strong and practical motivation of AT over existing methods, the necessity of AT for training encapsulated deep PNNs, and the enhanced performance and efficiency of AT, we have made the following modifications to the manuscript:

Abstract (P1): Entire abstract

Introduction section (P1, 2): Paragraph 2, 3, 4: “PNN implementation and training... allows AT to significantly improve the overall time and energy efficiency compared to existing training methods by eliminating AD conversions wherever possible during the training”

Introduction section (P2): Table 1

Results section (P3-11): Entire subsections of “Asymmetrical training method”, “AT method for training fully encapsulated PhPNNs”, “AT for training deep PNN with only output neuron information”, and “Generalization and repeatability of the AT method”

Results section (P3-11): Modifications have been made to Fig.2, 3, 4, 5

Discussion section (P11-13): Entire discussion section

Discussion section (P12): Fig.6

“The motivation for designing the training method as a grey-box is unclear, as the fundamental building blocks for chip design are typically known. As stated in the manuscript, models of MMI and MZI-based photonic integrated circuits can be explicitly described. Therefore, the scenarios for the proposed grey-box photonic neural network training method should be clearly provided.”

Thank you for mentioning this, we have realized that the term “grey-box” can cause confusion to the readers. Consequently, we have replaced it with “encapsulated” as a more appropriate description. The scenario for training with AT is different from methods such as PAT. In PAT for example, each layer is treated as a grey box, but the entire deep PNN structure isn’t, the output neuron information is extracted at each layer.

There are many advantages for constructing the deep PNN system as an encapsulated structure. When the propagation of the analogue information needs to be truncated for extracting the hidden layer neuron information, the in-propagation processing of the PNN is stopped and constrained by the interface speed between the analogue and digital domain. This breaks the advantage of PNN’s analogue in-propagation high-bandwidth processing. The additional readouts also create significant traffic at the interface, leading to more time and energy required for training. On the other hand, by constructing the system as a fully encapsulated deep PNN, the speed and energy efficiency of AT are vastly improved compared to existing methods.

Despite the model of components such as MMI and MZI can be described, we would like to highlight that this rudimentary description of the system is not sufficient to allow training on its own. This can be seen from the repeated degradation observed for “in-silico BP” performance across the experiments and the simulation. Therefore, when only a low-level understanding of physical transformation is available, there needs to be some additional techniques such as AT to achieve well-behaved training.

Also, it is necessary to have some sort of proxy such as the estimation profile used in the experiments. When the system is constructed as a fully encapsulated deep network, the overall transformation at the output layer is the encapsulation of the nonlinear transformations in all the previous layers. The encapsulated transformation of $f_{p,1} \left(f_{p,2} \left(f_{p,3} (\dots) \right) \right)$ means that the access to the explicit description of the physical components is not trivial in the hidden layers for an encapsulated deep network. As a result, despite individual component behavior can be

calibrated and described, the overall behavior can't. This is the reason why we must use some sort of estimation for applying the physical parameter controls.

Furthermore, we would like to highlight that the utilization of an estimation profile doesn't break the condition for constructing a fully encapsulated deep PNN. Acquisition of the estimation profile doesn't involve reading out from the physical processing module (PPM) at any stage of the training. Also, because the same estimation profile is applied to all the control parameters. AT enjoys the exact same advantage of any calibration-free methods. By utilizing the estimation profile, the required memory for controlling only scales as $O(R)$ (where R is the control resolution) rather than the typical $O(M^2R)$ in typical methods (where M is the total neuron number). This reduced memory requirement is as low as possible for physical training.

To better illustrate the advantages of constructing the encapsulated deep NN, and the necessity to use AT for training in such scenarios, we have made the following modifications:

Introduction section (P1, 2): Paragraph 2, 3, 4

Results section (P3-8): subsections of "Asymmetrical training method", "AT method for training fully encapsulated PhPNNs", "AT for training deep PNN with only output neuron information"

Results section (P6, 8): Fig.2, Fig.3 (a)

Discussion section (P11-13): Entire Discussion section

"The experimental results only cover a very small network scale, with a theoretical maximum MNIST classification accuracy of only 85%. Evaluation on a larger network scale should be performed"

The AT method is dedicated to train encapsulated deep PNNs, allowing for an architecture with higher time and energy efficiency. Consequently, the demonstrations serve as validations of AT and form a complete logic for AT to significantly improve time and energy efficiency from existing methods:

To start with, we demonstrated the training for a fully encapsulated deep PNN with AT. The trained structure involves multiple PPMs directly connected in the analogue domain. The propagation of the physical information is fully preserved until the output layer. AT has shown a significantly improved performance from in-silico BP. This is illustrated by the subsection of "AT for training fully encapsulated PhPNNs".

The premise of constructing a fully encapsulated PNN is the ability to train even with only the output neuron information. Therefore, we experimentally demonstrated that AT could train with only the output layer neuron information to achieve performance near an ideal error-free BP in the subsection of "AT for training deep PNN with only output neuron information".

Knowing that due to the typical fixed dimensionality of the PPM, we demonstrated that AT is compatible with existing scaling up methods. The results showed that AT is not affected by the further accumulation of error from using the PPM multiple times. This is shown in the subsection of "AT's compatibility with scale-up techniques". We also presented the methodology for constructing the system and compatible digital parallel model in this subsection and the Methods section.

Then, we illustrated that AT can equally be applied to larger-sized encapsulated deep networks through simulation in the subsection "Generalization and repeatability of the AT method". We have simulated for networks over 2.6×10^5 parameters and over 2000 neurons. For the larger simulated network sizes, AT shows consistent good performance near the ideal BP maximum.

Consequently, we think that the novelty of the AT method shouldn't be associated with the scale of the demonstration as we have presented AT with a complete flow of logic to back the claims. We have provided the relevant methodologies for scaling where appropriate. We have also demonstrated the compatibility and generality of the AT method for larger-sized networks.

To better present this work to the readers, we have made the following modifications to the manuscript:

Abstract and Introduction section: Entire abstract and introduction section for highlighting the motivation of AT and its difference from existing methods.

Results section: Entire results section, where the four subsections present a complete logical flow and provide the methodology for implementing the AT methods.

Discussion section: Entire discussion section, illustrating the practical benefits of improved time and energy efficiency to use AT over existing methods.

Methods section: Entire methods section, revised to clarify the theoretical foundation of AT and provide the appropriate methodology for constructing digital parallel models for scaled-up systems.

Response to Reviewer #4:

Original comment:

Reviewer #4 (Remarks to the Author):

This paper proposes asymmetric training of photonic neural networks as a way of training neural networks by aligning gradient directions with a parallel model of the device. Overall, the concept of averaging errors from the output of the actual and parallel models for training is an interesting idea for the practical training of PNNs without calibrating each of the layers.

As far as the results are concerned, I think the results for AT are solid and most graphs / eqs are fairly clear but I am still unclear on many things. What do "direct BP" results mean. "We see a significant performance degradation with BP when the physical information is not available." What physical information? What is an "error-free digital reference"? Unclear on what "calibrate and transfer" entails; why are the results so poor for the simulated results? I'd like more details on how the calibration is done so others can reproduce; I don't understand why the calibration resulted in such poor performance.

I have a few higher level issues with some of the attempts to compare backpropagation based approaches in the paper. Mainly, this paper invokes a lot of the same motivations as "Deep physical neural networks trained with backpropagation" (Wright et al, Nature, 2022) but appears to address these problems in a different manner through alignment. I think a closer comparison of these methods is warranted both in simulation and in experiment because of the similarities, i.e. backpropagating through a "simulated model" of the system. What are the advantages / disadvantages and what's different here (the results in the Wright paper seemed quite good...)? Is there a relationship between Wright et al's method to any of the "BP" approaches discussed in this paper?

Additionally, what makes calibration inferior (this is made fairly clear in the intro)? I would like some very specific calculations of the memory and time cost of calibration. Calibration is very useful for debugging and probing physical systems (the paper dismisses this using a somewhat vague analogy to Ising machines / reservoir computing, when MZI networks are really a fairly specific system). Most PNNs are made of MZI meshes which are readily calibrateable, and such procedures have been successfully demonstrated at fairly large scales in the commercial sector (e.g. Lightmatter in the past). So why revert to a procedure like this for such systems? What benefit would it provide (and to what extent?) compared to the costs in convenience of directly flashing NN weights and especially less effective debugging of the overall system?

Surely there are other systems to experimentally demonstrate AT than a PNN; and if that was the intent of the paper, there is currently no compelling argument for the superiority of this method over the accelerated training method proposed by Bandyopadhyay et al. which is actually model free and doesn't require calibration and is much more energy efficient (since you don't have to backprop on a digital machine). Is it convergence time?

Although most of the points above are addressable, acceptance to this journal in my view relies on better addressing existing literature (comparison to Wright et al's approach especially); if the authors can show AT has sufficient advantage over calibration-based approaches and make sufficient novelty claim over what Wright et al did, that would really help their acceptance case in my opinion. Happy to give more feedback after the authors address the above major points.

Response:

Dear Reviewer #4,

Thank you for your interest in our work. We have now fully revised our manuscript to clarify the motivation and back the claims with more experimental and simulation results where appropriate. We hope the revised manuscript and the following response answers your questions.

“As far as the results are concerned, I think the results for AT are solid and most graphs / eqs are fairly clear but I am still unclear on many things. What do "direct BP" results mean. "We see a significant performance degradation with BP when the physical information is not available." What physical information? What is an "error-free digital reference"? Unclear on what "calibrate and transfer" entails; why are the results so poor for the simulated results? I'd like more details on how the calibration is done so others can reproduce; I don't understand why the calibration resulted in such poor performance.”

We apologize that the terminologies appeared to be confusing in the first version. We have fully revised the terminologies for a more clarified explanation.

"direct BP": This term was originally intended to refer to the process where we train the network within a digital model and then directly transfer the parameters to the physical system. We have decided to change this term to the more commonly used “in-silico BP” to avoid confusion for the readers.

"We see a significant performance degradation with BP when the physical information is not available." What physical information?

Previously, the term “physical information” was intended to refer to the lack of knowledge for how the applied control affects the resulting transformation in the physical processing module (PPM). The lack of knowledge of control-transformation mapping is common for uncalibrated PNN training. However, AT takes a step further: unlike previous methods such as physics-aware training (PAT) which requires the neuron information in every hidden layer, AT only requires the neuron information at the output layer. We have now clarified the terminologies and explained how AT achieves training with less neuron output information compared to PAT in the Results section.

Here is an example where we made the modification to clarify what is meant by the lack of information about the control-transformation mapping:

“Foreseeably, due to the significant mismatch between the estimation profile and the actual transmission controls (equivalently, the mismatch between the expected transformation $f_m(\mathbf{W})$ and the actual physical transformation $f_p(\mathbf{W})$), in-silico BP training fails to classify the species.”

What is an "error-free digital reference"? The term “error-free digital reference” was intended to describe the performance of an ideal BP (where the training and testing are solely conducted in the digital domain without any error). This serves as a benchmark for indicating the level of maximum achievable performance for BP-based methods with the given network structure. We have now changed the term to “ideal BP” to avoid confusion.

Here is an example where we made the modification:

“The ideal BP performance (defined as the maximum achievable accuracy when the same network structure is trained with BP solely in the digital domain without error) ...”

Unclear on what "calibrate and transfer" entails: This term was originally intended to refer to a process where we transfer the in-silico BP-trained parameters with a very rough calibration (which is only an element-wise calibration rather than a system-level calibration). We realized that this comparison seems to be confusing and no longer necessary with the new results added. Therefore, we have removed this benchmark from the article to avoid

confusion. On top of the comparison with the “in-silico BP”, we added a new experiment result of “pseudo IP-BP”, comparing AT’s performance to alternative BP-based methods such as physics-aware training (PAT) or hybrid training (HT). Please see Fig.3 (a) for the new results.

why are the results so poor for the simulated results?

The “direct BP” results in the previous draft refers to “in-silico BP” for a PNN structure with errors and lack of knowledge for control-transformation relation. For the simulation, we added the level of physical error seen in the experiments. Consequently, in-silico BP, which we know is sensitive to error, sees significant degradation of performance. On the other hand, AT can still achieve very good results near the performance of an ideal BP.

I'd like more details on how the calibration is done so others can reproduce; I don't understand why the calibration resulted in such poor performance.

We have removed the benchmark of a “rough” calibration-and-transfer to avoid confusion for the readers. Originally, the “rough” calibration process was defined as the “elementwise” calibration where we only consider the transmission behavior of each individual Mach-Zehnder interferometer (MZI) cell. However, as we know that there are fabrication variations for photonic chips. The transmission behaviors of the cells and channels are not identical. For example, the maximum power from different channels of the chip can be different due to the non-identical insertion loss at each path. The non-ideal edge coupling also means the signal input are slightly different. This is why the calibration for the analogue computation always needs to be conducted at system-level (considering the relation between different channels and unit cells) rather than as elementwise. After careful thought, we decided to remove this analysis because it can be confusing and redundant after we have added more significant experiment results to support the improved performance of AT.

I have a few higher level issues with some of the attempts to compare backpropagation based approaches in the paper. Mainly, this paper invokes a lot of the same motivations as "Deep physical neural networks trained with backpropagation" (Wright et al, Nature, 2022) but appears to address these problems in a different manner through alignment. I think a closer comparison of these methods is warranted both in simulation and in experiment because of the similarities, i.e. backpropagating through a "simulated model" of the system. What are the advantages / disadvantages and what's different here (the results in the Wright paper seemed quite good...)? Is there a relationship between Wright et al's method to any of the "BP" approaches discussed in this paper?

The key difference between asymmetrical training (AT) and physics-aware training (PAT) originates from the motivation for creating AT. Despite some BP-based PNN training methods such as PAT exist, the key limitation of these method is the requirement to know the physical neuron information at each hidden layer to construct a differentiable digital twin model for performing backpropagation (BP). However, this necessarily means two things: 1. The propagation of the analogue signal must be truncated at each layer. 2. The need to access the neuron information creates excessive readouts and conversions between the analogue-digital interfaces.

The need to truncate the physical information propagation in methods such as PAT inevitably leads to slower processing, compromising the initiative of PNN implementations for fast processing. On the other hand, AT preserves the completeness of the physical information propagation within the analogue domain until the output layer, allowing for fast in-propagation processing for the entire deep network structure to be encapsulated within the analogue domain.

Furthermore, the speed of methods such as PAT is bottlenecked by the interface speed between the analogue and digital domain. For existing BP-based methods such as PAT, the need to readout the intermediate hidden neuron scales with $O(M)$, where M represents the total number of neurons in the system excluding the output layer. This means that by extracting the physical information, and then convert the information back into the analogue domain requires at least $O(2M - P)$ analogue-digital interfaces. In comparison, AT only needs to access neuron information at the output layer, scaling as $O(P)$ with P output neurons. We know that for any sensible deep network structure, $M \geq 2P$. Therefore, for any deep network structure, we can see a significant improvement in speed for AT-compatible encapsulated systems compared to PAT-compatible systems.

Also, PAT can't train with solely the output layer neuron information; this can be seen with the newly added experimental demonstration in Fig.3 (a). The pseudo intermediate-access physics-aware backpropagation (IP-BP) performance indicates the training behavior when PAT only has access to the output layer information and not the hidden layer information. The constructed differentiable model of PAT is no longer an accurate description of the actual physical transformation, leading to degradation of performance.

There are also practical advantages for using AT in realistic systems compared to PAT. PAT relies on the fidelity of the analogue information acquired at each hidden layer. However, we know that for realistic physical systems, there are external effects which can affect the system state and the signal fidelity. For example, when the physical system state experiences a sudden unexpected strong perturbation, the twin model in PAT is no longer a suitable description for the physical system. This can lead to failure to train with a diverging loss. Similarly, when the system is sensitive to the operational environment (such as temperature fluctuations), the signal fidelity can be affected, and the system state might experience slight fluctuations. This can also lead to failure of training with PAT. On the other hand, the additional forward digital pass in AT is always being trained by an ideal BP. The digital twin in AT is not affected by the variation in the physical domain, protecting the update at each time instance. These scenarios are studied in simulation and shown in Fig.5 (g) and (h) of the revised paper.

As briefly mentioned above, AT allows faster, and more energy-efficient training compared to PAT due to the continuous propagation of analogue signal and bypassed analogue-digital interfaces. Even when considering the additional forward pass required in the digital domain, we can see a significant improvement in efficiency for different network sizes. We duplicate the results from Fig.6 here for easier viewing:

We have now fully revised manuscript to better illustrate the difference between AT and PAT, and further highlight the motivation for utilizing AT. Please see the new version of the paper for details. We would like to highlight a few places where we made the modifications:

Introduction (P1, 2): “Backpropagation¹³ (BP) finds the gradient update to the loss function for the network parameters, offering fast training time step convergence¹⁴. However, existing BP-based PNN training methods such as physics-aware training¹⁵ (PAT), in-situ BP¹⁶, dual adaptive training¹⁷ (DAT), and hybrid training¹⁸ (HT), relies on knowing the net output and activation level information for hidden layer neurons in a deep network structure. This means that the propagation of the analogue signal needs to be truncated and readout at each layer. The truncation of the physical information propagation undermines the advantage of ultra-fast in-propagation processing with PNNs¹⁹ (See Fig.1 (a)). Furthermore, the need to shuttle the information back-and-forth across the analogue-digital (AD) interface creates excessive analogue-to-digital conversions (ADCs) and digital-to-analogue conversions (DACs). ADC and DAC are known to be speed bottlenecks^{20,21}, where the AD interface speed²² is much slower than solely in-analogue¹⁹ or in-digital⁴ computation (see Fig.1 (b)). These factors lead to overheads of increased training time, additional energy consumption, and increased fabrication cost, compromising PNNs’ goals for fast, energy-efficient and cheap processing²³.”

Results section (P3-8): Entire subsection of “Asymmetrical training method”, “AT method for training fully encapsulated PhPNNs”, and “AT for training deep PNN with only output neuron information”

Results section, subsection of “Generalization and repeatability of the AT method” (P9-11): Fig.5 (g) and (h), and last paragraph of the subsection: “Existing IP-BP PNN training methods rely on the accuracy of neuron information acquired at the hidden layers ... the update at each time step is protected by the overall AT estimator, thus not affected by the noisy system.”

Discussion section (P11-13): Entire section and Fig.6.

“Additionally, what makes calibration inferior (this is made fairly clear in the intro)? I would like some very specific calculations of the memory and time cost of calibration. Calibration is very useful for debugging and probing physical systems (the paper dismisses this using a somewhat vague analogy to Ising machines / reservoir computing, when MZI networks are really a fairly specific system). Most PNNs are made of MZI meshes which are readily calibrateable, and such procedures have been successfully demonstrated at fairly large scales in the commercial sector (e.g. Lightmatter in the past). So why revert to a procedure like this for such systems? What benefit would it provide (and to what extent?) compared to the costs in convenience of directly flashing NN weights and especially less effective debugging of the overall system?”

The reason for using a “calibration-free” training in AT is because we wanted to demonstrate training for fully encapsulated networks. As we have briefly discussed in the previous part of the response, constructing the deep PNN structure as an encapsulation helps preserve the fast in-propagation processing for the entire structure rather than for each individual layer. The readout only at output layer neurons also allow the training to be more time and energy efficient. To illustrate AT’s ability for training fully encapsulated deep PNN, we restrict ourselves to not accessing the hidden neuron information at any stage of training.

When we are considering an encapsulated transformation of the deep PNN as $f_{p1} \left(f_{p2} \left(f_{p3} (\dots) \right) \right)$, the calibration process is non-trivial. If the individual transformations are linear, then we could separate the effect from each layer with only the output layer neuron information. However, when these transformations are non-linear, the separation of effect becomes complicated as we can’t tell where exactly the error comes from for an encapsulated system. Thus, the detailed calibration of such a deep network implies the requirement to access neuron information at the hidden layers. This is not something we want for a fully encapsulated deep network structure. Therefore, we are not against

the process of calibration by any means, but rather that we wanted to pervert the rigorousness of demonstrating AT for fully encapsulated networks without access to the hidden neuron information during the training.

Therefore, the “calibration-free” training of AT shouldn’t be viewed as the sole purpose of AT. Instead, its error-tolerance is only one of the advantages, alongside the key motivation to allow training for encapsulated networks and improving training time and energy efficiency from existing methods.

As you have mentioned, employing calibration-free training can bring very practical benefits. If we assume a finite analogue system control with the resolution of R , and the number of control parameters to grow $\propto M^2$ (where M is the neuron size of the PNN). Then the total memory required for a calibrated training scale with $O(RM^2)$. In comparison, the memory for calibration-free training only scales with $O(R)$.

Also, another way to view this is: for the larger-sized systems from the commercial sectors, even when those systems are calibratable, there will always be some inaccuracy in the calibration or change of the system state due to external factors (such as temperature change for integrated photonics). The inaccuracies can have significant effects on analogue computation (which typically lacks a cross-checking mechanism). If we were to encapsulate those systems for fast training and processing, the accumulation of slight errors can lead to significant performance degradation as the information needs to be passed many times through the devices. Therefore, a pure in-silico calibrate-and-transfer process wouldn’t be sufficient for those scenarios. Instead, AT can be used in conjunction with some rough calibration of those systems to allow the encapsulation of a deep network structure for much faster processing and training without sacrificing the performance.

To clarify the motivation for demonstrating calibration-free training with AT, we have made the following modifications:

Results section, “AT method for training fully encapsulated PhPNNs” subsection (P5): “Typically, most PNN training involves some calibration prior to the training process. However, this is a non-trivial process for fully encapsulated deep networks. The accurate calibration of the hidden layer connection implies access to the intermediate hidden neuron information and truncating the propagation of the physical information. Consequently, to demonstrate training for fully encapsulated deep PNNs, we don’t access the hidden neuron information at any stage of the training. Instead, we employ calibration-free training through an estimation profile for applying the physical control parameter. Calibration-free training also brings the advantage of reduced memory requirement compared to calibrated training. The total number of control parameters scales roughly as $\propto M^2$. For a finite resolution of R of the analogue system control and neuron size of M , the required memory scales as $O(RM^2)$ for lookup tables in calibrated training. In comparison, the calibration-free training offers a reduced memory requirement of $O(R)$.”

“Surely there are other systems to experimentally demonstrate AT than a PNN; and if that was the intent of the paper, there is currently no compelling argument for the superiority of this method over the accelerated training method proposed by Bandyopadhyay et al. which is actually model free and doesn't require calibration and is much more energy efficient (since you don't have to backprop on a digital machine). Is it convergence time?”

Thank you for mentioning the paper “Single chip photonic deep neural network with accelerated training” by Bandyopadhyay et al. While we do highly respect the work they have done, we think there are two reasons that our method can be preferable over their training algorithm: **1.** Faster convergence of AT **2.** The lower energy to operate the entire PNN system with AT for large datasets

1. The faster convergence and faster training time of AT

The methods that Bandyopadhyay et al. proposed involve a process of perturbing the parameters slightly in one direction, then the system is perturbed in the opposite direction. The overall effect is averaged out to find an

estimation to the update to the physical system. The foundation of the method is closer to concepts of finite-element stochastic process rather than gradient based methods. This means that the convergence is longer compared to gradient based methods.

We would like to invoke the newly added Table 1 in the main article, which summarize some key features of relevant methods:

Table 1: Comparison of relevant PNN training methods								
Method	No. of physical propagation	Digital model?	Need calibration?	Intermediate hidden layer readout?	Information access points	Readout time steps	Expected training convergence	Ref.
BP-based methods								
AT	1	Yes	No	No	P	1	$\approx O(T_0)$	This work
PAT (HT)	1	Yes	No	Yes	M	$N + 1$	$\approx O(T_0)$	20,23
In-situ BP	3	No	No	Yes	M	$N + 1$	$\approx O(T_0)$	21,31
DAT	1	Yes	Yes	Yes	M	$N + 1$	$\approx O(T_0)$	22
Non-BP-based methods								
DFA	1	No	Yes	No	P	1	$> O(T_0)$	32,33
FF	2	No	No	No	$2P$	2	$> O(T_0)$	16
Finite stochastic	1	No	No	Yes	M	$N + 1$	$\gg O(T_0)$	34,35
Gradient free	1	No	No	No	P	1	$\gg O(T_0)$	17,18

On the other hand, AT is a BP-based method, as can be seen from the expression of the AT estimator:

$$\Delta \mathbf{W}_{\text{AT}}^{[l]} = M_{w1} \cdot \Delta \mathbf{W}_{\text{dig}}^{[l]} + M_{w2} \cdot \Delta \mathbf{W}_{\text{pseudo}}^{[l]} \quad (3)$$

Due to the unity constraint of the two mixing ratios, $M_{w1} + M_{w2} = 1$, the AT estimator converges towards a pure BP update when the separation between the digital and physical transformation is small (towards the end of the training). The fixed M_{w1} and M_{w2} also allows the training to be self-modulated by the magnitudes of the terms (see Supplementary). Consequently, the overall update expression is regulated by the digital BP behavior, providing AT with a convergence time similar to an ideal BP.

This means that AT requires less training epochs than the stochastic method proposed by Bandyopadhyay et al. Furthermore, for the method proposed by Bandyopadhyay et al., each sample needs to be passed through the network with the two settings of $(W + \Delta)$ and $(W - \Delta)$ for an update to be computed.

This means that for a total sample size of S , there needs to be $2S$ physical propagations of samples for each epoch. In comparison, AT only requires S , allowing for faster training during every epoch. This together with the fewer epochs required to train for AT, the training time is significantly shorter than the method proposed by Bandyopadhyay et al. Therefore, from a training time efficiency perspective, it is preferable to use AT.

2. The lower energy to operate the entire PNN system with AT for larger networks and datasets

While it is true that the method proposed by Bandyopadhyay et al. doesn't require the energy for computing the backpropagation, we must be very careful when considering the total energy required for training a PNN system. All PNN systems are hybrid analogue-digital systems (even the ones that don't require a digital model), meaning that we must be aware of the overheads in the analogue domain, the digital domain, and at the analogue-digital interface.

We would like to start by invoking the discussion section in the main article for defining the time and energy required for training:

“We define the minimal extraction time, T_{extract} , as the sum of the hardware propagation time T_{prop} and the readout time at the analogue-digital interface T_{ADC} . On top of the extraction time, we add the computation time T_{comp} required

to compute the update based on the physical information. The training times for AT and IP-BP methods are defined as Eq. (5).

$$T_{AT} = \max[T_{\text{extract-AT}}, T_{\text{paral}}] + T_{\text{comp-AT}}; \quad T_{IP-BP} = T_{\text{extract-IPBP}} + T_{\text{comp-BP}} \quad (5)$$

T_{paral} is the forward propagation (computation) time of the digital parallel model, T_{comp} is the time required to compute the updates. The $\max[\dots]$ is the maximum argument operator for returning the higher argument. The neuron information extraction stage must be completed before updates are computed, thus forming a sequential process (summation of time). On the other hand, the two forward propagations in AT are simultaneous process, and only the process with the longer time need to be considered.”

And the energy:

“The faster training time of AT also allows for more energy-efficient PNN training. The energy to operate the hybrid PNN system for the two training methods is shown in Eq. (6).

$$\begin{aligned} E_{AT} &= P_{AT\text{-phy}} \cdot T_{\text{extract-AT}} + E_{\text{comp-AT}} + E_{\text{interface-AT}} \\ E_{IP-BP} &= P_{IPBP\text{-phy}} \cdot T_{\text{extract-IPBP}} + E_{\text{comp-BP}} + E_{\text{interface-IPBP}} \end{aligned} \quad (6)$$

The energy to operate the physical system is directly related to the extraction time. The power of the physical system must be maintained before extracting sufficient physical neuron information to compute the updates. Adding to the physical module’s energy, we add the computation energy (E_{comp}) in the digital domain and the interface energy consumption ($E_{\text{interface}}$). Access to the hidden neuron information implies the requirement for dedicated physical components to divide the signal, meaning that the signal (or amplification) to maintain the same fidelity in IP-BP requires a higher power than AT method. Like the training time analysis, we can estimate the time required to train for these systems based on the SOTA values (See Supplementary for sources of values).”

When looking at the detail of energy consumption for an integrated photonic-based PNN with the SOTA results of components (see Supplementary), we can realize that the power required for tuning of the phase shifters on PIC is a limiting factor for larger scaled systems. The stochastic method by Bandyopadhyay et al. requires two times the physical propagation of $2S$ for each epoch, while AT only requires S . This means that the extraction time of the stochastic method is twice that of AT. The compatible physical systems with AT and the stochastic method should have the same power requirement. Consequently, the energy consumption for operating the physical domain system in the stochastic method by Bandyopadhyay et al. always would require more energy than AT due to the longer extraction time.

Due to the physical domain being the limiting factor in power consumption at larger network sizes, for datasets with many samples S , the increased physical propagations required in the stochastic method might end up consuming more energy than computation of the backpropagation. This together with the fact that the convergence of the stochastic method is slower, requiring more iterations to train. Therefore, from the perspective of the entire hybrid PNN system, AT can be more energy efficient for larger networks and larger datasets.

To illustrate the time and energy efficiency of AT, we have made the following modifications:

Introduction section (P1, 2): Paragraph 2, 3 and Table 1

Discussion section (P11-13): Entire discussion section and Fig.6.

Message to editor and all reviewers:

We would like to thank the editor and all reviewers for their precious time spent in relation to our work. We are glad to hear that Reviewers #1 and #4 are happy with the changes we made to the manuscript. We understand that Reviewers #2 and #3 still have concerns about the manuscript. We hope the newly revised manuscript now addresses your concerns. The changes made in the main article are tracked as: revised content. For quoting text from the original manuscript to the response letter here, we use red text: quoted content.

The reviewers can find the specific response to their comments on the following pages:

Reviewer #1: Page 1

Reviewer #2: Page 1-36

Reviewer #3: Page 37-46

Reviewer #4: Page 46

The text color above is for specifying the comments from each reviewer.

Reviewer #1 (Remarks to the Author):

Thank you for your detailed explanations. I believe the manuscript has improved significantly. My concerns are now resolved.

Dear Reviewer #1,

Thank you for your precious time to review our revised manuscript. We are very glad to hear that you found the manuscript improved and our revision addresses your concerns.

Response to Reviewer #2:

Reviewer #2 (Remarks to the Author):

Thanks to the authors for their response to my comments. The main issue, as raised by all reviewers in the previous round, was the comparison to prior work. While I find AT to be interesting, I have multiple concerns about the author's new comparisons to prior training experiments.

1. I find the new claims in energy and time advantage for training are not supported. This is included in the abstract: "For any deep network structure, AT offers significantly improved time and energy efficiency compared to existing BP-PNN methods, and scales well for large network sizes." Unfortunately, their calculations are based on flawed assumptions and the algorithmic scaling claim, i.e. does it scale as well as BP as network size increases, is also not proven in the paper. The authors argue that near the end of the training, AT's update will converge to an ideal BP update, but this is only true at the end and not for most of the training runtime.

Moreover, the authors do not accurately consider the power consumption and speed of the digital system in their calculations. In the supplementary, they compute the energy and time of the digital system for training, arguing that an A/D conversion interface is much slower than running a model on the digital system. This is based on two assumptions:

- They estimate an ADC throughput of 60 Mbps. I'm not sure where this comes from, but it is clearly incorrect – commercial ADCs are available at 10s of Gbps and are used in the optical communications industry. For instance, see Murmann's survey at <https://github.com/bmurmman/ADC-survey>. It looks like they assume the bottleneck is SPI, but faster communication protocols such as JESD204B exist for precisely this application. Thus the estimate here is orders of magnitude off.

- They claim that a state of the art GPU (Nvidia A100) has a computation speed of 624 teraFLOPs, which is why the speed of the digital system used for executing the model in their algorithm is negligible. The authors are confusing throughput and latency here – a digital system can have high throughput, which is achieved through pipelining, amortizing memory access, and reducing bit precision, but that does not equate to the latency. For example, Nvidia's own numbers on the MLPerf benchmarks indicate about ms latencies for inference (probably larger for training): <https://developer.nvidia.com/deep-learning-performance-training-inference/ai-inference>.

In Figure 6 of the main text, the authors now introduce a new plot showing significant speed up over other methods that rely on digital systems for the backward pass. It is hard to evaluate this plot, as in addition to the flawed assumptions I mention, there is not enough information on the exact parameter values used or for what model this is being calculated. The GPU is estimated in their calculation to be 500 fJ/FLOP. This is not correct (624 TOPS number quoted by Nvidia is for INT8). The total energy per inference will depend on the model size (i.e. number of FLOPs and bit resolution) but it is not clear in the calculation what is assumed here. It would help if the calculation were explained, step-by-step, in the supplementary.

In summary: 1) it is not likely that A/D is the bottleneck compared to having to run the model on a digital system; 2) the digital system energy and latency is likely greatly underestimated, and I think it is likely the GPU will dominate these quantities, not any A/D interface; 3) the algorithmic scaling is not studied, and the authors claim without any simulation or experiment that it is equivalent to backprop; and 4) not enough information is provided on these calculations to evaluate the claimed advantage over other methods.

Dear Reviewer #2,

Thank you for the very detailed feedback on the manuscript. We have carefully looked into your comments and made modifications accordingly. We have now completely revised the manuscript to provide more rigorous discussion of the AT method. New discussions have also been added where appropriate to provide the readers with a more comprehensive view on the topic.

For a clearer presentation of the work, we have made two major changes in the Discussion section for the comparison between AT and other PNN training methods:

To start with, we re-scoped the efficiency comparison to a more general logical argument rather than the specific quotes for the combination of the equipment. As you have commented, the exact performance of the equipment used can vary for different combinations of setups used. Thus, to avoid confusion caused by the analysis based on a very specific equipment configuration, we now utilized a purely logical discussion about how the computational time

required for the truncated DPNNs always grows at a faster rate than the propagation time scaling of the photonic hardware. This argument is now presented together with the discussion on how the computational overhead can be distributed for multiple copies of the PNN system.

A second major revision is that we removed the table previously in the Introduction section. Different types of training methods all have their own unique advantages and suitability to the exact PNN construction. We believe there is no single category of training methods that should be regarded as the elixir for all tasks and PhyNN constructions. Comparing across categories of methods which are purposed for different PNN construction strategies can be confusing for the readers. Instead, it indeed provides more clarity to describe what category of training method is more appropriate for different applicational scenarios. For example, for a PNN construction with inherently isomorphic model description, BP-based methods are clearly very suitable and well-balanced. Consequently, we revised the comparison discussion to be centered around the BP-based methods as this is the only sensible direct comparison. We also added a new discussion in the Supplementary to clearly explain this, so that the readers can better understand the whole picture of PNN codesign. We hope these two points in combination now present the work more clearly to the readers without confusion.

We thank you for bringing up the faster protocols such as JESD204B. Whilst we have now changed the discussion to a purely logical analysis regarding the access time step and access points between the existing BP-based methods and AT, we still wanted to show that the computational time advantage of employing an encapsulated DPNN is significant over truncated DPNNs (for sample information propagation during both training and inference). Thus, the direct comparison for the operation speed (one input of sample) for extraction is important. We added a discussion in the Supplementary file showing that the A-D and I/O bottleneck time grows at a much faster rate compared to the propagation time growth of the PNN photonic device structure (even with the faster protocols such as JESD204B). This discussion backs the motivation for constructing the encapsulated PNN. Also, thanks for mentioning the resources such as the A/D conversion survey and the GPU performance, for which we have included in the reference of the Supplementary to direct the readers who have further interest in these topics.

To your specific questions you summarized: 1) As discussed, for the comparison between BP-based methods, the difference between an encapsulated and truncated PNN is significant. The truncation of the network structure is something that constantly affects the PNN operation, even for the post-training inference operations (which arguably would last longer for actual applications). Therefore, on top of the specific drawbacks with the extensive A-D interface, the more important message we want to get across is the advantages of constructing the encapsulated DPNNs. From our revised Discussion, even through a purely logical aspect of access points, the co-constructed truncated PNN is clearly a disadvantage for fast propagation platforms such as photonics.

2) Similar to the last point, the main point that we tried to bring with the argument of the A/D interface is the inability to construct the encapsulation of DPNN when intermediate states are required. Thus, the disadvantage of IP-BP methods is not only limited to the energy at interfaces, but also the computation latency (on the photonic side) due to the frequent data conversion and shuttling in a deep network structure (for both the training stage and the inference afterwards). The truncation of the signal propagation is particularly undesirable for platforms such as photonics as it potentially compromises the advantages of the analogue photonic implementations. In addition, in the revised Discussion, we also talked about how the overhead associated with the parallel model in AT can be distributed and reduced for more efficient overall operation.

3) Regarding algorithmic scaling and the reason why we call it a BP-based method, the key thing to note is how the methodology behind AT is different from other alignment-inspired methods (DFA and PAT). Unlike DFA, which utilizes the concept of alignment at its limit (i.e. away from the 90° , with the multiplication to the randomized fixed matrices set), PAT and AT both utilize the concept of alignment as a relaxation to the strict gradient condition (i.e. starting from 0°), with some feedback from the physical system (the intermediate readouts with PAT and the output layer with AT). For PAT, the relaxed condition of alignment is between the device's physical gradient δW_{phy} and the

PAT update ΔW_{PAT} (since without the exact characterization, the update direction in PAT is an estimator of the actual gradient). How AT utilizes the idea of alignment is different to PAT; instead of trying to find the physical gradient, AT has a complete forward-backward set in the parallel model. The parallel model, trained by BP, is known to have a loss-converging update at δW_{dig} . The goal of AT is to align the photonic transformation subject to the AT update direction ΔW_{AT} towards the digital transformation subject to the parallel update δW_{dig} . Consequently, two aspects can be noted: 1. The update dynamic of AT is the layer-by-layer BP update dynamic. 2. The update at each time-instance is regulated by the parallel model's BP behavior. With the $M_{w1} = M_{w2} = 0.5$ weighting, the overall update forms a self-regulating process (in terms of magnitude and direction) subject to the condition of the parallel BP update at $t = t'$. This is the reason why we call AT to be a BP-based method. We have included a new discussion in the Supplementary file (S13) to discuss this.

4) As discussed above, we have changed the approach to formulate the discussion. For the comparison to IP-BP methods, the best approach is to present the work through purely logical comparison between the number of access points and access timesteps. This allows the discussion to be logically applicable and unlimited by the specifications of a certain equipment combination. We also added discussion about how the computational overhead (of the additional forward pass) in the AT method can be distributed across the PNN device copies with its tolerance to variations.

We have made the modifications to the following parts of the manuscript related to this part of the response:

Abstract: The whole abstract

“Photonic neural networks (PNNs) are fast in-propagation and high bandwidth paradigms that aim to popularize reproducible NN acceleration with higher efficiency and lower cost. However, the training of PNN is known to be a challenge, where the device-to-device and system-to-system variations create imperfect knowledge of the PNN. Despite backpropagation (BP)-based training algorithms often being the industry standard for their robustness, generality, and fast gradient convergence for digital training, existing PNN-BP methods rely heavily on the accurate intermediate state extraction for a deep PNN (DPNN). These information accesses truncate the photonic signal propagation, bottlenecking DPNN's operation speed and increasing the system construction cost. Here, we introduce the asymmetrical training (AT) method, tailored for encapsulated DPNNs, where the signal is preserved in the analogue photonic domain for the entire structure. AT's minimum information readout for training bypasses analogue-digital interfaces wherever possible for fast operation and minimum system footprint. AT's error tolerance and generality aim to promote PNN acceleration in a widened operational scenario despite the fabrication variations and imperfect controls. We demonstrated AT for encapsulated DPNN with integrated photonic chips, repeatably enhancing the performance from in-silico BP for different network structures and datasets.”

Introduction section: Most of the introduction section

“Neural network (NN)-based machine learning algorithms are prevalently used in the state-of-the-art (SOTA) industry and academic research¹⁻³. The surging need for further acceleration of NN-based applications calls for emerging paradigms such as physical neural networks (PhyNNs), offering high-bandwidth in-propagation analogue computation^{4,5} on top of the increased parallelism of dedicated digital hardware such as GPU⁶ or TPU⁷.

Photonic-based analogue neural networks (PNNs) belong to the subclass of scalable isomorphic PhyNNs^{5,8}, which enjoy the complexity of PhyNN while maintaining the repeatability and reproducibility to provide acceleration for a wider range of users. Photonics is widely known for its high bandwidth and multiplexing ability in multiple domains⁹⁻¹². The matrix-vector multiplication accelerator with both on-chip^{13,14} and free-space^{15,16} optics already reveals great potential for photonic processing^{17,18} in general. Demonstrations of integrated PNNs have shown superior performance in both processing speed¹⁹ and energy efficiency⁸. These miniaturized integrated PNN systems hold great potential due to their fabrication reproducibility and compactness. Nonetheless, unlike pure digital deep neural networks (DNNs), where the network construction is a general mathematical description, PNN implementations are codesigning processes between the task, the PNN device platform, the application environment, and the training algorithm. The

difficulty of training PNN due to the device-to-device and system-to-system variations often means a complex and demanding codesign process.

Backpropagation²⁰ (BP) finds the gradient update to the loss function for the network parameters, offering fast training gradient convergence²¹. BP’s advantage for fast training with DNNs comes from two key aspects: BP’s root in gradient-based update logic and BP’s layer-by-layer update dynamics. BP’s robustness, generality and fast convergence make it a well-balanced industry standard approach for most digital DNN-based tasks. However, standard BP imposes requirements of highly accurate model control and knowledge. The inevitable fabrication variation inherently creates imperfect knowledge for the users. In addition, the lack of perfect control-transformation mapping on the typically noisy and lossy analogue computation platforms^{22,23} can’t satiate the high requirements of standard BP. The mismatch between the user’s interpretation and the PNN’s actual state can lead to failed training or dramatic performance degradation^{24,25}. Therefore, some non-gradient-based^{26–31} or model-free/stochastic methods^{32–35} are proposed to bypass the requirement for formulating a model description. Nonetheless, the natural trade-off with non-gradient methods is the higher difficulty to train complex tasks and slower convergence compared to the BP-based methods. When defining the convergence of standard BP to be $O(T_0)$, the convergences of other non-gradient PNN training methods are typically $> O(T_0)$ ³⁶. PNNs, being mostly operation-centered platforms, are highly compatible with BP-based gradient methods for the typically exhibited isomorphism. (See Supplementary for more discussion on the categorization of PhyNNs.)

Many efforts have been made to search for PNN-tailored BP-based methods. The general attempt of the BP-PNN methods is to find an estimator of the device’s physical parameter gradient. The approaches which utilize a separate digital model or propagation pass for estimating the gradient update include physics-aware training³⁷ (PAT), hybrid training³⁸ (HT), and dual adaptive training³⁹ (DAT). There are also approaches which utilize the physically reversed signal input through the structure to obtain in-situ computation of the gradient^{40–42}. However, the common limitation of these existing BP-PNN methods is the heavy reliance on accurate intermediate layer state extraction in a deep PNN (DPNN). We call these methods intermediate-access physics-aware backpropagation (IP-BP) methods. Due to the intermediate access need, IP-BP codesigned PNNs are truncated, in which the computation acceleration is only available within a hidden layer’s structure. For fast propagation platforms like photonics, the overall operation is bottlenecked by the analogue-digital (AD) conversion interfaces^{43,44} and data shuttling⁴⁵, creating a delay growing with the network complexity. This potentially compromises the incentive of employing photonics for fast and low-delay processing. Instead, it is more desirable to construct encapsulated DPNNs (encapsulation refers to systems whose input signal is maintained within the optical analogue domain without intermediate extraction), allowing the advantage of fast processing to be enjoyed for the entire deep network structure. The number of AD access in a truncated DPNN grows as $O(2M - P)$ (M is the total number of neurons excluding the input layer, and P is the number of output layer neurons, $M > P$ for any DNN), whereas it scales as $O(P)$ for encapsulated DPNN. The access timestep is also reduced from $N + 1$ (N is the number of hidden layers) in a truncated network to 1 for an encapsulated network. (See Discussion section.)

Consequently, it is desirable to find methods that bypass the IP-BP methods’ access limitations while still enjoying BP’s general fast convergence behavior. Here, we present the asymmetrical training (AT) method for well-balanced training of DPNN systems with single-structured design (doesn’t require special structure for training, needs the minimum photonic component requirement same as the inference for reduced cost), encapsulated computation (the signal is maintained within the optical domain for the entire DPNN structure), and error-tolerance (tolerant to PNN’s device-to-device and system-to-system variations). AT utilizes an additional forward pass in the digital parallel model compared to the existing estimator approaches^{37–39} to eliminate the requirement for accessing intermediate DPNN state information (for total access point of P). AT’s gradient-like layer-specific update dynamic is compatible with standard update optimizers such as gradient descent^{46,47} or Adam⁴⁸. AT’s goal to increase training efficiency and reduce cost is in unison with PNN’s general purpose for faster and cheaper computing⁹.

We demonstrate the AT method with encapsulated DPNN utilizing photonic integrated circuits (PICs) as the physical processing module (PPM). In this paper, we start by explaining the concept and working principle of the AT algorithm. We then experimentally demonstrated AT for a fully encapsulated DPNN for classification. We further experimentally validated AT’s ability to achieve training with only output neuron information for different tasks (Iris-flower classification and modified hand-written digit classification) and scaled-up structures through repetitive use of PPMs free of local characterization. We analyzed the scalability and repeatability of AT with more complex datasets such as digit-MNIST⁴⁹ (95.8%), fashion-MNIST (FMNIST⁵⁰, 87.5%) and Kuzushiji-MNIST⁵¹ (KMNIST, 85.6%) through

simulation under experimental level error, showcasing the repeating enhanced performance from in-silico BP. The results achieved are close to the benchmarks of ideal error-free BP training. We also systematically investigated AT’s performance for varying network structures and showed its high error tolerance. We finish by discussing the advantages of constructing encapsulated DPNN with AT, alongside a discussion of the applicational scenarios of AT and how most of the computational overheads of AT can be bypassed for overall efficient PNN training and operation.”

Discussion section: Entire discussion section

“Efficiency of AT-compatible encapsulated structures

While PNNs can offer unique NN acceleration strategies that are not possible on traditional digital NNs, they also face realistic challenges since the codesign (construction) process can be limited by the demanding training operations. Here, we discuss the efficiency that can be enabled by constructing an encapsulated DPNN with AT in comparison to the truncated PNN of existing IP-BP methods.

The encapsulation of DPNN bypasses most of the AD interfaces, significantly reducing the signal conversion and data shuttling time for the sample propagation (during both training and inference). For each propagation of information through the PNN structure, we can define the extraction time, T_{extract} , as the minimum time required to obtain an output for further operation (obtaining a prediction or computing update). A simple formulation of the extraction time is the combination of the propagation time T_{prop} with the readout/conversion time $T_{\text{interface}}$ at the AD interface. The propagation time for PNN is much shorter than the interface and readout time $T_{\text{prop}} \ll T_{\text{interface}}$ (see Supplementary). Consequently, we could approximate the extraction time scaling as $O(P)$ for encapsulated systems and $O(2M - P)$ for truncated systems (each AD interface corresponds to a pair of readout-and-write operations, thus the factor of 2). For DPNNs, $P < (2M - P)$ always holds true. The extraction time is directly related to how fast the prediction is made, and thus, the operation speed of the PNN. T_{extract} for encapsulated PNN is dependent on the specific task instead of the model complexity. Taking the MNIST 10-class task as example, P stays 10 regardless of the model’s complexity (Fig.6 (a)). Consequently, the encapsulated PNN offers better compatibility overall with increasing model complexity.

Furthermore, the sequential layer-by-layer operation of a truncated DPNN imposes an increased timestep for computation that is not addressable through higher parallelism. For the N hidden layers, a total of $N + 1$ timestep is required for each single forward propagation, while there is always only 1 timestep for encapsulated DPNN (Fig.6 (b)). The constant timestep offers good compatibility with schemes of higher parallelism, such as wavelength division multiplexing.

The extraction time can also be related to the energy consumption of the photonic device during the inference. For a photonic system with power consumption of P_{photonic} , the minimum operation time is the extraction time for obtaining a prediction. Thus, the minimum energy required to operate the photonic system would be $E_{\text{photonic}} = P_{\text{photonic}} \cdot T_{\text{extract}}$. For systems where the limiting power consumption is photonic hardware, the longer the T_{extract} , the larger the energy inefficiencies are created due to the prolonged operation.

Distributed overhead of AT with repeatable application to trainable PNN structures

While PNN training methods utilizing a parallel/twin digital model can provide a more accurate estimate of the physical gradients, the typical concern is the computational overhead that might come with the application of the parallel digital model. Here we discuss how computational overhead (and the associated additional energy consumption) in AT can be avoided to a computational overhead similar to standard BP.

As mentioned in the Results section, the digital update of the parallel model can be computed independent of the local PNN’s error, allowing for the same computational timestep as BP without sequential delay. The key motivation for reproducible PNN platforms such as PICs is often associated with reproducible acceleration despite the device-to-device variations. The same PNN structure might be fabricated for many copies to achieve local acceleration of a specific task. For a total of C copies of PNN devices, the overall description of each PNN system can be formulated

as $f_{p,1;i}(f_{p,2;i}(\dots))$ for $i = 1, 2, \dots, C$, denoting the specific system under discussion. In the case of chip fabrication variation, the overall descriptions of these systems form a statistically relevant set. Thus, we note the equivalence of different estimation profiles and physical device copies. The generality of successful training with different profiles (see Fig.3 (d)) can also be interpreted as different copies of the PNN systems trained equally with a single digital model.

From Eq.3, the overall AT estimator is obtained through two parts: the digital parallel update (ΔW_{dig}) and the pseudo update (ΔW_{pseudo}). For a given task and initialization condition, the update of the digital model is independent of the local PNN device and can be described as $\Delta W_{\text{dig}}^{t=t'}(W_0^{t=t_0}; X, Y, \alpha^{t=t'})$ at time instance $t = t'$. On the other hand, ΔW_{pseudo} is local to the specific PNN device copy (subject to variation) as $\Delta W_{\text{pseudo}}^{t=t'}(f_{p,i}; W_0^{t=t_0}; X, Y, \alpha^{t=t'})$. The generality of a single digital model to different variations of the PNN systems means that the collection of the digital updates can be stored and reapplied to different PNN devices without the need to repeat the process of computing the digital updates (thus no need to repeat the parallel model's backpropagation). For multiple PNN systems, the forward pass in the parallel model only needs to be computed once. Effectively, the computational overhead required to utilize the digital parallel model is distributed across the PNN copies (Fig.6 (c)). Thus, the overhead of operation is significantly reduced for each individual device. When $C \gg 1$, the overall computational overhead associated with each individual PNN device is reduced towards the overhead of a standard BP process. This allows the benefits of encapsulation to be enjoyed while little overhead is added on top of the standard BP process for training of many devices.”

Discussion section, Fig.6:

Supplementary: S10, S11, S12, and S13

“S10. Discussion on the interface speed, propagation speed and propagation time of the AT encapsulated PNN

As described in the main article, we have defined the extraction time T_{extract} to be the minimum time required before sufficient information is available from the PNN structure for useful operation. More specifically, T_{extract} is the summation of the propagation time through the PNN device and the interface time required to readout that piece of information. One of the assumptions we have made in the discussion is that the propagation time is a much shorter time than the access/interface time. Here we discuss further the details of this argument.

For representation of a A node to A node connection, the most general hardware representation would be a fully connected $A \times A$ connection. If we define the hardware depth as the number of repetitive components that the signal must encounter before fully modulated for the $A \times A$ connectivity. We can realize that the minimum scaling to be A as a direct mapping topology. Even with other decomposed topologies⁸⁻¹⁰, the scaling of the photonic hardware path grows linearly with the connectivity size. The propagation time of the in-transformation computation is directly proportional to the hard depth. If we take the SiN platform used in the experiments as an example, the refractive index is around 1.93¹¹. The total on-chip waveguide length of the device is roughly 12.5 mm, corresponding to an on-chip propagation time of 0.08 ns. We can use this as an estimation of the basic propagation time for the hardware component as 0.02 ns per photonic depth.

In comparison, even when considering specialized serial communication protocol of JESD204B¹², with a data transmission rate of 12.5 Gbps. (Here we are considering the data transmission to be the bottleneck of the interface speed. Depending on the exact setup, it is also possible that the ADC sampling rate is the bottleneck.) For the control and readouts of 16-bits, we can expect a roughly 1.28 ns time cost for each access point. The scaling of the access point is described as $O(2M - P)$. Consequently, even when we ignore the internal data shuttling time the scaling of the interface time with respect to the hidden neuron number is over $100\times$ (for the factor of 2) compared to the scaling of the propagation time. Thus, in the main article where relevant, the description of the extraction time scaling for the encapsulated deep PNN is defined as $O(P)$ for simplification. Furthermore, for the other more accessible protocol such as serial SPI communication, the data transmission rate is typically 60-100 Mbps (27 μ s), creating an even larger gap between the propagation time and the interface time. We can see from the above analysis that the construction of an encapsulated DPNN can be time efficient compared to truncated ones.

S11. Energy discussion for systems with distributed computational overhead

As formulated in the Discussion section, for multiple copies of the PNN devices, the computational overhead can be distributed by generally applying the same relevant digital model parameters to a series of PNN systems. For all the BP-based methods, the backwards pass computation for the gradient estimate is necessary. Methods such as PAT and HT all require this computation step. In comparison, the additional digital computation of AT would be the forward pass in the digital model. Here we discuss how the computational overhead of the parallel digital model in AT can be distributed with respect to each local PNN copy for an overall reduced resource requirement that is similar to the standard BP process. The digital computation of a matrix multiplication operation with dimensions of $(n \times p)$ and $(p \times m)$ can be described as $nm(2p - 1)$ ¹³ FLOPs. The number of FLOPs for the backward pass to the forward pass takes a standard ratio of 2:1^{14,15}, where this ratio can be further reduced with the optimization of batch size. Suppose that the energy needed for completing the digital training is $E_{\text{Dig-BP}}$.

In AT, the parallel digital model for a certain network structure and task isn't limited to a specific copy of the PNN device. Instead, it would be a generally applicable model for PNN copies with variations. (Essentially, this means that digital updates acquisition is a standalone process that can be performed prior to the local training of a specific PNN copy.) This means that the more PNN copies are employed, the closer the distributed overhead is compared to standard BP. AT's property is in line with reproducible PNNs' goal for computation acceleration despite the device variation. For example, if there is a total of N PNN copies being manufactured, the actual computational overhead associated

with each of the copy is only $E_{\text{Dig-BP}}/N$. Also, since all the PNN copies here in discussion are constructed with encapsulation, the individual operational and control overhead is reduced, allowing for an overall efficient operation.

S12. Different types of PhyNN training methods and their applicational scenarios

SFig.8 A decision flowchart for choosing the appropriate training method category depending on the applicational scenario.

There are many different categories of PhyNN training methods. Each category of training methods has their own unique properties and advantages. Unlike mathematical (digital) NNs, constructing a PhyNN is often a codesign process between the physical device structure, the task, the applicational scenario, and the training algorithm used. Consequently, it is not sensible to regard a single class of training methods as the elixir for every type of PhyNN construction. Instead, it is more appropriate to understand that each category has its own appropriate application scenarios. Here we present a summarized discussion on how different categories of training methods can be more suitable for different types of PhyNN codesign (implementation strategy).

SFig.8 shows a decision flowchart for choosing suitable methods for a specific PhyNN implementation technique. The construction/codesign of PhyNNs can be broadly divided into structure-centered ones and operation-centered ones. Structure-centered PhyNNs are typically the ones where the codesign logic is based on the attempt to map a device with high complexity to certain specific tasks. One great example of the structure-centered PhyNN is reservoir computing, where a device with high complexity (randomized spatial and temporal connectivity) is used for the ability to reduce a nonlinear training task to a linear separation problem. The structure-centered codesign often lacks isomorphism to a specific model description. Consequently, the common approaches are often stochastic or non-model based, such as particle swarm optimization and genetic algorithms.

On the other hand, operation-centered PhyNN implementations are cases where the codesign starts with the acceleration or partial network isomorphism in mind. PNNs are great examples of operation-based PhyNNs. For instance, the high bandwidth of photonic processing (through temporal and wavelength domain multiplexing) is often leveraged to construct PNNs which have high throughput. PNN's acceleration can most predominantly be found as the connectivity matrix (MVM operation) acceleration. In these types of operation-centered, since the PhyNN device structure are designed with the goal of acceleration in mind, these systems often exhibit sufficient levels of isomorphism for a model/connectivity description to be projected onto the system. For example, in PIC PNN, information such as connectivity (transmission) is fundamentally determined by physical dimensions such as system topology, path length, heater resistance, fabrication tolerance, etc. Thus, it is more sensible to utilize this existing information well rather than deliberately discarding it for the equal treatment to structure-centered codesigns. For these cases, the most desirable category is often the gradient-based methods like BP for the overall fast convergence and generality.

Both codesign logics have their advantages and limitations. For example, the operation-centered codesigns come at a higher repeatability, task generality, and faster convergence (through the compatible training methods). These implementations are generally targeted at the popularization of PhyNN acceleration through reproducibility. PIC-based PNN is a great example where motivation lies in the repeatable acceleration for a wide range of users. On the other hand, structure-centered codesigns have a lower requirement of isomorphism. Yet, the trade-off would be the lower task generality, longer training convergence, and higher difficulty in repeating the system. SFig.8 provides a concise decision flowchart for choosing the appropriate training method.

S13. The alignment-inspired training methods: AT, PAT, and DFA

While methods such as AT, PAT, and DFA are all technically alignment-inspired methods, the concept of alignment is slightly different for each of these methods. Here we describe the difference between these alignment-inspired methods and thus discuss the convergence logic of AT.

To start with, AT and PAT would be categorized into BP-based methods while DFA is alignment-based method. The difference in the categorization originates from the update dynamics of these methods. For example, both AT and PAT utilize the BP-like layer-by-layer update logic, in which the error at the output layer is backwards fed to optimize for the loss with respect to each layer. On the other hand, DFA's update to each layer is obtained through the multiplication between the error matrix and a set of random fixed matrices. Thus, error optimization is a process with respect to the entire deep network structure. (See SFig.9 for the comparison between the update dynamics.) This is also the reason why increasing the network complexity through adding deep layers often result in a less increased performance for DFA compared to BP. The layer-by-layer gradient optimization is the fastest loss reducing operation, which grants BP the faster convergence compared to DFA.

Consequently, despite all being alignment-inspired methods, the BP-based ones and non-BP-based ones still exhibit significant differences. For the two BP-based methods of AT and PAT, we also need to be aware of how each method leverages the concept of alignment differently. For PAT, the digital backwards pass grants a gradient estimator to the

SFig.9 The difference in update dynamics between the BP-based methods (PAT, AT) and alignment-based methods (DFA).

control updates. However, since the control-transformation mapping isn't perfect information for PAT, the update estimator is essentially a relaxed condition of alignment between the actual physical gradient δW_{phy} and the PAT update ΔW_{PAT} . If these two directions are within 90° , then we know the update behavior must be loss-reducing. On the other hand, AT's parallel digital model is a complete standalone forward-backward process trained directly by digital BP. This means that we know the loss behavior for the digital model in AT must be converging. Therefore, the utilization of the alignment concept is between the resulting digital update direction δW_{dig} and the AT update direction ΔW_{AT} . We know that δW_{dig} is a loss-reducing direction; so, within the alignment of 90° , the AT update ΔW_{AT} must also be a loss-reducing direction. This is the reason why we stated that while the learning rate can be time dependent as $\alpha(T)$, it must be the same for the digital and photonic model update within one epoch (see Results section). We can recognize that the overall update is regulated by the digital update (a direction that we know is converging), thus controlling AT's convergence rate with the BP behavior. While both δW_{phy} and δW_{dig} are directions with loss-reducing behavior, we can note that δW_{phy} is purely a local direction specific to the PNN copy while δW_{dig} is general direction that is applicable to multiple PNN copies. Consequently, it leads to the analysis in the main article's Discussion section on the generality of a single digital model in AT for multiple PNN copies to achieve the reduced distributed computational overhead."

2. The authors argue that AT is an improvement on physics aware training (PAT) in both training time and energy consumption, demonstrated by Wright & Onodera et al., because it skips the A/D conversion between layers and relies only on output information. While this is certainly better, it is not clear if it is a significant advantage because, as I mentioned in point 1, ADCs can convert at tens of Gbps. So it is not clear if the energy consumption/latency due to the A/D interface is significant compared to the energy of the digital processor. If the A/D cost is incremental relative to the total energy consumption/speed of the GPU, then PAT

may be better as it uses true BP, and therefore should train faster, while AT may not be as efficient.

Thanks for the comment regarding the comparison to PAT. With the revised Discussion section for the comparison between AT and the IP-BP methods, the focus of the discussion is now on the difference between the construction of the encapsulated DPNN and truncated DPNN. The training and inference of the PNN system is inseparable: it is not sensible to train a truncated structure and assume that the same control applies when connecting the truncated layers together. This means that the truncation of the PNN not only affects the training stage, but also the inference system (which arguably would be the process that is repeated more times for applications). Thus, the question essentially reduces back to the comparison between truncated and encapsulated deep networks. In the newly added analysis (in Supplementary), we made the comparison between the photonic device computation time growth between a readout-set A/D interface and encapsulated in-propagation computation. Through the analysis, we showed that the scaling of the signal conversion and data shuttling time is still more than 50-fold compared to the photonic propagation time even when considering the conversion rate to be 12.5 Gbps (of JESD204B). Consequently, the discussion supports the motivation for constructing an encapsulated PNN.

Also in the newly revised Discussion section, we discussed how the additional forward pass in the digital parallel model can be recognized as a separate standalone process to the local training of a specific PNN device copy. Thus, this process theoretically only needs to be repeated once. When distributed across PNN copies, the overall computational overhead associated with each device is close to a standard BP process, making AT particularly suitable for PNN platforms aimed at reproducibility and repeatability.

We also added a new discussion in the Supplementary file talking about the alignment-inspired training methods: PAT, AT, and DFA. We went through how each of these methods utilize slightly different logics and how the concept of “alignment” is slightly different for each case. Regarding the question of why we classify AT as BP-based method, please see the newly added S13 in Supplementary.

The revisions directly related to this part of the response include:

Results section: Some modifications on the description of how the computational overhead can be reduced and distributed

“In AT’s workflow, acquiring the digital update is independent of the local training of the PNN device (and the AT update is simply an additional operation between the pseudo and digital update). The digital update can be computed separately prior to PNN’s local training. Thus, the local PNN training has the same time-step as a standard BP to avoid any sequential delay. The same set of digital updates can be repeatedly used to train multiple local PNN copies (See later discussion). Consequently, the computational overhead of the parallel model spreads across PNN copies, reducing the local computational overhead associated with each device. (See Discussion section).”

Discussion section: Entire discussion section

“Efficiency of AT-compatible encapsulated structures

While PNNs can offer unique NN acceleration strategies that are not possible on traditional digital NNs, they also face realistic challenges since the codesign (construction) process can be limited by the demanding training operations. Here, we discuss the efficiency that can be enabled by constructing an encapsulated DPNN with AT in comparison to the truncated PNN of existing IP-BP methods.

The encapsulation of DPNN bypasses most of the AD interfaces, significantly reducing the signal conversion and data shuttling time for the sample propagation (during both training and inference). For each propagation of information through the PNN structure, we can define the extraction time, T_{extract} , as the minimum time required to obtain an output for further operation (obtaining a prediction or computing update). A simple formulation of the extraction time is the combination of the propagation time T_{prop} with the readout/conversion time $T_{\text{interface}}$ at the AD

interface. The propagation time for PNN is much shorter than the interface and readout time $T_{\text{prop}} \ll T_{\text{interface}}$ (see Supplementary). Consequently, we could approximate the extraction time scaling as $O(P)$ for encapsulated systems and $O(2M - P)$ for truncated systems (each AD interface corresponds to a pair of readout-and-write operations, thus the factor of 2). For DPNNs, $P < (2M - P)$ always holds true. The extraction time is directly related to how fast the prediction is made, and thus, the operation speed of the PNN. T_{extract} for encapsulated PNN is dependent on the specific task instead of the model complexity. Taking the MNIST 10-class task as example, P stays 10 regardless of the model’s complexity (Fig.6 (a)). Consequently, the encapsulated PNN offers better compatibility overall with increasing model complexity.

Furthermore, the sequential layer-by-layer operation of a truncated DPNN imposes an increased timestep for computation that is not addressable through higher parallelism. For the N hidden layers, a total of $N + 1$ timestep is required for each single forward propagation, while there is always only 1 timestep for encapsulated DPNN (Fig.6 (b)). The constant timestep offers good compatibility with schemes of higher parallelism, such as wavelength division multiplexing.

The extraction time can also be related to the energy consumption of the photonic device during the inference. For a photonic system with power consumption of P_{photonic} , the minimum operation time is the extraction time for obtaining a prediction. Thus, the minimum energy required to operate the photonic system would be $E_{\text{photonic}} = P_{\text{photonic}} \cdot T_{\text{extract}}$. For systems where the limiting power consumption is photonic hardware, the longer the T_{extract} , the larger the energy inefficiencies are created due to the prolonged operation.

Distributed overhead of AT with repeatable application to trainable PNN structures

While PNN training methods utilizing a parallel/twin digital model can provide a more accurate estimate of the physical gradients, the typical concern is the computational overhead that might come with the application of the parallel digital model. Here we discuss how computational overhead (and the associated additional energy consumption) in AT can be avoided to a computational overhead similar to standard BP.

As mentioned in the Results section, the digital update of the parallel model can be computed independent of the local PNN’s error, allowing for the same computational timestep as BP without sequential delay. The key motivation for reproducible PNN platforms such as PICs is often associated with reproducible acceleration despite the device-to-device variations. The same PNN structure might be fabricated for many copies to achieve local acceleration of a specific task. For a total of C copies of PNN devices, the overall description of each PNN system can be formulated as $f_{p,1;i} (f_{p,2;i}(\dots))$ for $i = 1, 2, \dots, C$, denoting the specific system under discussion. In the case of chip fabrication variation, the overall descriptions of these systems form a statistically relevant set. Thus, we note the equivalence of different estimation profiles and physical device copies. The generality of successful training with different profiles (see Fig.3 (d)) can also be interpreted as different copies of the PNN systems trained equally with a single digital model.

From Eq.3, the overall AT estimator is obtained through two parts: the digital parallel update (ΔW_{dig}) and the pseudo update (ΔW_{pseudo}). For a given task and initialization condition, the update of the digital model is independent of the local PNN device and can be described as $\Delta W_{\text{dig}}^{t=t'} (W_0^{t=t_0}; X, Y, \alpha^{t=t'})$ at time instance $t = t'$. On the other hand, ΔW_{pseudo} is local to the specific PNN device copy (subject to variation) as $\Delta W_{\text{pseudo}}^{t=t'} (f_{p,i}; W_0^{t=t_0}; X, Y, \alpha^{t=t'})$. The generality of a single digital model to different variations of the PNN systems means that the collection of the digital updates can be stored and reapplied to different PNN devices without the need to repeat the process of computing the digital updates (thus no need to repeat the parallel model’s backpropagation). For multiple PNN systems, the forward pass in the parallel model only needs to be computed once. Effectively, the computational overhead required to utilize the digital parallel model is distributed across the PNN copies (Fig.6 (c)). Thus, the overhead of operation is significantly reduced for each individual device. When $C \gg 1$, the overall computational overhead associated with each individual PNN device is reduced towards the overhead of a standard BP process. This allows the benefits of

encapsulation to be enjoyed while little overhead is added on top of the standard BP process for training of many devices.”

Supplementary: S10, S12, S13

“S10. Discussion on the interface speed, propagation speed and propagation time of the AT encapsulated PNN

As described in the main article, we have defined the extraction time T_{extract} to be the minimum time required before sufficient information is available from the PNN structure for useful operation. More specifically, T_{extract} is the summation of the propagation time through the PNN device and the interface time required to readout that piece of information. One of the assumptions we have made in the discussion is that the propagation time is a much shorter time than the access/interface time. Here we discuss further the details of this argument.

For representation of a A node to A node connection, the most general hardware representation would be a fully connected $A \times A$ connection. If we define the hardware depth as the number of repetitive components that the signal must encounter before fully modulated for the $A \times A$ connectivity. We can realize that the minimum scaling to be A as a direct mapping topology. Even with other decomposed topologies⁸⁻¹⁰, the scaling of the photonic hardware path grows linearly with the connectivity size. The propagation time of the in-transformation computation is directly proportional to the hard depth. If we take the SiN platform used in the experiments as an example, the refractive index is around 1.93¹¹. The total on-chip waveguide length of the device is roughly 12.5 mm, corresponding to an on-chip propagation time of 0.08 ns. We can use this as an estimation of the basic propagation time for the hardware component as 0.02 ns per photonic depth.

In comparison, even when considering specialized serial communication protocol of JESD204B¹², with a data transmission rate of 12.5 Gbps. (Here we are considering the data transmission to be the bottleneck of the interface speed. Depending on the exact setup, it is also possible that the ADC sampling rate is the bottleneck.) For the control and readouts of 16-bits, we can expect a roughly 1.28 ns time cost for each access point. The scaling of the access point is described as $O(2M - P)$. Consequently, even when we ignore the internal data shuttling time the scaling of the interface time with respect to the hidden neuron number is over $100\times$ (for the factor of 2) compared to the scaling of the propagation time. Thus, in the main article where relevant, the description of the extraction time scaling for the encapsulated deep PNN is defined as $O(P)$ for simplification. Furthermore, for the other more accessible protocol such as serial SPI communication, the data transmission rate is typically 60-100 Mbps (27 μs), creating an even larger gap between the propagation time and the interface time. We can see from the above analysis that the construction of an encapsulated DPNN can be time efficient compared to truncated ones.

S12. Different types of PhyNN training methods and their applicational scenarios

SFig.8 A decision flowchart for choosing the appropriate training method category depending on the applicational scenario.

There are many different categories of PhyNN training methods. Each category of training methods has their own unique properties and advantages. Unlike mathematical (digital) NNs, constructing a PhyNN is often a codesign process between the physical device structure, the task, the applicational scenario, and the training algorithm used. Consequently, it is not sensible to regard a single class of training methods as the elixir for every type of PhyNN construction. Instead, it is more appropriate to understand that each category has its own appropriate application scenarios. Here we present a summarized discussion on how different categories of training methods can be more suitable for different types of PhyNN codesign (implementation strategy).

SFig.8 shows a decision flowchart for choosing suitable methods for a specific PhyNN implementation technique. The construction/codesign of PhyNNs can be broadly divided into structure-centered ones and operation-centered ones. Structure-centered PhyNNs are typically the ones where the codesign logic is based on the attempt to map a device with high complexity to certain specific tasks. One great example of the structure-centered PhyNN is reservoir computing, where a device with high complexity (randomized spatial and temporal connectivity) is used for the ability to reduce a nonlinear training task to a linear separation problem. The structure-centered codesign often lacks isomorphism to a specific model description. Consequently, the common approaches are often stochastic or non-model based, such as particle swarm optimization and genetic algorithms.

On the other hand, operation-centered PhyNN implementations are cases where the codesign starts with the acceleration or partial network isomorphism in mind. PNNs are great examples of operation-based PhyNNs. For instance, the high bandwidth of photonic processing (through temporal and wavelength domain multiplexing) is often leveraged to construct PNNs which have high throughput. PNN's acceleration can most predominantly be found as the connectivity matrix (MVM operation) acceleration. In these types of operation-centered, since the PhyNN device structure are designed with the goal of acceleration in mind, these systems often exhibit sufficient levels of isomorphism for a model/connectivity description to be projected onto the system. For example, in PIC PNN,

information such as connectivity (transmission) is fundamentally determined by physical dimensions such as system topology, path length, heater resistance, fabrication tolerance, etc. Thus, it is more sensible to utilize this existing information well rather than deliberately discarding it for the equal treatment to structure-centered codesigns. For these cases, the most desirable category is often the gradient-based methods like BP for the overall fast convergence and generality.

Both codesign logics have their advantages and limitations. For example, the operation-centered codesigns come at a higher repeatability, task generality, and faster convergence (through the compatible training methods). These implementations are generally targeted at the popularization of PhyNN acceleration through reproducibility. PIC-based PNN is a great example where motivation lies in the repeatable acceleration for a wide range of users. On the other hand, structure-centered codesigns have a lower requirement of isomorphism. Yet, the trade-off would be the lower task generality, longer training convergence, and higher difficulty in repeating the system. SFig.8 provides a concise decision flowchart for choosing the appropriate training method.

S13. The alignment-inspired training methods: AT, PAT, and DFA

While methods such as AT, PAT, and DFA are all technically alignment-inspired methods, the concept of alignment is slightly different for each of these methods. Here we describe the difference between these alignment-inspired methods and thus discuss the convergence logic of AT.

To start with, AT and PAT would be categorized into BP-based methods while DFA is alignment-based method. The difference in the categorization originates from the update dynamics of these methods. For example, both AT and PAT utilize the BP-like layer-by-layer update logic, in which the error at the output layer is backwards fed to optimize for the loss with respect to each layer. On the other hand, DFA's update to each layer is obtained through the multiplication between the error matrix and a set of random fixed matrices. Thus, error optimization is a process with respect to the entire deep network structure. (See SFig.9 for the comparison between the update dynamics.) This is also the reason why increasing the network complexity through adding deep layers often result in a less increased performance for DFA compared to BP. The layer-by-layer gradient optimization is the fastest loss reducing operation, which grants BP the faster convergence compared to DFA.

Consequently, despite all being alignment-inspired methods, the BP-based ones and non-BP-based ones still exhibit significant differences. For the two BP-based methods of AT and PAT, we also need to be aware of how each method leverages the concept of alignment differently. For PAT, the digital backwards pass grants a gradient estimator to the

SFig.9 The difference in update dynamics between the BP-based methods (PAT, AT) and alignment-based methods (DFA).

control updates. However, since the control-transformation mapping isn't perfect information for PAT, the update estimator is essentially a relaxed condition of alignment between the actual physical gradient δW_{phy} and the PAT update ΔW_{PAT} . If these two directions are within 90° , then we know the update behavior must be loss-reducing. On the other hand, AT's parallel digital model is a complete standalone forward-backward process trained directly by digital BP. This means that we know the loss behavior for the digital model in AT must be converging. Therefore, the utilization of the alignment concept is between the resulting digital update direction δW_{dig} and the AT update direction ΔW_{AT} . We know that δW_{dig} is a loss-reducing direction; so, within the alignment of 90° , the AT update ΔW_{AT} must also be a loss-reducing direction. This is the reason why we stated that while the learning rate can be time dependent as $\alpha(T)$, it must be the same for the digital and photonic model update within one epoch (see Results section). We can recognize that the overall update is regulated by the digital update (a direction that we know is converging), thus controlling AT's convergence rate with the BP behavior. While both δW_{phy} and δW_{dig} are directions with loss-reducing behavior, we can note that δW_{phy} is purely a local direction specific to the PNN copy while δW_{dig} is general direction that is applicable to multiple PNN copies. Consequently, it leads to the analysis in the main article's Discussion section on the generality of a single digital model in AT for multiple PNN copies to achieve the reduced distributed computational overhead."

3. I still do not agree that AT is "calibration-free." My concern earlier was that it is demonstrated on a programmable PIC, and of course we know how MZIs work and that they have a cosine transfer function. The authors note that no readout is performed from the PIC – yes, I agree, but some a priori knowledge about the system is leveraged here.

What would we do for a physical system where we have no information about the transfer function at all? How would AT be used for any of the systems in the original PAT paper? Clearly some calibration would be

needed.

In any case, the authors have adequately narrowed this claim by indicating AT only applies to “scalable isomorphic physical neural networks.” Instead of inventing new terminology, I think it would be preferable to use the terminology already standard in the community – either photonic neural network or optical neural network, both of which are usually understood to utilize optical matrix-vector multipliers.

Thank you for the suggestions to use more conventional terminologies. We have now changed the definition in text of PNN to stand for “photonic neural networks” instead of “physical neural networks”. In places where physical neural networks are referred, we used the term PhyNN.

Regarding the terminology of “calibration-free”, we understand that the definition of this word can vary. The difference lies in that it seems your definition of calibration-free is more like no information is provided at all, while we defined calibration to be the detailed characterization process which sets up an input-output relationship as a measured look-up table.

We respect your opinion on this, and thus have replaced the term of “calibration-free” to “characterization-free” or “free of local characterization” where appropriate in the main article. We think the term “characterization” in multiple areas such as material science and engineering, quite universally refer to the process where the local variation of a sample or device is detailedly studied through repetitive input-output feedback. Consequently, the new description should now avoid confusion for readers to understand that the case we are describing here is free of need to know the exact local information of each PNN copy. We maintained the discussion on how the acquisition of some prior knowledge doesn’t break the encapsulation of PNN. The description in the Results section now clearly states how the construction of the estimation profile only requires very fundamental fabrication information.

Since we have now focused the discussion on photonic neural networks (whose topology is interpretable), there will always be sufficient information for the user to deduce the basic system state and the estimation profile. For example, the transfer function of a unit such as MZI is fundamentally determined by the physical dimensions of the components (path difference, heater length, resistance, etc.), all this information should be available through fabrication design. Thus, for the PNN systems, we think it is unnecessary to deliberately discard the information that we already have. We have added new descriptions to the main article to make sure that this is clear to the readers.

In the discussion of applying different training methods in the Supplementary section, we discuss the preferred training methods for different PhyNNs based on two broad categories of operation-centered and structure-centered. The PNNs are operation-centered implementations, which should provide us with sufficient knowledge for constructing an estimation profile regardless of the local variation of the specific copy.

To clarify the points mentioned in this response, we have made the following revisions:

Introduction section: Replaced the term “calibration-free” with “free of local characterization”

“We further experimentally validated AT’s ability to achieve training with only output neuron information for different tasks (Iris-flower classification and modified hand-written digit classification) and scaled-up structures through repetitive use of PPMs free of local characterization.”

Results section: Replaced the term “calibration-free” with “characterization-free” and talked about more details on how the estimation profile can be obtained from fundamental information of the system without breaking the encapsulation (and thus not affecting the novelty of the AT method).

“Conventional PNN training requires precise and detailed system-level characterization to control the physical parameters. However, when the entire deep structure is encapsulated, the input-to-output response is unreliable for

separating the effects from different layers. Consequently, the characterization of an encapsulated DPNN requires intermediate neuron access information. To rigorously construct full encapsulation where the local (device-specific) hidden neuron information is not accessed during any stage of the training, we employ an estimation profile for applying the physical control parameters. The estimation profile is a sensible statistical deduction of the PPM's control-transformation without considering the local variation of individual devices (characterization-free). In addition to the estimation profile's ability to avoid intermediate readouts, it also brings the advantage of reduced memory requirement compared to standard fully calibrated training. The total number of control parameters scales roughly as $\propto M^2$. For a finite resolution of R of the analogue system control and neuron size of M , the required memory scales as $O(RM^2)$ for lookup tables in fully calibrated training. In comparison, the estimation profile is applied to all controllable physical parameters, offering a reduced memory requirement of $O(R)$. (More discussions on the estimation profile in the next subsection and Supplementary.)

Taking the construction of PIC-PNN here as an example, the neuron connection strength is determined by the transmission of the MZI unit, which is directly controlled by the power added to the thermos-optic phase shifter via voltage V . We can thus formulate an estimation of how to apply the physical parameter. The transmission of the MZI cell can be estimated theoretically with a simple cosine-like shape as Eq.4, where γ is some constant. This profile can be obtained without the need to perform any local characterization.”

Supplementary: S12, includes a discussion of the operation-centered vs. structure-centered PhyNN implementation strategies

“S12. Different types of PhyNN training methods and their applicational scenarios

There are many different categories of PhyNN training methods. Each category of training methods has their own unique properties and advantages. Unlike mathematical (digital) NNs, constructing a PhyNN is often a codesign process between the physical device structure, the task, the applicational scenario, and the training algorithm used. Consequently, it is not sensible to regard a single class of training methods as the elixir for every type of PhyNN construction. Instead, it is more appropriate to understand that each category has its own appropriate application scenarios. Here we present a summarized discussion on how different categories of training methods can be more suitable for different types of PhyNN codesign (implementation strategy).

SFig.8 shows a decision flowchart for choosing suitable methods for a specific PhyNN implementation technique. The construction/codesign of PhyNNs can be broadly divided into structure-centered ones and operation-centered ones. Structure-centered PhyNNs are typically the ones where the codesign logic is based on the attempt to map a device with high complexity to certain specific tasks. One great example of the structure-centered PhyNN is reservoir computing, where a device with high complexity (randomized spatial and temporal connectivity) is used for the ability to reduce a nonlinear training task to a linear separation problem. The structure-centered codesign often lacks isomorphism to a specific model description. Consequently, the common approaches are often stochastic or non-model based, such as particle swarm optimization and genetic algorithms.

On the other hand, operation-centered PhyNN implementations are cases where the codesign starts with the acceleration or partial network isomorphism in mind. PNNs are great examples of operation-based PhyNNs. For instance, the high bandwidth of photonic processing (through temporal and wavelength domain multiplexing) is often leveraged to construct PNNs which have high throughput. PNN's acceleration can most predominantly be found as the connectivity matrix (MVM operation) acceleration. In these types of operation-centered, since the PhyNN device structure are designed with the goal of acceleration in mind, these systems often exhibit sufficient levels of isomorphism for a model/connectivity description to be projected onto the system. For example, in PIC PNN, information such as connectivity (transmission) is fundamentally determined by physical dimensions such as system topology, path length, heater resistance, fabrication tolerance, etc. Thus, it is more sensible to utilize this existing information well rather than deliberately discarding it for the equal treatment to structure-centered codesigns. For

SFig.8 A decision flowchart for choosing the appropriate training method category depending on the applicational scenario.

these cases, the most desirable category is often the gradient-based methods like BP for the overall fast convergence and generality.

Both codesign logics have their advantages and limitations. For example, the operation-centered codesigns come at a higher repeatability, task generality, and faster convergence (through the compatible training methods). These implementations are generally targeted at the popularization of PhyNN acceleration through reproducibility. PIC-based PNN is a great example where motivation lies in the repeatable acceleration for a wide range of users. On the other hand, structure-centered codesigns have a lower requirement of isomorphism. Yet, the trade-off would be the lower task generality, longer training convergence, and higher difficulty in repeating the system. SFig.8 provides a concise decision flowchart for choosing the appropriate training method.”

4. I think there is a miscommunication regarding my question about model initialization. I understood that when training the parameters should be randomly initialized. My question was whether the digital twin needs to start with parameter settings close to the physical hardware, or can the digital twin be completely “off” from the physical setup and AT will align the two over time. For instance, in an MZI mesh, suppose all of the hardware is set to the “cross” state. Does the digital model also need to be initialized such that all the software MZIs are in the “cross” state, or can they be random?

The authors’ response suggests they should be close to the physical hardware. This supports my point that AT is not calibration-free—if it were, it should not matter and we could start training with zero information about the system. The authors provide some experiments showing the use of different estimation profiles, but they all are qualitatively cosines with some shift.

To be clear, this is all fine and I do not consider this a significant limitation of the method. But the description of the method should be accurate to avoid confusing readers.

We really appreciate your rigorousness for the most accurate description to the readers. As mentioned in the response to the last part, we have now replaced the term “calibration-free” to “characterization-free”, which is stating that the training process doesn’t need to consider the local variation of the PNN copies (due to device-to-device and system-to-system variations). We also talked about how the relevant information to construct a theoretical profile is inherently available to readers for operation-based PNNs through fabrication design. The acquisition of the information doesn’t require an input-output feedback process and thus doesn’t affect the encapsulation of DPNNs.

For the specific case you have mentioned: for instance, if the randomized initialization for one specific element in the digital model is “cross” state, then the physical control is also set to the “cross” state through the estimation profile. However, the important thing is that the transmission of the MZI isn’t necessarily actually the “cross” state because the estimation profile isn’t supposed to be an accurate control look-up (the one same estimation profile is universally applied to all the controls). Thus, the actual transmission of the MZI might be halfway between cross and bar for example. The idea is that the control setting to the physical and digital model starts as identical, and gradually becomes asymmetrical through the training process; but the actual transmission encodings which start with a mismatch are gradually mapped together through the asymmetrical control parameters. Hopefully this answers your question regarding the initialization.

We also want to mention that AT’s correction to the mismatch isn’t limited to only the translational effect of the transfer function. For example, in all the simulations that we conducted, the element-wise deviation of the transformation is described as follows (more details can be found in Methods section and the Supplementary):

$$f_p^t(W^t) = (I + P_{\text{sys}}^t) \otimes (f_m^t(W^t) + N_{\text{init}}) + N_{\text{rand}}^t$$

For the error that we have defined, P_{sys}^t is matrix of the same size as the connectivity matrix of the given layer. Thus, each element in the transformation matrix is independently disturbed by the error matrix. Through our simulation in Fig.5 (d) (varying the level of error), we showed that even for the elementwise variation with magnitude equal to the set weight matrix, the training is still well-behaved. In another word, for the purpose of PNN, some of the cells are actually in the “cross” state, even though the user thought they are setting the cell to “bar” state. AT is still able to correct these through the update process. In the simulation, the training can still achieve a loss reducing behavior with the test accuracy of MNIST to be over 90% for all the physically relevant data points. Thus, the error correction process is general and not limited to the translational effect of the estimation profile. We have also revised the description in the subsection of “Asymmetrical training method” (Results section) to make sure the procedure is clearer to the readers.

The revisions related to this part of the response include:

Results section: Specifically in the description of the experiments, talked about the origin of the estimation profile. The description of the AT method has also been modified slightly to give more clarity.

“To understand the AT method, we first define two separate sets of parameters for the parallel model $\mathbf{W}_{(t,\text{dig})}$ and physical control $\mathbf{W}_{(t,\text{phy})}$. (We use \mathbf{W} to refer to all the controllable parameters in the following discussion for simplicity.) At the initialization of $t = 0$, we set the digital and physical control with the same randomized values, $\mathbf{W}_{(0,\text{dig})} = \mathbf{W}_{(0,\text{phy})}$ (note that the two resulting transformations are different: $f_m(\mathbf{W}_{(0,\text{dig})}) \neq f_p(\mathbf{W}_{(0,\text{phy})})$). The digital and physical systems simultaneously forward-propagate the same sample information, resulting in a pair of

predictions $\mathbf{a}_{(0,\text{dig})}^{[N+2]}$ and $\mathbf{a}_{(0,\text{phy})}^{[N+2]}$. The pair of predictions leads to two different sets of errors ($\mathbf{E}_{(0,\text{dig})}$ and $\mathbf{E}_{(0,\text{phy})}$) at the output layer for the same targets. Despite there is no access to intermediate neuron information of physical system, we have the complete information of the parallel model ($\mathbf{a}_{\text{dig}}, \mathbf{W}_{\text{dig}}, \mathbf{b}_{\text{dig}}, \mathcal{G}_{\text{dig}}$).

The collection of parallel model parameters is used for backpropagating the two sets of errors. We define two update terms based on backpropagating the two errors $\mathbf{E}_{(\text{dig})}$ and $\mathbf{E}_{(\text{phy})}$. When backpropagating $\mathbf{E}_{(\text{dig})}$, we get the gradient update to the digital parallel model as $\Delta \mathbf{W}_{\text{dig}} = \partial L / \partial \mathbf{W}_{\text{dig}} \equiv \delta \mathbf{W}_{\text{dig}}$. On the other hand, results from backpropagating $\mathbf{E}_{(\text{phy})}$ has no direct physical significance on its own; we call it the pseudo update $\Delta \mathbf{W}_{\text{pseudo}}^{[l]}$. The AT estimator is defined as the mixture of these two updates under the regulation of the mixing ratios M_{w1} and M_{w2} , which normally takes the equal values of $M_{w1} = M_{w2} = 0.5$ (see Methods and Supplementary). The overall expression is shown in Eq. (3), where the digital parallel and the pseudo updates are mixed at each layer for the AT update. (See Methods and Supplementary for more theoretical discussions)

$$\Delta \mathbf{W}_{\text{AT}}^{[l]} = M_{w1} \cdot \Delta \mathbf{W}_{\text{dig}}^{[l]} + M_{w2} \cdot \Delta \mathbf{W}_{\text{pseudo}}^{[l]} \quad (3)$$

At every later instance ($t > 0$), the digital parallel model is trained purely by BP's gradient update of $\Delta \mathbf{W}_{\text{dig}}^{[l]}$. PNN's physical control parameters are updated with $\Delta \mathbf{W}_{\text{AT}}^{[l]}$. We see that the pseudo update is always non-identical to the digital update for non-ideal PPMs. Consequently, the control settings to the digital and physical system become asymmetrical ($\mathbf{W}_{(t,\text{dig})} \neq \mathbf{W}_{(t,\text{phy})}$, for $t > 0$) as the training progresses, giving the name of this method. The key idea behind AT is that the initially identical control parameter leads to different resulting transformations. The later training updates attempt to use asymmetrical settings to correct the transformation of the photonic system towards the mathematical description in the parallel model. (See more discussion in Methods.) As long as the learning rates are the same for both systems at a given time instance, they can vary as time progresses, making the AT estimator compatible with conventional optimizers⁴⁶⁻⁴⁸. The overall workflow of the AT method is illustrated in Fig.1 (c)."

Results section: The terminology of “calibration-free” is replaced with “characterization-free” where appropriate.

“Conventional PNN training requires precise and detailed system-level characterization to control the physical parameters. However, when the entire deep structure is encapsulated, the input-to-output response is unreliable for separating the effects from different layers. Consequently, the characterization of an encapsulated DPNN requires intermediate neuron access information. To rigorously construct full encapsulation where the local (device-specific) hidden neuron information is not accessed during any stage of the training, we employ an estimation profile for applying the physical control parameters. The estimation profile is a sensible statistical deduction of the PPM's control-transformation without considering the local variation of individual devices (characterization-free). In addition to the estimation profile's ability to avoid intermediate readouts, it also brings the advantage of reduced memory requirement compared to standard fully characterized training. The total number of control parameters scales roughly as $\propto M^2$. For a finite resolution of R of the analogue system control and neuron size of M , the required memory scales as $O(RM^2)$ for lookup tables in fully characterized training. In comparison, the estimation profile is applied to all controllable physical parameters, offering a reduced memory requirement of $O(R)$. (More discussions on the estimation profile in the next subsection and Supplementary.)

Taking the construction of PIC-PNN here as an example, the neuron connection strength is determined by the transmission of the MZI unit, which is directly controlled by the power added to the thermos-optic phase shifter via voltage V . We can thus formulate an estimation of how to apply the physical parameter. The transmission of the MZI cell can be estimated theoretically with a simple cosine-like shape as Eq.4, where γ is some constant. This profile can be obtained without the need to perform any local characterization.”

5. My question on algorithmic scaling was not adequately addressed. The authors provide a new table in the main text comparing the scaling, but they focus primarily on number of readout time steps and intermediate readout. First, some of the entries for other algorithms are incorrect and thus the comparison is inaccurate. Finite stochastic (FS), as demoed in Bandyopadhyay et al., did not require intermediate readout. The demonstration of in-situ BP in Pai et al., 2023 did require multiple readout time steps, but it is not strictly necessary as the backpropagation can be performed all-optically – see Spall et al, arXiv:2308.05226. DFA should not require calibration, as the backwards pass can use a fixed, random weight matrix for feedback (for instance, Nakajima et al. demonstrated DFA on a reservoir computer).

The authors also claim that AT has identical scaling to backpropagation, but no simulation or experimental evidence is provided to support this. I'm not sure this is true, since alignment methods usually are not as efficient as true BP—the authors acknowledge this in the table, as for direct feedback alignment (DFA) they claim the scaling is worse than BP. They categorize the scaling of prior algorithms as " $O(T)$ ", " $>O(T)$ ", and " $>>O(T)$ ", where $O(T)$ is equivalent to backprop, but it is not clear how they determine these scalings for each given algorithm without some rigorous comparison. The exact scaling is important, as some of these algorithms, such as PAT and AT, require digital systems and others, such as FS and in-situ BP, do not, and thus the question is whether the added power consumption and slow processing of a digital system is worth the improved training efficiency, as what we care about is the total energy and time spent to fully train a model.

For the discussion of the algorithmic scaling, we have previously consulted the review article “Training of physical neural networks” by Momeni et al. at <https://arxiv.org/abs/2406.03372> . In the revised manuscript, we have now removed the table comparison and only quoted the key information that for the definition of BP-based training to be $O(T_0)$, other non-BP-based non-gradient methods have the general convergence of $> O(T_0)$ (please see Table 1 on Page 17 of Momeni et al.’s review). We agree that the specific convergence can be dependent on the task, hyperparameters, and initialization conditions, etc. For instance, the optimal learning rate for stochastic methods and BP can be very different, and we think that constructing an exhaustive list for comparison to include all the existing tasks is impossible. However, there are certainly trends and general properties of training methods within different categories. We think it will be easier for the readers to follow the discussion by directing them to the review paper, instead of us trying to extensively restate the discussion in that paper.

As discussed in previous parts, we included a new discussion about what types of training method might be preferable for the users in the Supplementary file. Different categories of training methods have their own unique applicational scenarios. Therefore, we think our job is to express this clearly to the readers with the new discussion in the Supplementary file. In the revision of the manuscript, we focused more on the comparison between BP-based methods, which are directly comparable without ambiguity. Apologies for the confusion in the previous manuscript.

Thank you for bringing up the paper by Spall et al. We have carefully checked through their work and included it as part of the reference for the interested users. It is, however, worth mentioning that we found their method also requires intermediate readout. For example, in Fig.1 (e) of their paper, they have listed the schematic of their method (see below circled part). For a deep network, they also need to access the intermediate state information for computing the gradient update estimator. Thus, the overall methodology is closer to the idea of PAT (and their previous work on hybrid training). While the backwards pass computation is accelerated with photonics, the backwards pass computation in AT can also theoretically be accelerated with photonics computation. Therefore, we don't think their work affects the novelty of AT. But thank you for mentioning this work, we have now added it to the reference for the interested readers.

Figure Redacted

(We have included more discussion about the FS method in the next part of the response and we want to talk first about the alignment-based method here.) For the purely alignment-based method, such as DFA, the idea is that the error matrix E is multiplied by a fixed set of random matrices B_i for $i = 1, 2, \dots, N$ in each layer. This is the original condition for the DFA update to have loss-reducing behavior. However, for the implementation of PNNs, we know that the resulting transformation is different from the set physical control parameters, $W_p \neq f_p(W_p)$. Therefore, if we directly set the update each time, the overall set parameter can't be deemed as a fixed matrix for multiplication. In our supplementary file "S7. More discussion on the concept of AT and why pure BP or DFA wouldn't be sufficient for training", we have included a simulation of training an erroneous DPNN with only DFA and showed that it is not sufficient for good performance.

For purely DFA alignment-based training, the system must also have some level prior knowledge to the transformation of the PNN. Thus, if the prior knowledge acquisition of AT from the fabrication can't be classified as calibration-free training, then we think purely alignment based DFA also can't be classified as completely calibration-free training as some primitive knowledge of the system is required for maintaining the alignment criterion. On the other hand, if it were a reservoir computing setting, then the overall loss-reducing behavior is likely to come from the linearly reducible property of a higher-complexity reservoir rather than the gradient alignment of update directions.

As mentioned in the previous response part, we now changed the terminology of "calibration-free" to "characterization-free", which we think this should now more clearly describe the method's ability to train regardless of the local variations where the estimation profile is inherently available through the very fundamental information such as fabrication design.

We have made revisions in the following places of the manuscript for this part of the response:

Introduction section: Removed the table comparison and revised the description

"Backpropagation²⁰ (BP) finds the gradient update to the loss function for the network parameters, offering fast training gradient convergence²¹. BP's advantage for fast training with DNNs comes from two key aspects: BP's robustness, generality and fast convergence make it a well-balanced industry standard approach for most digital DNN-based tasks. However, standard BP imposes requirements of highly accurate model control and knowledge. The inevitable fabrication variation inherently creates imperfect knowledge for the users. In addition, the lack of perfect control-transformation mapping

on the typically noisy and lossy analogue computation platforms^{22,23} can't satiate the high requirements of standard BP. The mismatch between the user's interpretation and the PNN's actual state can lead to failed training or dramatic performance degradation^{24,25}. Therefore, some non-gradient-based²⁶⁻³¹ or model-free/stochastic methods³²⁻³⁵ are proposed to bypass the requirement for formulating a model description. Nonetheless, the natural trade-off with non-gradient methods is the higher difficulty to train complex tasks and slower convergence compared to the BP-based methods. When defining the convergence of standard BP to be $O(T_0)$, the convergences of other non-gradient PNN training methods are typically $> O(T_0)$ ³⁶. PNNs, being mostly operation-centered platforms, are highly compatible with BP-based gradient methods for the typically exhibited isomorphism. (See Supplementary for more discussion on the categorization of PhyNNs.)"

Discussion section: Focused more on the comparison between BP-based methods

“Efficiency of AT-compatible encapsulated structures

While PNNs can offer unique NN acceleration strategies that are not possible on traditional digital NNs, they also face realistic challenges since the codesign (construction) process can be limited by the demanding training operations. Here, we discuss the efficiency that can be enabled by constructing an encapsulated DPNN with AT in comparison to the truncated PNN of existing IP-BP methods.

The encapsulation of DPNN bypasses most of the AD interfaces, significantly reducing the signal conversion and data shuttling time for the sample propagation (during both training and inference). For each propagation of information through the PNN structure, we can define the extraction time, T_{extract} , as the minimum time required to obtain an output for further operation (obtaining a prediction or computing update). A simple formulation of the extraction time is the combination of the propagation time T_{prop} with the readout/conversion time $T_{\text{interface}}$ at the AD interface. The propagation time for PNN is much shorter than the interface and readout time $T_{\text{prop}} \ll T_{\text{interface}}$ (see Supplementary). Consequently, we could approximate the extraction time scaling as $O(P)$ for encapsulated systems and $O(2M - P)$ for truncated systems (each AD interface corresponds to a pair of readout-and-write operations, thus the factor of 2). For DPNNs, $P < (2M - P)$ always holds true. The extraction time is directly related to how fast the prediction is made, and thus, the operation speed of the PNN. T_{extract} for encapsulated PNN is dependent on the specific task instead of the model complexity. Taking the MNIST 10-class task as example, P stays 10 regardless of the model's complexity (Fig.6 (a)). Consequently, the encapsulated PNN offers better compatibility overall with increasing model complexity.

Furthermore, the sequential layer-by-layer operation of a truncated DPNN imposes an increased timestep for computation that is not addressable through higher parallelism. For the N hidden layers, a total of $N + 1$ timestep is required for each single forward propagation, while there is always only 1 timestep for encapsulated DPNN (Fig.6 (b)). The constant timestep offers good compatibility with schemes of higher parallelism, such as wavelength division multiplexing.

The extraction time can also be related to the energy consumption of the photonic device during the inference. For a photonic system with power consumption of P_{photonic} , the minimum operation time is the extraction time for obtaining a prediction. Thus, the minimum energy required to operate the photonic system would be $E_{\text{photonic}} = P_{\text{photonic}} \cdot T_{\text{extract}}$. For systems where the limiting power consumption is photonic hardware, the longer the T_{extract} , the larger the energy inefficiencies are created due to the prolonged operation.

Distributed overhead of AT with repeatable application to trainable PNN structures

While PNN training methods utilizing a parallel/twin digital model can provide a more accurate estimate of the physical gradients, the typical concern is the computational overhead that might come with the application of the parallel digital model. Here we discuss how computational overhead (and the associated additional energy consumption) in AT can be avoided to a computational overhead similar to standard BP.

As mentioned in the Results section, the digital update of the parallel model can be computed independent of the local PNN’s error, allowing for the same computational timestep as BP without sequential delay. The key motivation for reproducible PNN platforms such as PICs is often associated with reproducible acceleration despite the device-to-device variations. The same PNN structure might be fabricated for many copies to achieve local acceleration of a specific task. For a total of C copies of PNN devices, the overall description of each PNN system can be formulated as $f_{p,1;i} (f_{p,2;i}(\dots))$ for $i = 1, 2, \dots, C$, denoting the specific system under discussion. In the case of chip fabrication variation, the overall descriptions of these systems form a statistically relevant set. Thus, we note the equivalence of different estimation profiles and physical device copies. The generality of successful training with different profiles (see Fig.3 (d)) can also be interpreted as different copies of the PNN systems trained equally with a single digital model.

From Eq.3, the overall AT estimator is obtained through two parts: the digital parallel update (ΔW_{dig}) and the pseudo update (ΔW_{pseudo}). For a given task and initialization condition, the update of the digital model is independent of the local PNN device and can be described as $\Delta W_{\text{dig}}^{t=t'} (W_o^{t=t_0}; X, Y, \alpha^{t=t'})$ at time instance $t = t'$. On the other hand, ΔW_{pseudo} is local to the specific PNN device copy (subject to variation) as $\Delta W_{\text{pseudo}}^{t=t'} (f_{p,i}; W_o^{t=t_0}; X, Y, \alpha^{t=t'})$. The generality of a single digital model to different variations of the PNN systems means that the collection of the digital updates can be stored and reapplied to different PNN devices without the need to repeat the process of computing the digital updates (thus no need to repeat the parallel model’s backpropagation). For multiple PNN systems, the forward pass in the parallel model only needs to be computed once. Effectively, the computational overhead required to utilize the digital parallel model is distributed across the PNN copies (Fig.6 (c)). Thus, the overhead of operation is significantly reduced for each individual device. When $C \gg 1$, the overall computational overhead associated with each individual PNN device is reduced towards the overhead of a standard BP process. This allows the benefits of encapsulation to be enjoyed while little overhead is added on top of the standard BP process for training of many devices.”

Supplementary S7: showing that DFA without any prior knowledge can’t provide satisfactory performance

“One part of the training with AT is the alignment of the physical system’s transformation towards the digital parallel model. However, the simple concept of alignment by itself isn’t sufficient for training PNNs, instead it must be combined with BP update in the digital model. The criterion of alignment-based methods^{4,5} such as DFA requires an additional set of random fixed matrices which the error matrix \mathbf{E} is multiplied by and directly fed to the layers as the updates. The original criterion of DFA relies on the fact that the multiplication with the fixed matrix is statistically aligned with the actual gradient update to the system. However, this criterion is only true for digital systems where the resulting transformation $f_m(\cdot)$ is clearly describable with the parameter W_{set} set to the system. For physical systems in PNNs, the resulting transformation of $f_g(\cdot)$ can’t be accurately described by the parameter input W_{set} . We have shown in the Methods section that the transformation in two domains at every time instance can be described as a projection $f_p^t(\cdot) = \mathbf{D}^t f_m^t(\cdot)$. For the set of random fixed matrices in DFA defined as \mathbf{B} , we see that the directly updating the system with the error matrix as DFA leads to a multiplication in the direction of $\mathbf{D}^t \mathbf{B}$ for each iteration.

Multiplication to E is no longer based on a set of fixed matrices, thus leading to breakage of the alignment criterion for DFA. A simulation showing the performance degradation of pure DFA training on the PNN is shown in SFig.6.”

SFig.6 DFA by itself is not sufficient for training erroneous DPNNs, resulting in significant performance degradation.

Supplementary S12: Discuss the applicational scenario of different categories of training method with their codesign of the photonic system.

“S12. Different types of PhyNN training methods and their applicational scenarios

SFig.8 A decision flowchart for choosing the appropriate training method category depending on the applicational scenario.

There are many different categories of PhyNN training methods. Each category of training methods has their own unique properties and advantages. Unlike mathematical (digital) NNs, constructing a PhyNN is often a codesign process between the physical device structure, the task, the applicational scenario, and the training algorithm used. Consequently, it is not sensible to regard a single class of training methods as the elixir for every type of PhyNN construction. Instead, it is more appropriate to understand that each category has its own appropriate application scenarios. Here we present a summarized discussion on how different categories of training methods can be more suitable for different types of PhyNN codesign (implementation strategy).

SFig.8 shows a decision flowchart for choosing suitable methods for a specific PhyNN implementation technique. The construction/codesign of PhyNNs can be broadly divided into structure-centered ones and operation-centered ones. Structure-centered PhyNNs are typically the ones where the codesign logic is based on the attempt to map a device with high complexity to certain specific tasks. One great example of the structure-centered PhyNN is reservoir computing, where a device with high complexity (randomized spatial and temporal connectivity) is used for the ability to reduce a nonlinear training task to a linear separation problem. The structure-centered codesign often lacks isomorphism to a specific model description. Consequently, the common approaches are often stochastic or non-model based, such as particle swarm optimization and genetic algorithms.

On the other hand, operation-centered PhyNN implementations are cases where the codesign starts with the acceleration or partial network isomorphism in mind. PNNs are great examples of operation-based PhyNNs. For instance, the high bandwidth of photonic processing (through temporal and wavelength domain multiplexing) is often leveraged to construct PNNs which have high throughput. PNN's acceleration can most predominantly be found as the connectivity matrix (MVM operation) acceleration. In these types of operation-centered, since the PhyNN device structure are designed with the goal of acceleration in mind, these systems often exhibit sufficient levels of isomorphism for a model/connectivity description to be projected onto the system. For example, in PIC PNN,

information such as connectivity (transmission) is fundamentally determined by physical dimensions such as system topology, path length, heater resistance, fabrication tolerance, etc. Thus, it is more sensible to utilize this existing information well rather than deliberately discarding it for the equal treatment to structure-centered codesigns. For these cases, the most desirable category is often the gradient-based methods like BP for the overall fast convergence and generality.

Both codesign logics have their advantages and limitations. For example, the operation-centered codesigns come at a higher repeatability, task generality, and faster convergence (through the compatible training methods). These implementations are generally targeted at the popularization of PhyNN acceleration through reproducibility. PIC-based PNN is a great example where motivation lies in the repeatable acceleration for a wide range of users. On the other hand, structure-centered codesigns have a lower requirement of isomorphism. Yet, the trade-off would be the lower task generality, longer training convergence, and higher difficulty in repeating the system. SFig.8 provides a concise decision flowchart for choosing the appropriate training method.”

Supplementary S13: Talked about the difference in the alignment-inspired methods.

“S13. The alignment-inspired training methods: AT, PAT, and DFA

While methods such as AT, PAT, and DFA are all technically alignment-inspired methods, the concept of alignment is slightly different for each of these methods. Here we describe the difference between these alignment-inspired methods and thus discuss the convergence logic of AT.

To start with, AT and PAT would be categorized into BP-based methods while DFA is alignment-based method. The difference in the categorization originates from the update dynamics of these methods. For example, both AT and PAT utilize the BP-like layer-by-layer update logic, in which the error at the output layer is backwards fed to optimize for the loss with respect to each layer. On the other hand, DFA’s update to each layer is obtained through the multiplication between the error matrix and a set of random fixed matrices. Thus, error optimization is a process with respect to the entire deep network structure. (See SFig.9 for the comparison between the update dynamics.) This is also the reason why increasing the network complexity through adding deep layers often result in a less increased performance for DFA compared to BP. The layer-by-layer gradient optimization is the fastest loss reducing operation, which grants BP the faster convergence compared to DFA.

Consequently, despite all being alignment-inspired methods, the BP-based ones and non-BP-based ones still exhibit significant differences. For the two BP-based methods of AT and PAT, we also need to be aware of how each method leverages the concept of alignment differently. For PAT, the digital backwards pass grants a gradient estimator to the

SFig.9 The difference in update dynamics between the BP-based methods (PAT, AT) and alignment-based methods (DFA).

control updates. However, since the control-transformation mapping isn't perfect information for PAT, the update estimator is essentially a relaxed condition of alignment between the actual physical gradient δW_{phy} and the PAT update ΔW_{PAT} . If these two directions are within 90° , then we know the update behavior must be loss-reducing. On the other hand, AT's parallel digital model is a complete standalone forward-backward process trained directly by digital BP. This means that we know the loss behavior for the digital model in AT must be converging. Therefore, the utilization of the alignment concept is between the resulting digital update direction δW_{dig} and the AT update direction ΔW_{AT} . We know that δW_{dig} is a loss-reducing direction; so, within the alignment of 90° , the AT update ΔW_{AT} must also be a loss-reducing direction. This is the reason why we stated that while the learning rate can be time dependent as $\alpha(T)$, it must be the same for the digital and photonic model update within one epoch (see Results section). We can recognize that the overall update is regulated by the digital update (a direction that we know is converging), thus controlling AT's convergence rate with the BP behavior. While both δW_{phy} and δW_{dig} are directions with loss-reducing behavior, we can note that δW_{phy} is purely a local direction specific to the PNN copy while δW_{dig} is general direction that is applicable to multiple PNN copies. Consequently, it leads to the analysis in the main article's Discussion section on the generality of a single digital model in AT for multiple PNN copies to achieve the reduced distributed computational overhead."

6. In their response, the authors also compare to the finite stochastic (FS) method. There are two issues with the comparison: first, they claim a faster training time because they only require S physical propagations of each training sample, while FS requires 2S. In principle, FS could do a single-sided estimate (just measuring $f(W+\delta)$, for instance), so this is not a fundamental distinction, and in any case a factor of two is not a big advantage anyway. Second, and more importantly, the authors do not consider that AT requires backprop on a digital system, while FS can be run entirely on the optical system. Thus, even if AT is algorithmically as fast as backprop (which the authors claim, but is not proven), it is slowed down by the need to operate at the

digital system's clock speed.

In fact, the use of a digital system essentially eliminates the advantage of in-domain processing for training, as the bottleneck will always be the digital system for BP. If the authors want to claim a wall-clock training time speedup, a far more rigorous analysis, including the latency of the digital system, is required to compare the two.

They also claim AT is more energy efficient, but again the comparison is not valid because they do not consider the energy of the digital system. This impacts the comparison not only to FS, but all the algorithms in Table 1 that train only on the photonics. A more detailed comparison, which accounts for the tradeoff between improved algorithmic efficiency using a digital system and the energy advantages of training in-situ on the photonics, would be needed to support this claim.

Ultimately, I find the proposed method interesting and it is important to develop new alternatives for backpropagation, so the topic of the paper is quite timely. Unfortunately, there are enough issues with the comparisons to other work and the claims of advantage in training time/energy that I feel the current version of the manuscript is not suitable for publication in Nature Communications without significant revision.

Thank you for your comments regarding the FS method. While the photonic structure of the FS method doesn't require intermediate access, the operation of FS training is also limited by the digital system's clock speed. Even for methods that don't require a digital model during the training, it doesn't mean that the overall PNN system is operating without digital parts. For the FS methods, the training is conducted through reading out the predictions and then computing the appropriate change to the control parameters. Firstly, all the physical components in the PNN system (such as MZI, MRR, etc.) require digital control system. Therefore, even the simplest PNN with FS is also limited to the digital clock speed. The updates also need to be computed based on the digital readouts at the output layer. Thus, we think that it is not accurate to think that FS systems are digital-free and photonic-only systems.

We do agree that the computation of each iteration's update in the FS method would obviously require less energy compared to BP due to the mathematical simplicity. However, this naturally comes as a trade-off of the training convergence speed. If we think about there to be total of a P control parameters for the given network structure. The input to the network has a total training sample of S and input dimension of I . If we now assume that the number of iterations required for BP is T_{BP} , and T_{FS} for FS methods. Then the total A-D interfaces for setting the controls throughout the training would be $(P + I \cdot S)T_{BP}$ for BP methods and $(P + I \cdot S)T_{FS}$ for FS methods. We know that $T_{FS} > T_{BP}$. For a large base number of $(P + I \cdot S)$, the control for additional access can be significant, which can mean prolonged operational time.

Now, whether the system is more efficient with BP or FS is dependent on a lot of factors: the task complexity, the model dimensionality, the hyperparameters, and the power consumption relation between the photonic and digital system. For example, on photonic platforms which require more energy for component tuning, the photonic side can be the more power-hungry factor for prolonged operation. And it will be beneficial to use BP-based methods which reduce the operational time of the photonic side. From a purely A/D and I/O perspective, we can recognize that the faster convergence of BP-based methods is advantageous for cases where the model parameter number is large, and task complexity is high (as the larger models would imply a significant number of control parameter setting within each iteration). This means that while some smaller tasks might be better off with FS methods, larger and more complex tasks might be better off with BP-based methods with faster convergence. Please see S14 in the Supplementary for more discussion on this topic.

The same argument applies to the discussion on the difference between the one or two forward for the FS method. As you mentioned, it is possible to do a one-sided FS estimation of $f(W + \Delta W)$ instead of $f(W + \Delta W)$ and $f(W - \Delta W)$. But the necessary trade-off is the further increased iterations needed.

Regarding the two inputs per sample scenarios: while it might only look like a twofold increase, we need to consider that it is multiplied by the based number of sample number and input channel number (which can be very large even for the standard tasks such as MNIST, i.e. 784×60000). We have now also added a new discussion in the Supplementary (S14) to summarize the discussion in the above response.

This again reflects that it is impossible to crown a single category of training method as the “best” for all scenarios as different users might have very different goals. For example, some users might emphasize the most on the convergence speed and some other users might care about a lowered computational complexity for update. There is no real fair way to quantify the different merits enjoyed by different users (with their different goals and different availability of equipment). And very obviously, there is no single training method that can claim its superiority over other methods for all aspects. Consequently, in our revised manuscript, the Discussion section focused on the BP-based methods through a purely logical argument which can be directly compared. Instead, we think it is more appropriate for us to provide discussion on what type of methods can be more suitable for certain scenarios and PhyNN implementation strategies.

For example, for PNNs which are known to be isomorphic implementations that are operation-based (i.e. having the goal of certain operation acceleration during the codesign), a rudimentary level of model description (topology) is typically available, making it very suitable with BP-based methods. Also, taking AT as an example, the training of the digital model is independent to the local PNNs variation, and can be reapplied to multiple PNN copies, making it particularly compatible with the scalable and reproducible platforms for the goal of popularizing and promoting PNN acceleration for a wider range of audience.

Overall, we have now fully revised our Abstract, Introduction, Results, Discussion, and Supplementary. We have removed the previous table in the Introduction and refined the Discussion section to a logical argument to provide clarity to the readers. We hope that our revision now answers your comments. Again, thank you for your time to provide detailed feedback.

Places where we made revision in relation to this part of the response include:

Discussion section: Entire discussion section

“Efficiency of AT-compatible encapsulated structures

While PNNs can offer unique NN acceleration strategies that are not possible on traditional digital NNs, they also face realistic challenges since the codesign (construction) process can be limited by the demanding training operations. Here, we discuss the efficiency that can be enabled by constructing an encapsulated DPNN with AT in comparison to the truncated PNN of existing IP-BP methods.

The encapsulation of DPNN bypasses most of the AD interfaces, significantly reducing the signal conversion and data shuttling time for the sample propagation (during both training and inference). For each propagation of information through the PNN structure, we can define the extraction time, T_{extract} , as the minimum time required to obtain an output for further operation (obtaining a prediction or computing update). A simple formulation of the extraction time is the combination of the propagation time T_{prop} with the readout/conversion time $T_{\text{interface}}$ at the AD interface. The propagation time for PNN is much shorter than the interface and readout time $T_{\text{prop}} \ll T_{\text{interface}}$ (see Supplementary). Consequently, we could approximate the extraction time scaling as $O(P)$ for encapsulated systems and $O(2M - P)$ for truncated systems (each AD interface corresponds to a pair of readout-and-write operations, thus

the factor of 2). For DPNNs, $P < (2M - P)$ always holds true. The extraction time is directly related to how fast the prediction is made, and thus, the operation speed of the PNN. T_{extract} for encapsulated PNN is dependent on the specific task instead of the model complexity. Taking the MNIST 10-class task as example, P stays 10 regardless of the model's complexity (Fig.6 (a)). Consequently, the encapsulated PNN offers better compatibility overall with increasing model complexity.

Furthermore, the sequential layer-by-layer operation of a truncated DPNN imposes an increased timestep for computation that is not addressable through higher parallelism. For the N hidden layers, a total of $N + 1$ timestep is required for each single forward propagation, while there is always only 1 timestep for encapsulated DPNN (Fig.6 (b)). The constant timestep offers good compatibility with schemes of higher parallelism, such as wavelength division multiplexing.

The extraction time can also be related to the energy consumption of the photonic device during the inference. For a photonic system with power consumption of P_{photonic} , the minimum operation time is the extraction time for obtaining a prediction. Thus, the minimum energy required to operate the photonic system would be $E_{\text{photonic}} = P_{\text{photonic}} \cdot T_{\text{extract}}$. For systems where the limiting power consumption is photonic hardware, the longer the T_{extract} , the larger the energy inefficiencies are created due to the prolonged operation.

Distributed overhead of AT with repeatable application to trainable PNN structures

While PNN training methods utilizing a parallel/twin digital model can provide a more accurate estimate of the physical gradients, the typical concern is the computational overhead that might come with the application of the parallel digital model. Here we discuss how computational overhead (and the associated additional energy consumption) in AT can be avoided to a computational overhead similar to standard BP.

As mentioned in the Results section, the digital update of the parallel model can be computed independent of the local PNN's error, allowing for the same computational timestep as BP without sequential delay. The key motivation for reproducible PNN platforms such as PICs is often associated with reproducible acceleration despite the device-to-device variations. The same PNN structure might be fabricated for many copies to achieve local acceleration of a specific task. For a total of C copies of PNN devices, the overall description of each PNN system can be formulated as $f_{p,1;i} (f_{p,2;i}(\dots))$ for $i = 1, 2, \dots, C$, denoting the specific system under discussion. In the case of chip fabrication variation, the overall descriptions of these systems form a statistically relevant set. Thus, we note the equivalence of different estimation profiles and physical device copies. The generality of successful training with different profiles (see Fig.3 (d)) can also be interpreted as different copies of the PNN systems trained equally with a single digital model.

From Eq.3, the overall AT estimator is obtained through two parts: the digital parallel update (ΔW_{dig}) and the pseudo update (ΔW_{pseudo}). For a given task and initialization condition, the update of the digital model is independent of the local PNN device and can be described as $\Delta W_{\text{dig}}^{t=t'}(W_o^{t=t_0}; X, Y, \alpha^{t=t'})$ at time instance $t = t'$. On the other hand, ΔW_{pseudo} is local to the specific PNN device copy (subject to variation) as $\Delta W_{\text{pseudo}}^{t=t'}(f_{p,i}; W_o^{t=t_0}; X, Y, \alpha^{t=t'})$. The generality of a single digital model to different variations of the PNN systems means that the collection of the digital updates can be stored and reapplied to different PNN devices without the need to repeat the process of computing the digital updates (thus no need to repeat the parallel model's backpropagation). For multiple PNN systems, the forward pass in the parallel model only needs to be computed once. Effectively, the computational overhead required to utilize the digital parallel model is distributed across the PNN copies (Fig.6 (c)). Thus, the overhead of operation is significantly reduced for each individual device. When $C \gg 1$, the overall computational overhead associated with each individual PNN device is reduced towards the overhead of a standard BP process. This allows the benefits of encapsulation to be enjoyed while little overhead is added on top of the standard BP process for training of many devices."

Introduction section: Re-written and removed the table comparison

“Backpropagation²⁰ (BP) finds the gradient update to the loss function for the network parameters, offering fast training gradient convergence²¹. BP’s advantage for fast training with DNNs comes from two key aspects: BP’s root in gradient-based update logic and BP’s layer-by-layer update dynamics. BP’s robustness, generality and fast convergence make it a well-balanced industry standard approach for most digital DNN-based tasks. However, standard BP imposes requirements of highly accurate model control and knowledge. The inevitable fabrication variation inherently creates imperfect knowledge for the users. In addition, the lack of perfect control-transformation mapping on the typically noisy and lossy analogue computation platforms^{22,23} can’t satiate the high requirements of standard BP. The mismatch between the user’s interpretation and the PNN’s actual state can lead to failed training or dramatic performance degradation^{24,25}. Therefore, some non-gradient-based²⁶⁻³¹ or model-free/stochastic methods³²⁻³⁵ are proposed to bypass the requirement for formulating a model description. Nonetheless, the natural trade-off with non-gradient methods is the higher difficulty to train complex tasks and slower convergence compared to the BP-based methods. When defining the convergence of standard BP to be $O(T_0)$, the convergences of other non-gradient PNN training methods are typically $> O(T_0)$ ³⁶. PNNs, being mostly operation-centered platforms, are highly compatible with BP-based gradient methods for the typically exhibited isomorphism. (See Supplementary for more discussion on the categorization of PhyNNs.)

Many efforts have been made to search for PNN-tailored BP-based methods. The general attempt of the BP-PNN methods is to find an estimator of the device’s physical parameter gradient. The approaches which utilize a separate digital model or propagation pass for estimating the gradient update include physics-aware training³⁷ (PAT), hybrid training³⁸ (HT), and dual adaptive training³⁹ (DAT). There are also approaches which utilize the physically reversed signal input through the structure to obtain in-situ computation of the gradient⁴⁰⁻⁴². However, the common limitation of these existing BP-PNN methods is the heavy reliance on accurate intermediate layer state extraction in a deep PNN (DPNN). We call these methods intermediate-access physics-aware backpropagation (IP-BP) methods. Due to the intermediate access need, IP-BP codesigned PNNs are truncated, in which the computation acceleration is only available within a hidden layer’s structure. For fast propagation platforms like photonics, the overall operation is bottlenecked by the analogue-digital (AD) conversion interfaces^{43,44} and data shuttling⁴⁵, creating a delay growing with the network complexity. This potentially compromises the incentive of employing photonics for fast and low-delay processing. Instead, it is more desirable to construct encapsulated DPNNs (encapsulation refers to systems whose input signal is maintained within the optical analogue domain without intermediate extraction), allowing the advantage of fast processing to be enjoyed for the entire deep network structure. The number of AD access in a truncated DPNN grows as $O(2M - P)$ (M is the total number of neurons excluding the input layer, and P is the number of output layer neurons, $M > P$ for any DNN), whereas it scales as $O(P)$ for encapsulated DPNN. The access timestep is also reduced from $N + 1$ (N is the number of hidden layers) in a truncated network to 1 for an encapsulated network. (See Discussion section.)”

Supplementary: S12, S14

“S12. Different types of PhyNN training methods and their applicational scenarios

There are many different categories of PhyNN training methods. Each category of training methods has their own unique properties and advantages. Unlike mathematical (digital) NNs, constructing a PhyNN is often a codesign process between the physical device structure, the task, the applicational scenario, and the training algorithm used. Consequently, it is not sensible to regard a single class of training methods as the elixir for every type of PhyNN construction. Instead, it is more appropriate to understand that each category has its own appropriate application scenarios. Here we present a summarized discussion on how different categories of training methods can be more suitable for different types of PhyNN codesign (implementation strategy).

SFig.8 A decision flowchart for choosing the appropriate training method category depending on the applicational scenario.

SFig.8 shows a decision flowchart for choosing suitable methods for a specific PhyNN implementation technique. The construction/codesign of PhyNNs can be broadly divided into structure-centered ones and operation-centered ones. Structure-centered PhyNNs are typically the ones where the codesign logic is based on the attempt to map a device with high complexity to certain specific tasks. One great example of the structure-centered PhyNN is reservoir computing, where a device with high complexity (randomized spatial and temporal connectivity) is used for the ability to reduce a nonlinear training task to a linear separation problem. The structure-centered codesign often lacks isomorphism to a specific model description. Consequently, the common approaches are often stochastic or non-model based, such as particle swarm optimization and genetic algorithms.

On the other hand, operation-centered PhyNN implementations are cases where the codesign starts with the acceleration or partial network isomorphism in mind. PNNs are great examples of operation-based PhyNNs. For instance, the high bandwidth of photonic processing (through temporal and wavelength domain multiplexing) is often leveraged to construct PNNs which have high throughput. PNN's acceleration can most predominantly be found as the connectivity matrix (MVM operation) acceleration. In these types of operation-centered, since the PhyNN device structure are designed with the goal of acceleration in mind, these systems often exhibit sufficient levels of isomorphism for a model/connectivity description to be projected onto the system. For example, in PIC PNN, information such as connectivity (transmission) is fundamentally determined by physical dimensions such as system topology, path length, heater resistance, fabrication tolerance, etc. Thus, it is more sensible to utilize this existing information well rather than deliberately discarding it for the equal treatment to structure-centered codesigns. For these cases, the most desirable category is often the gradient-based methods like BP for the overall fast convergence and generality.

Both codesign logics have their advantages and limitations. For example, the operation-centered codesigns come at a higher repeatability, task generality, and faster convergence (through the compatible training methods). These

implementations are generally targeted at the popularization of PhyNN acceleration through reproducibility. PIC-based PNN is a great example where motivation lies in the repeatable acceleration for a wide range of users. On the other hand, structure-centered codesigns have a lower requirement of isomorphism. Yet, the trade-off would be the lower task generality, longer training convergence, and higher difficulty in repeating the system. SFig.8 provides a concise decision flowchart for choosing the appropriate training method.

S14. The trade-off between gradient-based (BP) and non-gradient-based (FS) methods

Finite stochastic (FS) methods are non-gradient-based methods often used for training. While BP offers faster convergence than FS, FS methods do have a lower update computation complexity compared to BP. However, in the implementation of PhyNNs, the overall efficiency should consider the overall consumption of the system rather than only the computation of the update. Since the exact efficiency of each method would be highly dependent on the task (due to the varying complexity and convergence behavior of the task), it would be impossible to construct an exhaustive list discussing all the existing machine learning tasks. Instead, here we formulate a generic discussion using BP and FS methods as example to highlight some of the aspects that the users should consider when deciding which category of training method would be the best for them.

We know that all PhyNNs are hybrid digital-analogue systems. Even the simplest ones that don't require digital models for training still need digital system for the parameter controls. For each iteration, we can formulate the total energy requirement as follows for BP and FS PNNs.

$$E_{BP-PNN} = P_{\text{photonic}} \cdot t_{\text{opt-BP}} + E_{BP} \cdot I_{BP} + E_{\text{interface}} \cdot I_{BP}$$

$$E_{FS-PNN} = P_{\text{photonic}} \cdot t_{\text{opt-FS}} + E_{FS} \cdot I_{FS} + E_{\text{interface}} \cdot I_{FS}$$

P_{photonic} represents the operating power of the photonic device (that is, including the component control, input signal power, etc.). $t_{\text{opt-XX}}$ is the total operational time required for each method. $E_{BP/FS}$ is the energy required to compute the update for each iteration. $I_{BP/FS}$ is the number of iterations required for a specific convergence level. $E_{\text{interface}}$ is the energy required at the interface for setting the new parameters and readouts between each iteration.

Since the photonic structure of FS and BP (via AT) can both be constructed as encapsulated networks, we can assume that the power consumption of the two systems to be roughly the same. We know that due to the simplicity of the FS update computation, the energy required for computing each update would be lower than the BP computation as $E_{FS} < E_{BP}$. Nonetheless, the natural trade-off comes with the easier computation is the slower convergence and longer operation time, $I_{BP} < I_{FS}$ and $t_{\text{opt-BP}} < t_{\text{opt-FS}}$. Depending on whether the photonic or the digital system is the more power-hungry side, one method might be more efficient than the other for a specific case. For example, in complex tasks (large I_{XX} and high $t_{\text{opt-XX}}$ baseline) with a large network structure (high power consumption of P_{photonic} and large number of A-D control interfaces, i.e. high $E_{\text{interface}}$), fast convergence is a top priority so BP methods might be more favorable. It is very likely that the exact more efficient method would vary for different combinations of PNN platform, equipment, and the digital system's specifications.

Therefore, the above analysis again shows that no single category of training method is the elixir for all types of PhyNN implementation. Depending on the end goal and the applicational scenario (also the available equipment), it is advised that the user should choose the appropriate method by considering the overall PhyNN construction rather than assuming a single category as the best.”

Response to Reviewer #3:

Reviewer #3 (Remarks to the Author):

After reviewing the author's response letter and revised manuscript, I still feel that I cannot support the publication of this work in Nature Communications.

Although the author has added a comparison with the PAT method, I believe it is not entirely fair, and the provided arguments are not very solid because the PAT method can also function without measuring intermediate states.

In each training session of AT, the same sample is input into both the simulation model and the physical system, and measurements are obtained at the final output to calculate the Loss. Due to errors in the simulation model and the physical system, the Loss values differ. Different Loss values are backpropagated through the simulation model to calculate gradients for the parameters, referred to in the paper as ΔW_{dig} and ΔW_{pseudo} , as seen in Equation (3) of the main text. The gradients used to update the parameters are a weighted sum of the first two gradients, with coefficients of 0.5 used in the paper.

The AT proposed in this paper is actually closer to the work of PAT. The AT in this paper does not involve modeling errors, and the biggest difference from PAT is that PAT updates gradients using pure ΔW_{pseudo} , while AT uses a weighted sum. I believe this method involves relatively minor changes.

Regarding the multiple mentions in the text and rebuttal that "For existing BP-based methods such as PAT, the need to readout the intermediate hidden neuron scales... AT only needs to access neuron information at the output layer...", this statement is incorrect. The reasons are: (1) The frequent conversion between AD and DA is closely related to the structure of photonic neural networks, not network training. (2) PAT can also train the network by measuring only the final output state. The full text of PAT does not state that it needs to measure intermediate states.

Dear Reviewer #3,

Thank you for your time to provide feedback on our manuscript. While we believe your comments are highly valuable, we also find that there might be some misunderstanding, especially regarding the difference between the AT and PAT methods. We hope this response addresses your questions.

You mentioned that PAT can also train with solely with the output layer state. This should be inaccurate, and the original text of PAT explicitly stated the necessity of the intermediate state readouts for their method.

For example, in the main text of the manuscript, in the Methods section (Page 8, first subsection of the Methods section), the original authors state "Thus, in addition to utilizing the output of the PNN via physical computations in the forward pass, intermediate outputs are also utilized to facilitate the computation of accurate gradients in PAT". A screenshot of the original text is also attached below:

Redacted

Also, in the Supplementary text of the PAT paper, which goes through more details on the update dynamics of the PAT method (Fig.3C on Page 9 of the Supplementary file of the PAT paper). As shown below, where the blue arrow refers to the readouts which are extracted from each hidden layer for computing the estimation of the physical gradient.

Figure Redacted

Also, when looking at the specific setup used for the PAT demonstrations, the original authors often use re-structuring or dimension-changing techniques for each layer within the deep network structure. Consequently, it is quite clear that the original paper of PAT does mention the necessity of intermediate readouts, and it is inaccurate to say that the PAT method also trains with only the output layer information.

In our manuscript, we have performed demonstrations on how the pseudo update by itself is not sufficient for getting well-performed training. We have demonstrated this both experimentally and through simulation. In the Results section, in Fig.3 (a). We can see that the line of the pseudo IP-BP method (which is training with only the pseudo update or essentially PAT with only the output layer readout), the performance is largely affected.

Similarly, in the Supplementary file of our manuscript, section “Analysis on the mixing ratio of M_{w1} and M_{w2} ” shows with simulation that choosing $M_{w1} = 0$ (i.e. only with pseudo update) can’t grant well-behaved training. We also attached the figure below:

Consequently, AT and PAT are fundamentally two very different methods, and AT shouldn’t be regarded as a simple modification from PAT.

As you have mentioned, the readouts are associated with the deep network structure of PNN and not only the training stage. In fact, this point further supports the novelty and motivation of AT. The goal of AT is to construct encapsulated PNNs that are trainable with BP-based methods for the benefits of both worlds. The truncated structure of the DPNN will affect both the training and the later inference stage (which would theoretically be operated more for real applications). Therefore, AT, being able to train an encapsulated DPNN structure while enjoying the fast convergence of BP is highly desirable as it provides benefits for operation of both the training and inference stages.

In our revised Discussion section, we went through how the truncation of a DPNN means higher access cost and difficulty to scale with higher model complexity while the encapsulated DPNN’s access is task-dependent, and the computation (sample information propagation) time grows at a much slower rate compared to the A-D conversions and data shuttling in truncated network structures.

You can find the specific parts in relation to this part of the response from:

Results section: The results on the pseudo IP-BP methods

“We first showcase AT for the complete three-class classification of the Iris dataset for Setosa, Versicolor, and Virginica. To showcase the insufficiency of the sole backward pass in the parallel model of IP-BP methods for encapsulated systems, we further define a benchmark called the pseudo IP-BP methods, referring to the scenario where IP-BP methods only have access to the output neuron information.

In-silico BP training is insufficient to train the DPNN, resulting in a significantly degraded performance of only 38%, a value near the random guessing level of the three-class task. Pseudo IP-BP has access to some level of physical response at the output layer. Consequently, the performance is slightly improved compared to pure in-silico BP, achieving 64% during training (Fig.3 (a)). However, the absence of the intermediate hidden neuron information implies that the differentiable model in the backward pass is no longer a good description of the actual physical transformation, showcasing IP-BP methods’ inability to train encapsulated PNNs.

In comparison, when AT is applied, the training accuracy is improved to 96% and testing accuracy to 94%. For reference, an ideal BP achieves the maximum training accuracy of 97% and testing accuracy of 94% for the same network structure. In both cases, AT can retrieve near ideal-BP performances. The improvement of AT accuracy (96%) from pseudo IP-BP methods (64%) showcases the necessity to employ the additional forward pass in the digital domain. The comparison between different training results can be found in Fig.3 (a).”

Supplementary: S8

“S8. Analysis on the mixing ratio of M_{w1} and M_{w2}

For most scenarios, it is sufficient to set the two mixing ratios as equal contributing of $M_{w1} = M_{w2} = 0.5$. This setting allows for the equal contribution of the information from the physical system and digital parallel model. From the simulation study, we see that the training can be sufficed with different mixing as long as the mix is not one of the extremes where only one contribution remains (i.e. $M_{w1} = 1$ or $M_{w2} = 1$). The result for varying M_{w1} is illustrated in SFig.7. All mixings apart from $M_{w1} = 0$ and $M_{w1} = 1$ can allow for training, with the equal contributing of $M_{w1} = M_{w2} = 0.5$ resulting in the highest performance.

As described in the subsection of “Asymmetrical training method”, the procedure for implementing AT includes maintaining the learning rate of the digital update and the overall AT estimator the same. The learning rate α itself can vary for different update instance t , but must be maintained the same for update to the two sets of parameters at a given time. The fixed mixing ratio during the training acts as a self-regulating process for compensating the effect of magnitude difference from the digital domain and the physical system. When one of the errors is larger than the other, we see that the overall magnitude of the AT estimator update $\alpha\Delta\mathbf{W}_{AT}^{[t]}$ will be different from the digital parallel model update of $\alpha\Delta\mathbf{W}_{dig}^{[t]}$. This means that when the physical system is further away from the loss function minimum than the digital model, the update tries to catch up with the training progress of the digital model. On the other hand, when the physical transformation is at a position which is closer to the minimum, the AT estimator slows down the process of update for retaining the relevance between the digital and AT updates.

The physical boundaries of the PNNs again help with preserving the robustness of AT (see the Methods section in the main article for first part of the discussion). If there is a significant magnitude difference between the two contributing parts of $\Delta\mathbf{W}_{dig}^{[t]}$ and $\Delta\mathbf{W}_{pseudo}^{[t]}$, the overall AT estimator might lose relevance to the digital parallel model. However, for the bounded system of PNNs, the allowed magnitude deviation is restricted by the physical boundaries as discussed in the Methods section. The training is thus well-behaved for any statistical deviation of the physical module.”

Entire Discussion section:

“Efficiency of AT-compatible encapsulated structures

While PNNs can offer unique NN acceleration strategies that are not possible on traditional digital NNs, they also face realistic challenges since the codesign (construction) process can be limited by the demanding training operations. Here, we discuss the efficiency that can be enabled by constructing an encapsulated DPNN with AT in comparison to the truncated PNN of existing IP-BP methods.

The encapsulation of DPNN bypasses most of the AD interfaces, significantly reducing the signal conversion and data shuttling time for the sample propagation (during both training and inference). For each propagation of information through the PNN structure, we can define the extraction time, $T_{extract}$, as the minimum time required to obtain an output for further operation (obtaining a prediction or computing update). A simple formulation of the extraction time is the combination of the propagation time T_{prop} with the readout/conversion time $T_{interface}$ at the AD interface. The propagation time for PNN is much shorter than the interface and readout time $T_{prop} \ll T_{interface}$ (see Supplementary). Consequently, we could approximate the extraction time scaling as $O(P)$ for encapsulated systems and $O(2M - P)$ for truncated systems (each AD interface corresponds to a pair of readout-and-write operations, thus the factor of 2). For DPNNs, $P < (2M - P)$ always holds true. The extraction time is directly related to how fast the

prediction is made, and thus, the operation speed of the PNN. T_{extract} for encapsulated PNN is dependent on the specific task instead of the model complexity. Taking the MNIST 10-class task as example, P stays 10 regardless of the model's complexity (Fig.6 (a)). Consequently, the encapsulated PNN offers better compatibility overall with increasing model complexity.

Furthermore, the sequential layer-by-layer operation of a truncated DPNN imposes an increased timestep for computation that is not addressable through higher parallelism. For the N hidden layers, a total of $N + 1$ timestep is required for each single forward propagation, while there is always only 1 timestep for encapsulated DPNN (Fig.6 (b)). The constant timestep offers good compatibility with schemes of higher parallelism, such as wavelength division multiplexing.

The extraction time can also be related to the energy consumption of the photonic device during the inference. For a photonic system with power consumption of P_{photonic} , the minimum operation time is the extraction time for obtaining a prediction. Thus, the minimum energy required to operate the photonic system would be $E_{\text{photonic}} = P_{\text{photonic}} \cdot T_{\text{extract}}$. For systems where the limiting power consumption is photonic hardware, the longer the T_{extract} , the larger the energy inefficiencies are created due to the prolonged operation.

Distributed overhead of AT with repeatable application to trainable PNN structures

While PNN training methods utilizing a parallel/twin digital model can provide a more accurate estimate of the physical gradients, the typical concern is the computational overhead that might come with the application of the parallel digital model. Here we discuss how computational overhead (and the associated additional energy consumption) in AT can be avoided to a computational overhead similar to standard BP.

As mentioned in the Results section, the digital update of the parallel model can be computed independent of the local PNN's error, allowing for the same computational timestep as BP without sequential delay. The key motivation for reproducible PNN platforms such as PICs is often associated with reproducible acceleration despite the device-to-device variations. The same PNN structure might be fabricated for many copies to achieve local acceleration of a specific task. For a total of C copies of PNN devices, the overall description of each PNN system can be formulated as $f_{p,1;i} (f_{p,2;i}(\dots))$ for $i = 1, 2, \dots, C$, denoting the specific system under discussion. In the case of chip fabrication variation, the overall descriptions of these systems form a statistically relevant set. Thus, we note the equivalence of different estimation profiles and physical device copies. The generality of successful training with different profiles (see Fig.3 (d)) can also be interpreted as different copies of the PNN systems trained equally with a single digital model.

From Eq.3, the overall AT estimator is obtained through two parts: the digital parallel update (ΔW_{dig}) and the pseudo update (ΔW_{pseudo}). For a given task and initialization condition, the update of the digital model is independent of the local PNN device and can be described as $\Delta W_{\text{dig}}^{t=t'} (W_0^{t=t_0}; X, Y, \alpha^{t=t'})$ at time instance $t = t'$. On the other hand, ΔW_{pseudo} is local to the specific PNN device copy (subject to variation) as $\Delta W_{\text{pseudo}}^{t=t'} (f_{p,i}; W_0^{t=t_0}; X, Y, \alpha^{t=t'})$. The generality of a single digital model to different variations of the PNN systems means that the collection of the digital updates can be stored and reapplied to different PNN devices without the need to repeat the process of computing the digital updates (thus no need to repeat the parallel model's backpropagation). For multiple PNN systems, the forward pass in the parallel model only needs to be computed once. Effectively, the computational overhead required to utilize the digital parallel model is distributed across the PNN copies (Fig.6 (c)). Thus, the overhead of operation is significantly reduced for each individual device. When $C \gg 1$, the overall computational overhead associated with each individual PNN device is reduced towards the overhead of a standard BP process. This allows the benefits of encapsulation to be enjoyed while little overhead is added on top of the standard BP process for training of many devices."

Discussion section, Fig.6:

Therefore, I have concerns about the methodological innovation and motivation of this paper. Regarding the method itself, my doubts mainly lie in the gradient weighting. The paper mentions that AT training requires two conditions, one of which is: "The first condition is that the AT estimator update must conform to the alignment criterion of within 90° from the ideal BP update." The intuitive explanation is that the directional angle between ΔW_{dig} and ΔW_{pseudo} must be less than 90 degrees, meaning their dot product is positive, and the two gradients must be positively correlated. However, how can this be ensured in environments with large errors? For example, when environmental errors are large, the update directions of the gradient ΔW_{dig} calculated by the simulation model and the gradient ΔW_{pseudo} calculated by the physical model may be completely opposite, resulting in parameters that cannot be updated at all. How can the convergence of the algorithm be guaranteed? Therefore, I also have concerns about the method's ability to correct errors.

To answer your question regarding the error correcting ability of AT, we would like to first start with the theoretical analysis (this was also included in the previous draft of the manuscript in the Methods section). When interpreting the concept of the alignment angle, we need to first recognize the key difference between digital model NNs and physical NNs. For PNN systems, the value encoding is bounded by the physical boundaries of the control components. For example, the optical signal power encoding can't exceed the maximum transmission of 1 for the input power. The physical boundaries of the system help protect the behavior of PNN training. This is very different from purely mathematical models where the transformation deviation between two models can take arbitrary values. In the Methods section's analysis on the robustness of AT, the possibility of breaking the 90 degrees condition for

high distortion accounting 10% of the entire tunable range is $1.5 \times 10^{-21}\%$. Even for extreme level of errors where the deviation accounts for 25% of the entire tunable range, the probability that the alignment is broken is $6.3 \times 10^{-3}\%$. The physical boundaries of the PNNs system statistically protect the robustness of the training (more details can be found in the Methods section of the main article).

This is further backed by the simulation analysis that we carried out in Fig.5 (d). We see that test accuracy maintains good performance even when pressure testing the method to extreme levels of errors. As we mentioned earlier, we can define a physically relevant region for training of PNNs. Through our simulation, we find that AT consistently provides loss reducing behavior for testing accuracies over 90% for the MNIST task for all of this range.

Both examples mentioned above are overestimations beyond the modern PNN’s quoted fabrication variation. Thus, for any realistic modern fabrication of PNNs, this would not be a problem. In fact, to be concerned about the values over this range is equivalent to saying that the fabricated system’s “on” state (encoding of 1) would deviate beyond the “off” state (encoding of 0), which is not possible for both range-bounded (e.g. MRRs) and periodically bounded systems (e.g. MZIs).

Furthermore, the concept of how methods such as DFA utilize alignment is very different from how BP-based methods (AT, PAT) utilize the idea of alignment (S13). Unlike DFA, which utilizes the concept of alignment at its limit (i.e. away from the 90° , with the multiplication to the randomized fixed matrices set), PAT and AT both utilize the concept of alignment as a relaxation to the strict gradient condition (i.e. starting from 0°), with some feedback from the physical system (the intermediate readouts with PAT and the output layer with AT). For PAT, the relaxed condition of alignment is between the device’s physical gradient δW_{phy} and the PAT update ΔW_{PAT} (since without the exact characterization, the update direction in PAT is an estimator of the actual gradient). How AT utilizes the idea of alignment is different to PAT; instead of trying to find the physical gradient, AT has a complete forward-backward set in the parallel model. The parallel model, trained by BP, is known to have a loss-converging update at δW_{dig} . The goal of AT is to align the photonic transformation subject to the AT update direction ΔW_{AT} towards the digital transformation subject to the parallel update δW_{dig} . Consequently, two aspects can be noted: 1. The update dynamic of AT is the layer-by-layer BP update dynamic. 2. The update at each time-instance is regulated by the parallel model’s BP behavior. With the $M_{w1} = M_{w2} = 0.5$ weighting, the overall update forms a self-regulating process (in terms of magnitude and direction) subject to the condition of the parallel BP update at $t = t'$. This is the reason why we call AT to be a BP-based method. We have included a new discussion in the Supplementary file (S13) to discuss this. The generality of AT’s digital model also means that the single set for digital updates can be re-applied to many

PNN copies for reduced overhead, particularly compatible with PNN’s goal to provide reproducible and repeatable NN acceleration for a wider range of users.

We hope that our response now clarifies your concerns. Thank you again for your precious time to provide comments.

More details to answer your concern about the error correction of AT can be found in:

Results section of main article (Fig. 5)

“For pressure testing the AT method, the standardized distortion varies between 0 and $2\sigma_{\text{phy}}$ to investigate the behavior under different error levels. AT shows a high error tolerance, maintaining the test accuracy for the MNIST dataset over 90% for up to twice the error seen at the experimental level (Fig.5 (d)). Due to the physical boundaries (representable range of control) of the PNN systems, we can define a physically relevant region for the system (see Methods). For all the physically relevant regions, AT shows consistent performance over 90% test accuracy while in-silico BP is heavily degraded.”

Methods section, subsections “The training mechanism of asymmetrical training estimator” and “The robustness of AT and how the physical boundaries of PNNs protects AT”

“The training mechanism of asymmetrical training estimator

AT is designed to be a BP-based method which utilizes both the information in the analogue and digital domain to achieve training. The expression for performing a standard backpropagation is given by the three equations in Eq. (7).

$$\begin{aligned} \frac{\partial L}{\partial \mathbf{z}^{[l-1]}} &= [\mathbf{W}^{[l]}]^T \cdot \frac{\partial L}{\partial \mathbf{z}^{[l]}} \otimes g^{[l-1]'}(\mathbf{z}^{[l-1]}) \\ \frac{\partial L}{\partial \mathbf{w}^{[l]}} &= \frac{\partial L}{\partial \mathbf{z}^{[l]}} \cdot [\mathbf{a}^{[l-1]}]^T \\ \frac{\partial L}{\partial \mathbf{b}^{[l]}} &= \frac{\partial L}{\partial \mathbf{z}^{[l]}} \end{aligned} \quad (7)$$

The three equations utilize chain rules to relate the error at the output layer to the parameters in the earlier layers. We have previously defined the two transformations for the digital model and the physical model as: $f_m(\mathbf{a}; \mathbf{W}, \mathbf{b}, g) \neq f_p(\mathbf{a}; \mathbf{W}, \mathbf{b}, g)$, where the insertion of the same control parameters will result in different transformations due to distortion of the transformation behavior understanding for the physical module. For each time instance, defined by the specific set of parameters, the deviation between the physical and digital transformations can be described as a projection term \mathbf{D} as $f_p^t(\cdot) = \mathbf{D}^t f_m^t(\cdot)$. We can invoke the general expression for the alignment angle between two transformations to be Eq. (8).

$$\cos(\beta) = \frac{\mathbf{B}^T \mathbf{A}}{\|\mathbf{B}^T\| \cdot \|\mathbf{A}\|} \quad (8)$$

For a total control parameter of X in an arbitrary system, we can view the loss-function space of the system as a description of a tensor for the parameter pair $(W_{\text{set}}, f_{\text{set}}(\cdot))$ for a total $2X$ dimension space. The mismatch between the parameters that we set, and the resulting transformation determines the position on different loss-function surfaces for different mismatch levels. We note that the ideal “mathematical” system of $(W_m, f_m(\cdot))$ is simply a special case where our knowledge of the expected transformation matches perfectly with the actual resulting transformation. Consequently, the physical domain of the PNN $(W_p, f_p(\cdot))$ and the digital domain $(W_m, f_m(\cdot))$ are describable within the same space. This means that by setting the control parameters asymmetrically to the two systems, the two systems can reach a similar representation in the loss-function space even though they are on different surfaces.

We can see that two things must be satisfied for the training behavior of AT to converge towards an ideal BP as if there were no error. The first condition is that the AT estimator update must conform the alignment criterion of within 90° from the ideal BP update $\delta \mathbf{W}_{\text{dig}}$ ²⁶. The second is that the AT estimator must contain sufficient physical information for the update to be relevant to the physical gradient update of the system. Consequently, we logically see the necessity

of utilizing two terms, as described by the expression of the AT estimator in Eq. (3), with the overall update as a mixture of the two domains. The parallel term makes connection to the digital model, and the pseudo term makes connection to the physical domain as a proxy for the physical gradient.

The robustness of AT and how the physical boundaries of PNNs protects AT

To ensure the robustness of the AT method such that it always trains. The first condition can be easily satisfied through using a mixing ratio M_{w1} of 0.5, such that ΔW_{AT} always aligns with δW_{dig} (for the two updates with similar magnitudes). The second criterion is essentially asking for how each individual digital transformation $f_{m,i}(\cdot)$ aligns with each individual physical transformation $f_{p,i}(\cdot)$. This thus traces back to the initial relation of the transformations at $t = 0$ ($f_m^{t=0}(\cdot)$ and $f_p^{t=0}(\cdot)$). As we have defined above, we can describe this term for each individual timestep as D^t . For $D^{t=0}$, the fulfilment of the alignment condition thus becomes a statistical problem of how much deviation can naturally exist between the two transformations when inserted with the same set of controls $W^{t=0}$ (remember that we are initializing the control for the two domains with the same set of parameters).

We see that for two unbounded systems (where the transformations can take arbitrary values), we potentially need to worry about the alignment. However, physical systems are always bounded systems with their relevant physical boundaries. For example, the signal power encoding for a neuron in PhPNN system can never have an activation that is higher than the input signal plus the amplification. Consequently, we see that physical boundaries of the PNNs enforce an upper limit for possible mismatch between the mathematical and physical transformations. For periodically bounded system such as MZIs, we return to a state with “less” mismatch when going beyond the turning point of π difference. On the other hand, for range bounded systems such as MRR or phase change materials, the action of further increase/decrease the tuning at the boundaries always returns the system to the boundary state (similar to the concept of the ladder operators in quantum mechanics).

Consequently, the robustness of AT is always statistically protected by the physical boundaries. For example, for a high level of distortion that accounts for 10% of the entire tunable range, the probability of breaking the alignment is $1.5 \times 10^{-21}\%$. For an extreme level of distortion which accounts for a quarter of the entire tunable range, the probability that alignment is broken is $6.3 \times 10^{-3}\%$. Therefore, we see that the robustness of the AT method is statistically protected by the physical boundary requirements of the PNN systems. For realistic PPM platforms, the deviation we described above far exceeds the standard three deviation variation for fabrication. Consequently, AT always offers robust training for bounded isomorphic physical modules as long as there is any statistical relevance in the estimation profile used, a condition that is easy to achieve for fabrication on existing scalable platforms.”

Supplementary S13

S13. The alignment-inspired training methods: AT, PAT, and DFA

While methods such as AT, PAT, and DFA are all technically alignment-inspired methods, the concept of alignment is slightly different for each of these methods. Here we describe the difference between these alignment-inspired methods and thus discuss the convergence logic of AT.

To start with, AT and PAT would be categorized into BP-based methods while DFA is alignment-based method. The difference in the categorization originates from the update dynamics of these methods. For example, both AT and PAT utilize the BP-like layer-by-layer update logic, in which the error at the output layer is backwards fed to optimize for the loss with respect to each layer. On the other hand, DFA’s update to each layer is obtained through the multiplication between the error matrix and a set of random fixed matrices. Thus, error optimization is a process with respect to the entire deep network structure. (See SFig.9 for the comparison between the update dynamics.) This is also the reason why increasing the network complexity through adding deep layers often result in a less increased performance for DFA compared to BP. The layer-by-layer gradient optimization is the fastest loss reducing operation, which grants BP the faster convergence compared to DFA.

Consequently, despite all being alignment-inspired methods, the BP-based ones and non-BP-based ones still exhibit significant differences. For the two BP-based methods of AT and PAT, we also need to be aware of how each method

SFig.9 The difference in update dynamics between the BP-based methods (PAT, AT) and alignment-based methods (DFA).

leverages the concept of alignment differently. For PAT, the digital backwards pass grants a gradient estimator to the control updates. However, since the control-transformation mapping isn't perfect information for PAT, the update estimator is essentially a relaxed condition of alignment between the actual physical gradient δW_{phy} and the PAT update ΔW_{PAT} . If these two directions are within 90° , then we know the update behavior must be loss-reducing. On the other hand, AT's parallel digital model is a complete standalone forward-backward process trained directly by digital BP. This means that we know the loss behavior for the digital model in AT must be converging. Therefore, the utilization of the alignment concept is between the resulting digital update direction δW_{dig} and the AT update direction ΔW_{AT} . We know that δW_{dig} is a loss-reducing direction; so, within the alignment of 90° , the AT update ΔW_{AT} must also be a loss-reducing direction. This is the reason why we stated that while the learning rate can be time dependent as $\alpha(T)$, it must be the same for the digital and photonic model update within one epoch (see Results section). We can recognize that the overall update is regulated by the digital update (a direction that we know is converging), thus controlling AT's convergence rate with the BP behavior. While both δW_{phy} and δW_{dig} are directions with loss-reducing behavior, we can note that δW_{phy} is purely a local direction specific to the PNN copy while δW_{dig} is general direction that is applicable to multiple PNN copies. Consequently, it leads to the analysis in the main article's Discussion section on the generality of a single digital model in AT for multiple PNN copies to achieve the reduced distributed computational overhead."

Reviewer #4 (Remarks to the Author):

I believe the authors have addressed my main concerns showing PAT performing below AT with new experiments. I would recommend for acceptance.

Dear Reviewer #4,

We are very glad to hear that your concerns are addressed with our revision of the manuscript. Thank you for your time to review the manuscript and provide feedback.

Message to editor and all reviewers:

We would like to thank the editor and all reviewers for their precious time spent in relation to our work. We are glad to hear that Reviewers #1, 2, and 4 are satisfied with our work and revisions. We understand that Reviewers #3 still have some further questions regarding the manuscript. We hope the newly revised manuscript now addresses the remaining concerns. The changes made in the main article are tracked as: revised content. For quoting text from the original manuscript to the response letter here, we use red text: quoted content.

The reviewers can find the specific response to their comments on the following pages:

Reviewer #2: Page 1

Reviewer #3: Page 1-16

The text color above is for specifying the comments from each reviewer.

Reviewer #2 (Remarks to the Author):

Thanks to the authors for considering my comments. The authors' revisions have addressed my remaining concerns, and I'm happy to recommend publication in Nature Communications.

Dear Reviewer #2,

We are glad to hear that our revision has addressed your remaining questions. Thank you again for your valuable time to provide feedback.

Response to Reviewer #3:

Reviewer #3 (Remarks to the Author):

I would like to thank the authors for addressing the questions raised earlier and for clarifying my misunderstanding of PAT regarding the measuring of intermediate state readouts. I also have gained a deeper understanding of this work during the revision. However, I still have a concern about the comparison between the proposed method and other methods as the comparison with the relevant PNN training methods has been removed during the revision. Besides, I am also curious about the effectiveness of the proposed method on spatial DPNNs.

1. The manuscript categorizes DAT [39] as an IB-BP method, stating that it heavily relies on intermediate layer state extraction. However, the original DAT paper explicitly notes that the use of intermediate layer states is optional during training. While providing intermediate states can enhance performance, DAT can still function effectively without them (referred to as DAT w/o IS). Moreover, DAT w/o IS has been demonstrated on both diffractive PNN and MZI-based PNN architectures. I suggest that the authors clarify this point and discuss the differences in methodology and application scope between AT and DAT w/o IS. Besides, I would also suggest using other abbreviations, instead of "AT", for the proposed asymmetrical estimator training method, as "AT" could also stand for adaptive training method proposed in [16].

Dear Reviewer #3,

Thank you for your feedback to our revision on the manuscript. We strive to present our work clearly to the readers, and we are glad to hear your previous concerns regarding the difference between AT and PAT have been addressed by our revision. We understand that you still have concerns about the difference between AT and DAT, and whether AT can be applied to diffractive optical neural networks. We hope that our revised manuscript answers your questions.

Firstly, regarding your suggestion of using an alternative abbreviation for our asymmetrical training method: in the manuscript (as well as the following part of the response), we now replace AT with AsyT to refer to our asymmetrical training method. You can now see the changes in both the text and figures for the main article and the supplementary file. Hopefully, now the readers won't mistake this abbreviation with other methods.

As you mentioned, in our previous revision of the manuscript, we have removed the table in the Introduction section stating some of the metric such as the number of access points and time step. This decision was made to allow the work to be more comprehensively and unambiguously presented to the audience as it is impossible to quantify the importance of each merit without talking about a specific task goal, equipment list, etc. For example, AsyT's low requirement of physical information extraction as well as the ability to apply a global parallel model's training updates with the local PNN copies make it particularly suitable for PhyNN system emphasizing reproducibility and propagation speed, such as integrated PNNs. However, in other instances, some users' applications might not be concerned with reproducibility or even the efficiency and resource requirements, then they might choose alternative methods. The most appropriate method for training would be dependent on the exact combination of motivation, application scenario, and even the availability of equipment. Consequently, we think that no single training method should be described as the elixir for all situations. Instead, we have included the discussion in S12 talking about how different types of methods might be better suited to different situations. Hopefully, this provides the readers with the most accurate and comprehensive discussion on the topic.

Thank you for mentioning the case of DAT w/o IS, we have now clarified the description in the main article where applicable. However, we would like to highlight that AsyT is designed with very different philosophy in comparison to DAT and is a much more lightweight solution compared to DAT for reproducible PNN systems while providing comparable performance. For each PNN device copy, the training process of DAT involves four steps (as described by the Methods section in DAT's original paper). The physical PNN system first forward-propagates the physical sample, followed by the measurement of the physical states. A digital model must then simulate the photonic network's forward propagation for a digital prediction. A similarity loss function is minimized for between the photonic structure and the digital simulated model, which is then passed through backpropagation to effectively make the digital simulation's parameter closer to the actual physical model. The system then utilizes another loss function for the specific training task for another backpropagation process with respect to the physical parameters. Therefore, for each PNN, the backpropagation process needs to be carried out twice, considerably increasing the time and energy resources required to conduct the training. Furthermore, the training in DAT is a feedback process between physical photonic system and the digital model simulation, forming a sequential step in the training process that can't be accelerated through higher parallelism. This increases the computation steps, and the overall training time compared to AsyT (assuming the same equipment). Also, the back-and-forth process during DAT's training means that for each copy of the PNN device, a new digital model must be created each time. One of the key goals of PNN is to achieve computation acceleration with reparability and reproducibility. In realistic application scenarios, the users might face the need to train many PNN copies. Thus, the training resources and cost with DAT can quickly pile up.

In comparison, unlike DAT's digital model creation, which is a detailed simulation of the photonic system, AsyT's parallel model is a simple mathematical description of the transformation (like the case with a digital forward feed), allowing for much simpler model construction. Furthermore, for training of each local copy in AsyT, the process

only involves the measurement of the physical state and its backwards feed. Since AsyT is about tuning the physical transformation of the local PNN towards the appropriate parallel model's update, the general parallel model's training updates are globally applicable for different variations of local PNN copies. AsyT only involves one loss minimization process with respect to the task. This means that computation complexity for both the forward and backward pass in AsyT is much lower than DAT. In addition, the more PNN copies to be trained, the more the computational overhead can be distributed. When there are many PNN copies, the computational resource requirement associated with each local device is reduced to a level very close to a standard digital BP process. This also means that the packaged complementary digital control system in AsyT can be of lower complexity and cost.

Overall, AsyT and DAT train with very distinct logics and have very different application scopes. For benchtop systems application setting in lab environments, where the resource requirement, energy and system complexity are typically not factors of concern, the users might choose to use DAT. However, in cases where PNNs are reproducible accelerators for the machine learning task, the lightweight and efficient characteristics of AsyT are highly desirable, especially with the upcoming advancement of edge computing and photonic networking for device near or at point of native data generation.

We have demonstrated AsyT's ability to tackle erroneous diffractive PNN, and AsyT's ability to retrieve comparable performance to DAT when utilizing much lower resources and computational complexity (the further discussions on these two parts are detailed in the next two responses; also, in supplementary S15 and S16). On top of the existing Discussion section which talks about the efficiency of AsyT, we have added a new discussion in the Supplementary file as "S16. AsyT as a lightweight and efficient option for training reproducible DPNNs" talking about the difference between AsyT and DAT, and why it is preferable to use AsyT for reproducible PNN acceleration.

The content and revision in relation to this part of the response can be found at:

Supplementary file: S16

S16. AsyT as a lightweight and efficient option for training reproducible DPNNs

SFig.11 PM: Parallel Model; PS: Photonic System; DM: Digital Model. For reproducible copies of the PNN system, AsyT applies a single PM generally to all the PS copies, whereas alternative methods have device specific digital model construction which can add computational complexity and reduce efficiency.

AsyT is designed to be a lightweight PNN training solution with the efficiency of training reproducible PNNs, which is essential for the popularization and commercialization of PNN accelerators. As mentioned in S12, different training methods each have different application scope. Compared to methods whose application scope is leaning towards the benchtop setting in a designated environment (and where the users are typically not concerned with the resources associated with the training process), AsyT has lowered computational complexity and resource requirement. The computation efficiency of AsyT originates from both the training of each device due to the simplicity of the parallel model, and the training of multiple reproducible PNN copies due to the generality of the parallel model's updates. Taking the DAT as an example for comparison, the digital model in DAT is a detailed simulation of the photonic system, in which the information transformation is encoded by the specific photonic system's characteristics ($S_n = |S_n| \exp(j\Phi_{S_n})$). In comparison, AsyT's parallel model is considered as a simple mathematical transformation, as is in the case of a digital forward propagation. This allows AsyT to save computational resources for the creation of the digital side. Furthermore, the similarity loss and task loss both need to be optimized with respect to the physical parameter in DAT. In comparison, only the task loss needs to be backpropagated for AsyT, allowing for significantly less resource associated (SFig.11 (c)) when number of parameters in the network grows. Furthermore, since a global parallel model in AsyT can be applied to different copies of local PNN devices, the computation overhead can be distributed for multiple devices (SFig.11(d)). The overhead associated with each local PNN copy is significantly reduced while the resource requirement for each copy in DAT can't be distributed. (Here the computational resources required is estimated by considering the number of FLOPs where the number of operations is scaling with number of parameters. The number of operation in the backward pass is estimated to be twice the forward pass¹⁵.) For the same error setting, AsyT can achieve comparable performance to alternative methods (92.4% vs 92.9%, SFig.10 (b)) while

significantly reducing the computational resources required (and thus the computation time for the updates). These factors show the advantages of utilizing AsyT for training reproducible PNN devices (such as based on integrated photonics). The lowered computation requirement for the complementary system also shows potentials for integrated device close to the native data generation source, paving way for upcoming computation schemes such as edge computing with photonic networking.

Supplementary File: S15

S15. Discussion on AsyT for diffractive PNNs

SFig.10 The simulation of AsyT for spatial PNNs. In realistic implementations, the system might suffer from geometric and fabrication errors. AsyT shows significant improvement in performance compared to in-silico BP.

AsyT trains the non-ideal physical structure by interpreting its general mathematical transformation description towards the parallel model, which means the generality of application on different PhyNN platforms with reasonably describable isomorphism. Here we discuss the compatibility of AsyT with diffractive PNNs through simulation to showcase the generality of AsyT method's application. The construction of the diffractive PNN includes an input controlled by a digital micro-mirror device (DMDs) for tuning each pixel of the input image. The input signal is then passed through a lens which acts as a two-dimensional FFT on the image. The Fourier transformed signal is modulated by spatial light modulators (SLMs). The combination of the modulating blocks is viewed as the deep network structure of the PNN. On top of the systematic knowledge imperfections (such as due to the non-ideal fabrication and control system), we add the effect of geometric error to the system (as the form of signal shift). For example, when a translational shifting acts on the PNN system, the resulting system experiences both a systematic error and some loss in the information (see SFig.10 (a), where part of the signal is assumed to be outside of the detection area of CCD camera). Similar to the case of the integrated PNNs, at each place of tuning (i.e. applying the physical control parameters), the mismatch between how the user interprets the transformation and the actual physical transformation is fundamentally resulting from imperfect fabrication and system construction. Random noise is also added to the input image to emulate the undesired light sources in realistic implementations. For investigating the ability of AsyT to train imperfect spatial PNN, we utilize the geometric error of five pixels' shift. The results are shown in SFig.10: in-silico BP training suffers a significant degradation in performance, with the test accuracy for the MNIST task is reduced to 39%. On the other hand, when AsyT is used for PNN training, test accuracy is improved to 92.4%, showing significant performance improvement. AsyT's generality theoretically allows the training to be conducted to different PhyNN implementation platforms.

Supplementary File: S12

S12. Different types of PhyNN training methods and their applicational scenarios

SFig.8 A decision flowchart for choosing the appropriate training method category depending on the applicational scenario.

There are many different categories of PhyNN training methods. Each category of training methods has their own unique properties and advantages. Unlike mathematical (digital) NNs, constructing a PhyNN is often a codesign process between the physical device structure, the task, the applicational scenario, and the training algorithm used. Consequently, it is not sensible to regard a single class of training methods as the elixir for every type of PhyNN construction. Instead, it is more appropriate to understand that each category has its own appropriate application scenarios. Here we present a summarized discussion on how different categories of training methods can be more suitable for different types of PhyNN codesign (implementation strategy).

SFig.8 shows a decision flowchart for choosing suitable methods for a specific PhyNN implementation technique. The construction/codesign of PhyNNs can be broadly divided into structure-centered ones and operation-centered ones. Structure-centered PhyNNs are typically the ones where the codesign logic is based on the attempt to map a device with high complexity to certain specific tasks. One great example of the structure-centered PhyNN is reservoir computing, where a device with high complexity (randomized spatial and temporal connectivity) is used for the ability to reduce a nonlinear training task to a linear separation problem. The structure-centered codesign often lacks isomorphism to a specific model description. Consequently, the common approaches are often stochastic or non-model based, such as particle swarm optimization and genetic algorithms.

On the other hand, operation-centered PhyNN implementations are cases where the codesign starts with the acceleration or partial network isomorphism in mind. PNNs are great examples of operation-based PhyNNs. For instance, the high bandwidth of photonic processing (through temporal and wavelength domain multiplexing) is often leveraged to construct PNNs which have high throughput. PNN's acceleration can most predominantly be found as the connectivity matrix (MVM operation) acceleration. In these types of operation-centered, since the PhyNN device structure are designed with the goal of acceleration in mind, these systems often exhibit sufficient levels of isomorphism for a model/connectivity description to be projected onto the system. For example, in PIC PNN,

information such as connectivity (transmission) is fundamentally determined by physical dimensions such as system topology, path length, heater resistance, fabrication tolerance, etc. Thus, it is more sensible to utilize this existing information well rather than deliberately discarding it for the equal treatment to structure-centered codesigns. For these cases, the most desirable category is often the gradient-based methods like BP for the overall fast convergence and generality.

Both codesign logics have their advantages and limitations. For example, the operation-centered codesigns come at a higher repeatability, task generality, and faster convergence (through the compatible training methods). These implementations are generally targeted at the popularization of PhyNN acceleration through reproducibility. PIC-based PNN is a great example where motivation lies in the repeatable acceleration for a wide range of users. On the other hand, structure-centered codesigns have a lower requirement of isomorphism. Yet, the trade-off would be the lower task generality, longer training convergence, and higher difficulty in repeating the system. SFig.8 provides a concise decision flowchart for choosing the appropriate training method.

Discussion Section (Partially revised, please see main article for place of revision):

Distributed overhead of AsyT with repeatable application to trainable PNN structures

While PNN training methods utilizing a parallel/twin digital model can provide a more accurate estimate of the physical gradients, the typical concern is the computational overhead that might come with the application of the parallel digital model. Here we discuss how computational overhead (and the associated additional energy consumption) in AsyT can be avoided to a computational overhead similar to standard BP.

As mentioned in the Results section, the digital update of the parallel model can be computed independent of the local PNN’s error, allowing for the same computational timestep as BP without sequential delay. The key motivation for reproducible PNN platforms such as PICs is often associated with reproducible acceleration despite the device-to-device variations. The same PNN structure might be fabricated for many copies to achieve local acceleration of a specific task. For a total of C copies of PNN devices, the overall description of each PNN system can be formulated as $f_{p,1;i} (f_{p,2;i}(\dots))$ for $i = 1, 2, \dots, C$, denoting the specific system under discussion. In the case of chip fabrication variation, the overall descriptions of these systems form a statistically relevant set. Thus, we note the equivalence of different estimation profiles and physical device copies. The generality of successful training with different profiles (see Fig.3 (d)) can also be interpreted as different copies of the PNN systems trained equally with a single digital model.

From Eq.3, the overall AsyT estimator is obtained through two parts: the digital parallel update (ΔW_{dig}) and the pseudo update (ΔW_{pseudo}). For a given task and initialization condition, the update of the digital model is independent of the local PNN device and can be described as $\Delta W_{\text{dig}}^{t=t'} (W_0^{t=t_0}; X, Y, \alpha^{t=t'})$ at time instance $t = t'$. On the other hand, ΔW_{pseudo} is local to the specific PNN device copy (subject to variation) as $\Delta W_{\text{pseudo}}^{t=t'} (f_{p;i}, W_0^{t=t_0}; X, Y, \alpha^{t=t'})$. The generality of a single digital model to different variations of the PNN systems means that the collection of the digital updates can be stored and reapplied to different PNN devices without the need to repeat the process of computing the digital updates (thus no need to repeat the parallel model’s backpropagation). For multiple PNN systems, the forward pass in the parallel model only needs to be computed once. Effectively, the computational overhead required to utilize the digital parallel model is distributed across the PNN copies (Fig.6 (e)). In comparison, existing method requires detailed and specific digital model construction for each PNN copy (see Supplementary for more discussion). Thus, the overhead of operation in AsyT is significantly reduced for each individual device. When $C \gg 1$, the overall computational overhead associated with each individual PNN device is reduced towards the overhead of a standard BP process. This allows the benefits of encapsulation to be enjoyed while little overhead is added on top of the standard BP process for training of many devices.

Abstract:

Abstract

Photonic neural networks (PNNs) are fast in-propagation and high bandwidth paradigms that aim to popularize reproducible NN acceleration with higher efficiency and lower cost. However, the training of PNN is known to be challenging, where the device-to-device and system-to-system variations create imperfect knowledge of the PNN. Despite backpropagation (BP)-based training algorithms being the industry standard for their robustness, generality, and fast gradient convergence for digital training, existing PNN-BP methods rely heavily on accurate intermediate state extraction or extensive computational resources for deep PNNs (DPNNs). The truncated photonic signal propagation and the computation overhead bottleneck DPNN’s operation efficiency and increase system construction cost. Here, we introduce the asymmetrical training (AsyT) method, tailored for encapsulated DPNNs, where the signal is preserved in the analogue photonic domain for the entire structure. AsyT offers a lightweight solution for DPNNs with minimum readouts, fast and energy-efficient operation, and minimum system footprint. AsyT’s ease of operation, error tolerance, and generality aim to promote PNN acceleration in a widened operational scenario despite the fabrication variations and imperfect controls. We demonstrated AsyT for encapsulated DPNN with integrated photonic chips, repeatably enhancing the performance from in-silico BP for different network structures and datasets.

Introduction Section: Clarified descriptions:

Many efforts have been made to search for PNN-tailored BP-based methods. The general attempt of the BP-PNN methods is to find an estimator of the device’s physical parameter gradient. The approaches which utilize a separate digital model or propagation pass for estimating the gradient update include physics-aware training³⁷ (PAT), hybrid training³⁸ (HT), and dual adaptive training³⁹ (DAT). There are also approaches which utilize the physically reversed signal input through the structure to obtain in-situ computation of the gradient⁴⁰⁻⁴². However, the common limitation of these existing BP-PNN methods is either the heavy reliance on accurate intermediate layer state extraction in a deep PNN (DPNN) or the extensive requirement of computational resources to simulate the training model. We call methods with the need to access internal information intermediate-access physics-aware backpropagation (IP-BP) methods. Due to the intermediate access need, IP-BP codesigned PNNs are truncated, in which the computation acceleration is only available within a hidden layer’s structure. For fast propagation platforms like photonics, the overall operation is bottlenecked by the analogue-digital (AD) conversion interfaces^{43,44} and data shuttling⁴⁵, creating a delay growing with the network complexity. This potentially compromises the incentive of employing photonics for fast and low-delay processing. Instead, it is more desirable to construct encapsulated DPNNs (encapsulation refers to systems whose input signal is maintained within the optical analogue domain without intermediate extraction), allowing the advantage of fast processing to be enjoyed for the entire deep network structure. The number of AD access in a truncated DPNN grows as $O(2M - P)$ (M is the total number of neurons excluding the input layer, and P is the number of output layer neurons, $M > P$ for any DNN), whereas it scales as $O(P)$ for encapsulated DPNN. The access timestep is also reduced from $N + 1$ (N is the number of hidden layers) in a truncated network to 1 for an encapsulated network. (See Discussion section.) It is also undesirable for the training algorithm’s complexity to be too high, as the additional computations increase training time, energy consumption, and cost. The high complexity would also imply the difficulty for application on reproducible platforms, as the complementary control system’s overhead is too high.

Consequently, it is desirable to find methods that bypass the IP-BP methods’ access limitations, train with reduced computation complexity, and still enjoy BP’s general fast convergence behavior. Here, we present the asymmetrical training (AsyT) method for well-balanced training of DPNN systems with single-structured design (doesn’t require special structure for training, needs the minimum photonic component requirement same as the inference for reduced cost), encapsulated computation (the signal is maintained within the optical domain for the entire DPNN structure), error-tolerance (tolerant to PNN’s device-to-device and system-to-system variations), and low computation resource requirement (comparable to standard BP). AsyT utilizes an additional forward pass in the digital parallel model compared to the existing estimator approaches³⁷⁻³⁹ to eliminate the requirement for accessing intermediate DPNN state information (for total access point of P). AsyT’s gradient-like layer-specific update dynamic is compatible with standard update optimizers such as gradient descent^{46,47} or Adam⁴⁸. AsyT’s goal to increase training efficiency and reduce cost is in unison with PNN’s general purpose for faster and cheaper computing⁹.

Alternation of the abbreviation from “AT” to “AsyT”:

The entire main article and supplementary wherever applicable.

2. The authors only compared AT with PAT (I assume the 'pseudo IP-BP' in Fig. 3 represents PAT without intermediate states) and in-silico BP. How does the performance of AT compare to DAT w/o IS? DAT models errors using an additional error prediction network. Although this increases training complexity, it also enhances training performance. I guess that the performance of AT may represent a trade-off between PAT and DAT w/o IS.

Thank you for your comment on this point. The key motivation for promoting PhyNN acceleration in general is to look for differential advantages over the existing mainstream digital NN accelerators. As we mentioned in S12 of the Supplementary, when the PhyNN implementation is through a structure-centered codesign, the lack of an isomorphic description means that the training process often disregards the efficiency and resource requirement. However, for PhyNN with reproducibility in mind (such as the case of PNN), then a good balance is highly desirable. For AsyT and DAT, which aim at systems with isomorphism (DAT requires detailed knowledge of the photonic structure for building the digital model to compute similarity loss, thus high level of isomorphism), we must also be aware of the efficiency and resources put in during the training process. This is also the reason why when there is isomorphism for the description of the physical system, the users typically wouldn't choose purely stochastic methods as that requires too much computation power.

In our newly added discussion in the Supplementary file S15, we discussed the effectiveness of AsyT in spatial PNNs and showed that it can achieve comparable performances (AsyT 92.4% vs DAT 92.9%) while AsyT's computation complexity and resource requirement is much lower due to the difference in working principle between AsyT and DAT. Furthermore, in our manuscript (both experimental and simulation), we compared the performance of AsyT with the maximum achievable performance of ideal BP to show consistent retrieval of comparable performances. As mentioned in the last part of the response, we have now included a more detailed discussion of the computation overhead difference between AsyT and DAT in S16 of the Supplementary file.

We highly respect the work done by the authors of the DAT paper. However, it is very clear that their vision of the application scope for PNN is very different from ours. In the case of DAT, as the diffractive PNN is their main point of discussion, they are likely thinking about a scenario setting closer to the laboratory where the computation complexity and the associated energy consumption is not of the users' concern. However, for our discussion of the integrated DPNNs, which has the purpose of compact acceleration at many different places, the efficiency, computation complexity, and cost of the co-packaged digital complementary device become highly important. We do think that AsyT's low computation complexity (and thus the faster update calculation) and the generality of parallel mode's description for re-applicability to local PNN devices is highly desirable for promoting PNNs to wider audience.

The content and revision with relation to this part of the response can be found at:

Supplementary file: S16

S16. AsyT as a lightweight and efficient option for training reproducible DPNNs

SFig.11 PM: Parallel Model; PS: Photonic System; DM: Digital Model. For reproducible copies of the PNN system, AsyT applies a single PM generally to all the PS copies, whereas alternative methods have device specific digital model construction which can add computational complexity and reduce efficiency.

AsyT is designed to be a lightweight PNN training solution with the efficiency of training reproducible PNNs, which is essential for the popularization and commercialization of PNN accelerators. As mentioned in S12, different training methods each have different application scope. Compared to methods whose application scope is leaning towards the benchtop setting in a designated environment (and where the users are typically not concerned with the resources associated with the training process), AsyT has lowered computational complexity and resource requirement. The computation efficiency of AsyT originates from both the training of each device due to the simplicity of the parallel model, and the training of multiple reproducible PNN copies due to the generality of the parallel model's updates. Taking the DAT as an example for comparison, the digital model in DAT is a detailed simulation of the photonic system, in which the information transformation is encoded by the specific photonic system's characteristics ($S_n = |S_n| \exp(j\Phi_{S_n})$). In comparison, AsyT's parallel model is considered as a simple mathematical transformation, as is in the case of a digital forward propagation. This allows AsyT to save computational resources for the creation of the digital side. Furthermore, the similarity loss and task loss both need to be optimized with respect to the physical parameter in DAT. In comparison, only the task loss needs to be backpropagated for AsyT, allowing for significantly less resource associated (SFig.11 (c)) when number of parameters in the network grows. Furthermore, since a global parallel model in AsyT can be applied to different copies of local PNN devices, the computation overhead can be distributed for multiple devices (SFig.11(d)). The overhead associated with each local PNN copy is significantly reduced while the resource requirement for each copy in DAT can't be distributed. (Here the computational resources required is estimated by considering the number of FLOPs where the number of operations is scaling with number of parameters. The number of operation in the backward pass is estimated to be twice the forward pass¹⁵.) For the same error setting, AsyT can achieve comparable performance to alternative methods (92.4% vs 92.9%, SFig.10 (b)) while

significantly reducing the computational resources required (and thus the computation time for the updates). These factors show the advantages of utilizing AsyT for training reproducible PNN devices (such as based on integrated photonics). The lowered computation requirement for the complementary system also shows potentials for integrated device close to the native data generation source, paving way for upcoming computation schemes such as edge computing with photonic networking.

Supplementary File: S15

S15. Discussion on AsyT for diffractive PNNs

SFig.10 The simulation of AsyT for spatial PNNs. In realistic implementations, the system might suffer from geometric and fabrication errors. AsyT shows significant improvement in performance compared to in-silico BP.

AsyT trains the non-ideal physical structure by interpreting its general mathematical transformation description towards the parallel model, which means the generality of application on different PhyNN platforms with reasonably describable isomorphism. Here we discuss the compatibility of AsyT with diffractive PNNs through simulation to showcase the generality of AsyT method's application. The construction of the diffractive PNN includes an input controlled by a digital micro-mirror device (DMDs) for tuning each pixel of the input image. The input signal is then passed through a lens which acts as a two-dimensional FFT on the image. The Fourier transformed signal is modulated by spatial light modulators (SLMs). The combination of the modulating blocks is viewed as the deep network structure of the PNN. On top of the systematic knowledge imperfections (such as due to the non-ideal fabrication and control system), we add the effect of geometric error to the system (as the form of signal shift). For example, when a translational shifting acts on the PNN system, the resulting system experiences both a systematic error and some loss in the information (see SFig.10 (a), where part of the signal is assumed to be outside of the detection area of CCD camera). Similar to the case of the integrated PNNs, at each place of tuning (i.e. applying the physical control parameters), the mismatch between how the user interprets the transformation and the actual physical transformation is fundamentally resulting from imperfect fabrication and system construction. Random noise is also added to the input image to emulate the undesired light sources in realistic implementations. For investigating the ability of AsyT to train imperfect spatial PNN, we utilize the geometric error of five pixels' shift. The results are shown in SFig.10: in-silico BP training suffers a significant degradation in performance, with the test accuracy for the MNIST task is reduced to 39%. On the other hand, when AsyT is used for PNN training, test accuracy is improved to 92.4%, showing significant performance improvement. AsyT's generality theoretically allows the training to be conducted to different PhyNN implementation platforms.

Supplementary File: S12

S12. Different types of PhyNN training methods and their applicational scenarios

SFig.8 A decision flowchart for choosing the appropriate training method category depending on the applicational scenario.

There are many different categories of PhyNN training methods. Each category of training methods has their own unique properties and advantages. Unlike mathematical (digital) NNs, constructing a PhyNN is often a codesign process between the physical device structure, the task, the applicational scenario, and the training algorithm used. Consequently, it is not sensible to regard a single class of training methods as the elixir for every type of PhyNN construction. Instead, it is more appropriate to understand that each category has its own appropriate application scenarios. Here we present a summarized discussion on how different categories of training methods can be more suitable for different types of PhyNN codesign (implementation strategy).

SFig.8 shows a decision flowchart for choosing suitable methods for a specific PhyNN implementation technique. The construction/codesign of PhyNNs can be broadly divided into structure-centered ones and operation-centered ones. Structure-centered PhyNNs are typically the ones where the codesign logic is based on the attempt to map a device with high complexity to certain specific tasks. One great example of the structure-centered PhyNN is reservoir computing, where a device with high complexity (randomized spatial and temporal connectivity) is used for the ability to reduce a nonlinear training task to a linear separation problem. The structure-centered codesign often lacks isomorphism to a specific model description. Consequently, the common approaches are often stochastic or non-model based, such as particle swarm optimization and genetic algorithms.

On the other hand, operation-centered PhyNN implementations are cases where the codesign starts with the acceleration or partial network isomorphism in mind. PNNs are great examples of operation-based PhyNNs. For instance, the high bandwidth of photonic processing (through temporal and wavelength domain multiplexing) is often leveraged to construct PNNs which have high throughput. PNN's acceleration can most predominantly be found as the connectivity matrix (MVM operation) acceleration. In these types of operation-centered, since the PhyNN device structure are designed with the goal of acceleration in mind, these systems often exhibit sufficient levels of isomorphism for a model/connectivity description to be projected onto the system. For example, in PIC PNN,

information such as connectivity (transmission) is fundamentally determined by physical dimensions such as system topology, path length, heater resistance, fabrication tolerance, etc. Thus, it is more sensible to utilize this existing information well rather than deliberately discarding it for the equal treatment to structure-centered codesigns. For these cases, the most desirable category is often the gradient-based methods like BP for the overall fast convergence and generality.

Both codesign logics have their advantages and limitations. For example, the operation-centered codesigns come at a higher repeatability, task generality, and faster convergence (through the compatible training methods). These implementations are generally targeted at the popularization of PhyNN acceleration through reproducibility. PIC-based PNN is a great example where motivation lies in the repeatable acceleration for a wide range of users. On the other hand, structure-centered codesigns have a lower requirement of isomorphism. Yet, the trade-off would be the lower task generality, longer training convergence, and higher difficulty in repeating the system. SFig.8 provides a concise decision flowchart for choosing the appropriate training method.

Results section:

Generalization and repeatability of the AsyT method

We further investigate AsyT’s generalization and repeatability across different datasets and larger network structures through simulation. The level of information mismatch is replicated based on the experiments for reliable simulation results (see Methods for determining the error level and how to apply it to the simulation). The experimental level mismatch is defined as one standardized unit of distortion σ_{phy} . For an overestimation of the error, we choose a noise floor of 10 dB signal-to-noise ratio where applicable, a higher noise than the typical operation of analogue systems.

We use three datasets of MNIST, FMNIST, and KMNIST for the simulation analysis. We implement an MLP model with the network structure of [784, 256, 256, 10] for all three cases. With the AsyT method, the test accuracies of the encapsulated deep networks are 95.8% (97.0%), 87.5% (88.6%), and 85.6% (87.6%) for MNIST, FMNIST, and KMNIST, respectively (the value in the bracket indicates the maximum ideal BP performance). AsyT repeatably achieves performance comparable to the ideal error-free BP model. For comparison, the in-silico BP performances are significantly degraded to 25.7%, 16.6%, and 12.8%, respectively. The performance improvement by employing AsyT is significant. The results are shown in Fig.5 (a-c).

3. The evaluation of AT for large-scale DPNNs is based solely on simulations of MZI-based DPNNs with a noise level of 10 dB signal-to-noise ratio. While this is valuable, the applicability of AT to other types of DPNNs, such as spatial architectures (e.g., diffractive DPNNs), which are prone to geometric and fabrication errors, remains unaddressed. How effective is AT in handling geometric and fabrication errors present in spatial DPNNs like diffractive DPNNs? Would the authors consider conducting simulation or physical experiments with discussions on a type of spatial DPNN to evaluate the effectiveness of AT?

Thanks again for this comment. We would like to highlight that for our investigation of AsyT for the larger integrated DPNN systems, we have already considered systematic errors due to fabrication errors as well as random errors in forms of SNR. Therefore, the current simulation results in the Result section (and in Supplementary) have already included the effect of fabrication error. As we mentioned in the Results section, the imperfect knowledge of the state for an isomorphic PhyNN fundamentally arises from the inability to perfectly manufacture and tune the structure. If the fabrication is 100% perfect, then we shouldn’t expect any systematic error when applying the physical parameters to the system (apart from the degradation due to random errors).

As you mentioned, the context for systematic error can be different for integrated and diffractive PNNs. Spatial PNNs are likely to suffer from systematic errors such as geometric errors. To illustrate the generality of applying AsyT to isomorphic PhyNNs, we have now added a simulation and discussion of AsyT for spatial PNNs in S15 of

the Supplementary file. For the different types of systematic errors (discussed in the DAT paper), we chose the translational error for our investigation. This is because the other two types of errors (rotation and scaling) still represent a systematic error (as denoted in the Methods section by interpreting the system’s condition as different functional spaces). However, the translational shift represents a case that is not commonly observed for integrated PNN systems, which is the partial loss of sample information when the signal is outside of the detection region. (As the integrated PNNs are signal confinements, losing part of the information through the propagation would mean defective hardware to start with.) We think this scenario is the most interesting and appropriate for a demonstration here as a proof of concept since it provides a case that is not observable in integrated systems. For this simulation, we consider the fabrication error (imperfect control) and random error (noise, as can be seen by the fluctuating background for SFig.10 (a)) as usual, with the addition of the geometric error. As shown in SFig.10 (b) of S15 in the Supplementary, when considering a translational shift of 5 pixels, AsyT can improve the testing accuracy from 39% of the directly employed in-silico BP to 92.4%. Also, the performance achieved is comparable to DAT’s performance (92.9%) while significantly reducing the computation complexity and resources required (S16). Since the fundamental logic of AsyT is to describe the system by the encodable physical boundary and not the absolute value, for most isomorphic systems, where the physical transformation can be described as a functional form of the interpreted ideal control behavior, the process should be applicable.

Overall, we would like to thank you for your precious time providing feedback on our manuscript. We have revised our content based on your comments and suggestions. We hope the newly revised manuscript now addresses your remaining concerns.

The content and revision in relation to this part of the response can be found at:

Supplementary File: S15

S15. Discussion on AsyT for diffractive PNNs

SFig.10 The simulation of AsyT for spatial PNNs. In realistic implementations, the system might suffer from geometric and fabrication errors. AsyT shows significant improvement in performance compared to in-silico BP.

AsyT trains the non-ideal physical structure by interpreting its general mathematical transformation description towards the parallel model, which means the generality of application on different PhyNN platforms with reasonably describable isomorphism. Here we discuss the compatibility of AsyT with diffractive PNNs through simulation to showcase the generality of AsyT method’s application. The construction of the diffractive PNN includes an input controlled by a digital micro-mirror device (DMDs) for tuning each pixel of the input image. The input signal is then passed through a lens which acts as a two-dimensional FFT on the image. The Fourier transformed signal is modulated by spatial light modulators (SLMs). The combination of the modulating blocks is viewed as the deep network structure of the PNN. On top of the systematic knowledge imperfections (such as due to the non-ideal fabrication and control system), we add the effect of geometric error to the system (as the form of signal shift). For example, when a

translational shifting acts on the PNN system, the resulting system experiences both a systematic error and some loss in the information (see SFig.10 (a), where part of the signal is assumed to be outside of the detection area of CCD camera). Similar to the case of the integrated PNNs, at each place of tuning (i.e. applying the physical control parameters), the mismatch between how the user interprets the transformation and the actual physical transformation is fundamentally resulting from imperfect fabrication and system construction. Random noise is also added to the input image to emulate the undesired light sources in realistic implementations. For investigating the ability of AsyT to train imperfect spatial PNN, we utilize the geometric error of five pixels' shift. The results are shown in SFig.10: in-silico BP training suffers a significant degradation in performance, with the test accuracy for the MNIST task is reduced to 39%. On the other hand, when AsyT is used for PNN training, test accuracy is improved to 92.4%, showing significant performance improvement. AsyT's generality theoretically allows the training to be conducted to different PhyNN implementation platforms.

Supplementary file: S16

S16. AsyT as a lightweight and efficient option for training reproducible DPNNs

SFig.11 PM: Parallel Model; PS: Photonic System; DM: Digital Model. For reproducible copies of the PNN system, AsyT applies a single PM generally to all the PS copies, whereas alternative methods have device specific digital model construction which can add computational complexity and reduce efficiency.

AsyT is designed to be a lightweight PNN training solution with the efficiency of training reproducible PNNs, which is essential for the popularization and commercialization of PNN accelerators. As mentioned in S12, different training methods each have different application scope. Compared to methods whose application scope is leaning towards the benchtop setting in a designated environment (and where the users are typically not concerned with the resources associated with the training process), AsyT has lowered computational complexity and resource requirement. The computation efficiency of AsyT originates from both the training of each device due to the simplicity of the parallel model, and the training of multiple reproducible PNN copies due to the generality of the parallel model's updates.

Taking the DAT as an example for comparison, the digital model in DAT is a detailed simulation of the photonic system, in which the information transformation is encoded by the specific photonic system’s characteristics ($S_n = |S_n| \exp(j\Phi_{S_n})$). In comparison, AsyT’s parallel model is considered as a simple mathematical transformation, as is in the case of a digital forward propagation. This allows AsyT to save computational resources for the creation of the digital side. Furthermore, the similarity loss and task loss both need to be optimized with respect to the physical parameter in DAT. In comparison, only the task loss needs to be backpropagated for AsyT, allowing for significantly less resource associated (SFig.11 (c)) when number of parameters in the network grows. Furthermore, since a global parallel model in AsyT can be applied to different copies of local PNN devices, the computation overhead can be distributed for multiple devices (SFig.11(d)). The overhead associated with each local PNN copy is significantly reduced while the resource requirement for each copy in DAT can’t be distributed. (Here the computational resources required is estimated by considering the number of FLOPs where the number of operations is scaling with number of parameters. The number of operation in the backward pass is estimated to be twice the forward pass¹⁵.) For the same error setting, AsyT can achieve comparable performance to alternative methods (92.4% vs 92.9%, SFig.10 (b)) while significantly reducing the computational resources required (and thus the computation time for the updates). These factors show the advantages of utilizing AsyT for training reproducible PNN devices (such as based on integrated photonics). The lowered computation requirement for the complementary system also shows potentials for integrated device close to the native data generation source, paving way for upcoming computation schemes such as edge computing with photonic networking.

Methods section:

Simulation of PNN training with AsyT method

For determining the experimental level error to be applied to the simulations, we characterize the actual transmission behaviors of the MZI cells and compare how much they deviate from the estimation profile used. The grey lines in the background of Fig.2 (e) and (f) show the actual transmission behaviors of the MZI cells. We define the worst-case deviation between the leftmost and rightmost transmissions of the MZIs as one standardized distortion of $1 \sigma_{\text{photon}}$. This is intended to serve as an overestimation of the realistic deviation that we see in realistic implementations.

For adding the distortion to the simulation, apart from the systematic distortion in our knowledge of the transformation, we also add other physical noise and fluctuations to the model. The overall expression for the transformation of the physical module is given as Eq. (9).

$$f_p^t(W^t) = (I + P_{\text{sys}}^t) \otimes (f_m^t(W^t) + N_{\text{init}}) + N_{\text{rand}}^t \quad (9)$$

The systematic distortion in the transformation at each time step is defined by P_{sys}^t , where the \otimes denotes the element wise Hadamard product for linear overestimation of the deviation across tunable range. The transformation is subject to some fixed initial distortion of N_{init} . At each instance the information is passed through the physical structure, a randomized error N_{rand} is added to represent the signal degradation. See Supplementary for more information on the simulation.

For the three datasets of MNIST, FMNIST, and KMNIST, we choose one-hot encoding for the labels with the network structure of [784, 256, 256, 10]. Each sample image has an original 28×28 pixels resolution, which is flattened to 784 input features. The hidden layers are connected by ReLU activation functions emulating on-chip MRR nonlinearities¹⁹, while the output activation is a SoftMax layer. The learning rate is varied depending on the task, but within a general range of 3×10^{-5} to 1×10^{-4} . We use a simple gradient descent optimizer for our simulations, alternative optimizer can also be used in conjunction with the AsyT estimator. We use a batch size of 600 for training. At each epoch, the data is shuffled to avoid memorization of the dataset. All the simulations are written with Python predominantly using Numpy⁵⁸, Tensorflow⁵⁹, Sklearn⁶⁰, and Pytorch⁶¹ packages.

Message to editor and all reviewers:

We would like to again thank the editor and all reviewers for their precious time spent in relation to our work. We are glad to hear that all Reviewers now are satisfied with our revision.

Reviewer #3 (Remarks to the Author):

The authors have addressed my concerns during the revision. Therefore, I can now recommend the publication of this work in Nature Communications.

Thank you for your time to provide comments and suggestions throughout the revision process. We are delighted to hear that your concerns have been addressed by our revision.